# DEGREE-AWARE SPIKING GRAPH DOMAIN ADAPTATION FOR CLASSIFICATION

## ABSTRACT

Spiking Graph Networks (SGNs) have garnered significant attraction from both researchers and industry due to their ability to address energy consumption challenges in graph classification. However, SGNs are only effective for in-distribution data and cannot tackle out-of-distribution data. In this paper, we first propose the domain adaptation problem in SGNs, and introduce a novel framework named **De**gree-aware **S**piking **G**raph **D**omain **A**daptation for Classification (DeSGDA). The proposed DeSGDA addresses the spiking graph domain adaptation problem by three aspects: node degree-aware personalized spiking representation, adversarial feature distribution alignment, and pseudo-label distillation. First, we introduce the personalized spiking representation method for generating degree-dependent spiking signals. Specifically, the threshold of triggering a spike is determined by the node degree, allowing this personalized approach to capture more expressive information for classification. Then, we propose the graph feature distribution alignment module that is adversarially trained using membrane potential against a domain discriminator. Such an alignment module can efficiently maintain high performance and low energy consumption in the case of inconsistent distribution. Additionally, we extract consistent predictions across two spaces to create reliable pseudo-labels, effectively leveraging unlabeled data to enhance graph classification performance. Extensive experiments on benchmark datasets validate the superiority of the proposed DeSGDA compared with competitive baselines.

## 1 INTRODUCTION

Spiking Graph Networks (SGNs) (Zhu et al., 2022; Xu et al., 2021b) are a specialized type of artificial neural network engineered to process graph information by mimicking the human brain. SGNs transform static and real-valued graph features into discrete spikes by simulating neurons' charging and discharging cycles, facilitating spike-based representations for graph node classification. Notably, SGNs excel in capturing semantic spiking representations with low energy consumption, which proves advantageous for event-based processing tasks (Yao et al., 2021) such as object recognition (Gu et al., 2020; Li et al., 2021b), real-time data analysis (Zhu et al., 2020; Bauer et al., 2019), and graph classification (Li et al., 2023; Zhu et al., 2022; Xu et al., 2021b).

Currently, SGNs are usually tested within the same distribution as the training dataset (Li et al., 2023; Yin et al., 2024; Duan et al., 2024). However, in realistic scenarios, the testing set can have different distributions from the training set, and such a distribution shift may lead to a degradation in performance. For instance, Electroencephalography (EEG) data (Binnie & Prior, 1994; Biasiucci et al., 2019), typically represented as a graph structure with nodes for neurons and edges for connections, is ideally processed by bio-inspired SGNs that mimic neuronal charging and discharging. Despite the suitability, EEGs often exhibit varying distributions over time or among different groups (Zhao et al., 2020; 2021; Wang et al., 2022), leading to suboptimal performance of models trained on specific distributions when applied to others. This significant issue underscores the necessity of exploring domain adaptation for spiking graphs. Traditionally, SNNs transfer learning methods (Zhan et al., 2021; Zhang et al., 2021; Zhan et al., 2024; Guo et al., 2024) have been applied in event-based or computer vision scenarios. However, there's no existing research on spiking graph domain adaptation.

In this paper, we address the development of energy-efficient SGNs tailored for scenarios involving distribution shifts. Both domain adaptation and SGNs are particularly well-suited for real-world

applications where data distributions vary across environments, and efficient processing of graph-structured, dynamic data under resource constraints is crucial. These challenges are common across numerous fields that require solutions capable of handling distribution shifts while minimizing energy consumption. However, designing an effective spiking graph domain adaptation framework is non-trivial due to the following major challenges: (1) *How to meticulously design an SGN under the circumstance of domain shift?* SGNs usually utilize a global threshold for the firing of each node (Xu et al., 2021a; Yin et al., 2024; Zhao et al., 2024). However, we observe that the degree of each node influences the difficulty of triggering spikes. Specifically, nodes with high degrees can integrate more information from neighbors, making it easier for membrane potential to accumulate and trigger a spike. Conversely, nodes with lower degrees are more challenging to reach the firing threshold, denoted as the inflexible architecture challenge. (2) *How to design a framework that effectively addresses spiking graph domain adaptation for classification?* Current research primarily focuses on graph node classification within the same distribution (Li et al., 2023; Yao et al., 2023; Duan et al., 2024). However, spike-based graph classification under domain shift remains unexplored. (3) *How to guarantee the stability of the proposed framework?* Though some works have been proposed to address the spiking transfer learning challenges (Zhan et al., 2021; Zhang et al., 2021; Zhan et al., 2024), there is still no theoretical research on spiking graphs under domain shift.

To tackle these challenges, we propose a framework named **De**gree-aware **S**piking **G**raph **D**omain **A**daptation for Classification (DeSGDA), which comprises three components: degree-aware personalized spiking representation, graph feature distribution alignment, and pseudo-label distillation. To address the first challenge, we establish variable node thresholds based on their degrees. By adaptively updating these thresholds, we can achieve a more expressive and personalized spiking representation for each node. Then, we introduce a adversarial feature distribution alignment module that is adversarially trained using membrane potential against a domain discriminator. To further enhance performance, we extract consistent predictions from different spaces to generate reliable pseudo-labels. Additionally, to explore the generalization ability of the proposed DeSGDA, we first propose the error bound for spiking graph domain adaptation and demonstrate that our pseudo-label distillation module effectively reduces this upper bound. In summary, we utilize simple yet effective techniques to address a novel problem while providing insightful analysis of the background mechanisms and model capabilities of our proposed method.

Our contributions can be summarized as follows: (1) **Problem Formulation:** We first introduce the problem of spiking graph domain adaptation for classification, which is non-trivial due to the challenges of the inflexible architecture of SGNs and theoretical deficiency. (2) **Novel Architecture:** We propose DeSGDA, a framework that efficiently learns personalized spiking representations for nodes using degree-aware thresholds and aligns domain distributions through adversarial training on membrane potential. Furthermore, we utilize pseudo-label distillation to improve the performance further. (3) **Theoretical Analysis:** To guarantee the stability of DeSGDA, we provide theoretical proof of the error bound for spiking graph domain adaptation. Furthermore, we demonstrate that DeSGDA maintains a lower theoretical bound than standard spiking graph domain adaptation through the effective use of the pseudo-label distillation module. (4) **Extensive Experiments**. We evaluate the proposed DeSGDA on extensive spiking graph domain adaptation learning datasets, which shows that our proposed DeSGDA outperforms the variety of state-of-the-art methods.

## 2 RELATED WORK

**Spiking Graph Networks (SGNs).** SGNs are a specialized type of neural network that combines Spiking Neural Networks (SNNs) with Graph Neural Networks (GNNs), preserving energy efficiency while achieving competitive performance in various graph tasks (Li et al., 2023; Yao et al., 2023; Duan et al., 2024). Existing research on SGNs focuses on capturing the dynamic temporal information contained within graphs and enhancing model scalability. For instance, Xu et al. (2021a) utilizes spatial-temporal feature normalization within SNNs to effectively process dynamic graph data, ensuring robust learning and improved predictive performance. Zhao et al. (2024) propose a method that dynamically adapts to evolving graph structures and relationships through a novel architecture that updates node representations in real time. Additionally, Yin et al. (2024) adapts SNNs to dynamic graph settings and employs implicit differentiation for the node classification task. However, existing methods still suffer from data distribution shift issues when training and testing data come from different domains, resulting in degraded model performance and generalization. To address this, we propose a novel domain adaptation method based on SGNs to tackle these challenges.

**Spiking Transfer Learning.** Spiking transfer learning focuses on adjusting SNNs to handle data distribution shifts across various domains effectively. Recent advances in spiking transfer learning have been extensively applied in vision tasks, enhancing model performance while maintaining energy efficiency (Zhan et al., 2021; Zhang et al., 2021; Zhan et al., 2024). For instance, Guo et al. (2024) leverages a Jaccard attention mechanism within SNNs to effectively adapt to target domains without requiring source domain data. Similarly, He et al. (2024) facilitates the transfer of learned representations from static to dynamic event-based domains by adapting SNNs to process temporal information. Additionally, Zhan et al. (2024) converts RGB images into spike-based neuromorphic data, enabling SNNs to process visual information across various domains efficiently. However, the difficulty of graph topologies makes it infeasible to apply spiking transfer learning to SGNs directly. To this end, we introduce a specialized domain adaptation method tailored for SGNs.

## 3 PRELIMINARIES

**Bound for Graph Domain Adaptation (GDA).** Applying GDA with optimal transport (OT), if the covariate shift holds on representations that $\mathbb{P}_S(Y|Z) = \mathbb{P}_T(Y|Z)$, the target risk $\epsilon_T(h, \hat{h})$ is bounded with the theorem:

**Theorem 1** *(You et al., 2023) Assuming that the learned discriminator is $C_g$-Lipschitz continuous as described in (Redko et al., 2017), and the graph feature extractor $f$ (also referred to as GNN) is $C_f$-Lipschitz that $||f||_{Lip} = \max_{G_1, G_2} \frac{||f(G_1) - f(G_2)||_2}{\eta(G_1, G_2)} = C_f$ for some graph distance measure $\eta$. Let $\mathcal{H} := \{h : \mathcal{G} \to \mathcal{Y}\}$ be the set of bounded real-valued functions with the pseudo-dimension $Pdim(\mathcal{H}) = d$ that $h = g \circ f \in \mathcal{H}$, with probability at least $1 - \delta$ the following inequality holds:*

$$\epsilon_T(h, \hat{h}) \leq \hat{\epsilon}_S(h, \hat{h}) + \sqrt{\frac{4d}{N_S} \log(\frac{eN_S}{d}) + \frac{1}{N_S} \log(\frac{1}{\delta})} + 2C_f C_g W_1(\mathbb{P}_S(G), \mathbb{P}_T(G)) + \omega,$$

*where the (empirical) source and target risks are $\hat{\epsilon}_S(h, \hat{h}) = \frac{1}{N_S} \sum_{n=1}^{N_S} |h(G_n) - \hat{h}(G_n)|$ and $\epsilon_T(h, \hat{h}) = \mathbb{E}_{\mathbb{P}_T(G}\{|h(G) - \hat{h}(G)|\}$, respectively, where $\hat{h} : \mathcal{G} \to \mathcal{Y}$ is the labeling function for graphs and $\omega = \min_{||g||_{Lip} \leq C_g, ||f||_{Lip} \leq C_f} \{\epsilon_S(h, \hat{h}) + \epsilon_T(h, \hat{h})\}$. The first Wasserstein distance is defined as (Villani et al., 2009): $W_1(\mathbb{P}, \mathbb{Q}) = \sup_{||g||_{Lip} \leq 1} \{\mathbb{E}_{\mathbb{P}_S(Z)} g(Z) - \mathbb{E}_{\mathbb{P}_T(Z)} g(Z)\}$.*

The comprehensive justification of the OT-based graph domain adaptation bound demonstrates that the generalization gap relies on both the domain divergence $2C_f C_g W_1(\mathbb{P}_S(G), \mathbb{P}_T(G))$ and model discriminability $\omega$.

**Spiking Graph Networks.** In contrast to traditional artificial neural networks, SGNs (Xu et al., 2021a; Zhu et al., 2022) convert input data into binary spikes over time, with each neuron in the SGNs maintaining a membrane potential that accumulates input spikes. A spike is produced as an output when the membrane potential exceeds a threshold, which is formulated as:

$$u_{\tau+1,i} = \lambda(u_{\tau,i} - V_{th} s_{\tau,i}) + \sum_j w_{ij} \mathcal{A}(A, s_{\tau,j}) + b, \quad s_{\tau+1,i} = \mathbb{H}(u_{\tau+1,i} - V_{th}), \quad (1)$$

where $\mathbb{H}(x)$ is the Heaviside function, which is the non-differentiable spiking function. $\mathcal{A}$ is the graph aggregation operation, and $A$ is the adjacency matrix of graph. $s_{\tau,i}$ denotes the binary spike train of neuron $i$, and $\lambda$ is the constant. $w_{ij}$ and $b$ are the weights and bias of each neuron.

## 4 METHODOLOGY

This work studies the spiking graph domain adaptation problem and proposes a new approach DeSGDA. DeSGDA consists of three parts: **Degree-aware personalized spiking representation** utilizes different thresholds for different degrees, effectively addressing the inflexible architecture challenge; **Adversarial distribution alignment** uses the adversarial training on membrane potential against a domain discriminatory to align distribution between different domains, and **Pseudo-label distillation** further applies the pseudo-label to enhance model performance. We provide the theoretical guarantee of DeSGDA to ensure the effectiveness. The overview of DeSGDA is shown in Figure 1.

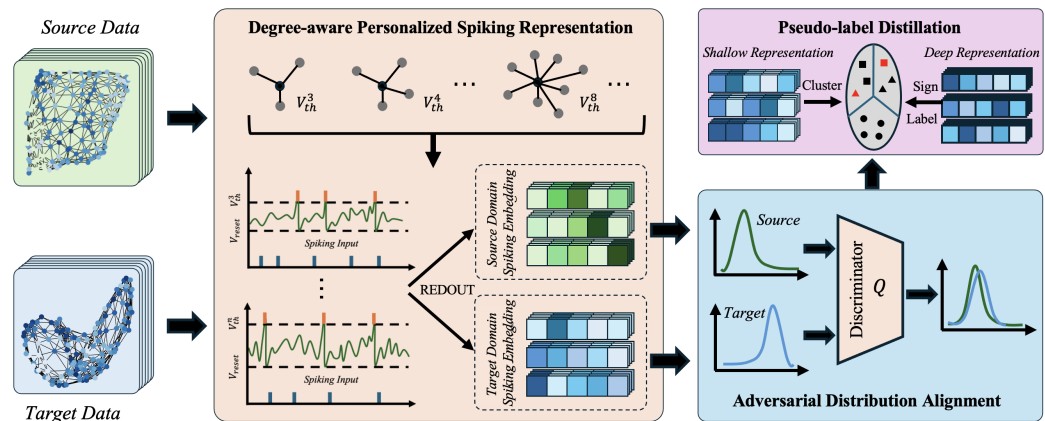

Figure 1: An overview of the proposed DeSGDA. To achieve personalized spiking representations, DeSGDA employs adaptive thresholds based on node degrees, enabling the generation of tailored spiking representations. To align domain distributions, DeSGDA leverages adversarial training on membrane potentials to counter domain discrimination. Furthermore, DeSGDA utilizes pseudo-labeling to identify and select reliable samples, thereby enhancing overall model performance.

**Problem Setup.** Given a graph $G = (V, E, \mathbf{X})$ with the node set $V$, the edge set $E$, and the node attribute matrix $\mathbf{X}$. Denote $S$ as the binary input sampled from Bernoulli distribution with probability of $\mathbf{X}$. The labeled source domain is denoted as $\mathcal{D}^s = \{(G_i^s, y_i^s)\}_{i=1}^{N_s}$, where $y_i^s$ denotes the labels of $G_i^s$; the unlabeled target domain is $\mathcal{D}^t = \{G_j^t\}_{j=1}^{N_t}$, where $N^s$ and $N^t$ denote the number of source graphs and target graphs. Both domains share the same label space $\mathcal{Y}$ but have different distributions in the graph space. We aim to train a spiking graph model using labeled source graphs and unlabeled target graphs to achieve superior performance in the target domain.

## 4.1 DEGREE-AWARE PERSONALIZED SPIKING REPRESENTATION

In this part, we first study the disadvantages of directly applying SNNs to graphs and then propose the degree-aware personalized spiking representation. Existing SGNs (Li et al., 2023; Yao et al., 2023; Duan et al., 2024) usually employ a global threshold for membrane potential firing. However, the global threshold can lead to the *inflexible architecture* issue since nodes with higher degrees are more likely to trigger spikes than those with lower degrees. As shown in Eq. 1, nodes with higher degrees have more neighbors, and the aggregation operation allows for more significant feature accumulation, making it easier for these nodes to trigger spikes compared to those with fewer neighbors. To alleviate this issue, we propose the degree-aware thresholds and iteratively update their values.

Specifically, we first *set* all the degrees of nodes in the source domain graphs, i.e., $D^s = set(D_1^s \cup \cdots \cup D_{N_s}^s)$, where $D_i$ denotes the degree set of graph $G_i^s$, and $set(\cdot)$ operation is an unordered sequence of non-repeating elements. Considering that low-degree nodes are more challenging to trigger while high-degree nodes trigger more easily, we propose setting higher thresholds for high-degree nodes and lower thresholds for low-degree nodes, which is formulated as:

$$s_\tau^{d_i^s} = \mathbb{H}(u_\tau - V_{th}^{d_i^s}), \quad S^{d_i^s} = avg(s_\tau^{d_i^s}), \quad V_{th}^{d_i^s} = (1 - \alpha)V_{th}^{d_i^s} + \alpha S^{d_i^s}, \tag{2}$$

where $V_{th}^{d_i^s}$ is the threshold of degree $d_i^s \in D^s$, initially set to $V_{th}$, and $\alpha$ is a hyper-parameter. The $avg(\cdot)$ operation takes the average of spiking representation with degree $d_i^s$. Consequently, high-degree nodes tend to achieve high $S^{d_i^s}$, which leads to an iterative increase in the threshold corresponding to degree $d_i^s$ and conversely for lower-degree nodes. To further explore the background mechanism of the hypothesis, we have the following analysis.

**Hypothesis 1** *In graph spiking networks, nodes with low-degree are more challenging to trigger while high-degree nodes trigger more easily.*

The details analysis are introduced in Appendix A. With different thresholds for different node degrees, we can obtain the personalized node spiking representation $\mathbf{s}_{v \in G_i^s}^{d_j^s}$. Then, we summarize

all node representations with a readout function into the graph-level representation and output the prediction with a multi-layer perception (MLP) classifier:

$$\mathbf{s}_i = \text{READOUT}\left(\left\{\mathbf{s}_v^{d_j^s}\right\}_{v \in G_i^s}\right), \quad \hat{y}_i^s = H(\mathbf{s}_i), \tag{3}$$

where $\hat{y}_i$ is the predicted result and $H(\cdot)$ is the classifier. After that, the source classification loss is:

$$\mathcal{L}_S = \mathbb{E}_{G_i^s \in \mathcal{D}^s} l(y_i^s, \hat{y}_i^s), \tag{4}$$

where $l(\cdot)$ is the loss function and $y_i^s$ is the ground truth of the $i$-th graph $G_i^s$ in the source domain.

However, in the scenario of domain adaptation, two significant issues remain unresolved. The first issue is that degrees in the target domain may be unseen in the source, rendering the thresholds ineffective for these degrees. The second is that the thresholds in the target may differ from those in the source, simply applying the source domain's thresholds could lead to performance degradation.

To alleviate the first issue, we initialize the threshold $V_{th}^{d_i^t}$ with the same value, where $d_i^t \notin D^s$. Then, with the training process of adversarial alignment, we iteratively update the threshold for degree $d_i^t$ with Eq. 2. To address the second issue, we incorporate the pseudo-label distillation module in Section 4.3 to guide the update of source degree thresholds on the target domain.

## 4.2 ADVERSARIAL DISTRIBUTION ALIGNMENT

To eliminate the discrepancy between the source and target domains, we propose the adversarial distribution alignment module. Specifically, for each source graph $G_i^s$ and target graph $G_i^t$, we use the degree-aware personalized spiking GNNs-based encoder $F(\cdot)$ and semantic classifier $H(\cdot)$ to produce predicted labels. Then, a domain discriminator $Q(\cdot)$ is trained to distinguish features from the source and target domains. The encoder and classifier are adversarial trained to align the feature spaces of the source and target domains.

$$\mathcal{L}_{AD} = \mathbb{E}_{G_i^s \in \mathcal{D}^s} \log Q\left(F(G_i^s), H(G_i^s)|V_{th}^{D^s}\right) + \mathbb{E}_{G_j^t \in \mathcal{D}^t} \log\left(1 - Q\left(F\left(G_j^t\right), H(G_j^t)|V_{th}^{D^s}\right)\right).$$

However, the degree in the target domain may be unseen by the source. Thus, we further initialize the threshold with $V_{th}^{d_j^t}$ and $d_j^t \notin D^s$, which is formulated as:

$$\begin{aligned}
\mathcal{L}_{AD} =& \mathbb{E}_{\substack{G_j^t \in \mathcal{D}^t \\ D_j^t \subset D^s}} \log\left(1 - Q\left(F\left(G_j^t\right), H(G_j^t)|V_{th}^{D^s}\right)\right) + \mathbb{E}_{G_i^s \in \mathcal{D}^s} \log Q\left(F(G_i^s), H(G_i^s)|V_{th}^{D^s}\right) \\
&+ \mathbb{E}_{\substack{G_j^t \in \mathcal{D}^t \\ \exists d_j^t \notin D^s}} \log\left(1 - Q\left(F\left(G_j^t\right), H(G_j^t)|V_{th}^{D^s}, V_{th}^{D^t}\right)\right),
\end{aligned} \tag{5}$$

where $D^t = \{d_i^t | d_i^t \in D^t, d_i^t \notin D^s\}$. Then, we iteratively update $V_{th}^{D^t}$ with Eq. 2 on each latency. Furthermore, we present an upper bound on the adversarial distribution alignment.

**Theorem 2** *Assuming that the learned discriminator is $C_g$-Lipschitz continuous as described in Theorem 1, the graph feature extractor $f$ (also referred to as GNN) is $C_f$-Lipschitz that $||f||_{Lip} = \max_{G_1, G_2} \frac{||f(G_1) - f(G_2)||_2}{\eta(G_1, G_2)} = C_f$ for some graph distance measure $\eta$ and the loss function bounded by $C > 0$. Let $\mathcal{H} := \{h : \mathcal{G} \to \mathcal{Y}\}$ be the set of bounded real-valued functions with the pseudo-dimension $Pdim(\mathcal{H}) = d$ that $h = g \circ f \in \mathcal{H}$, and provided the spike training data set $S_n = \{(\mathbf{X}_i, y_i) \in \mathcal{X} \times \mathcal{Y}\}_{i \in [n]}$ drawn from $\mathcal{D}^s$, with probability at least $1 - \delta$ the following inequality :*

$$\begin{aligned}
\epsilon_T(h, \hat{h}_T(\mathbf{X})) \leq& \hat{\epsilon}_S(h, \hat{h}_S(\mathbf{S})) + 2\mathbb{E}\left[\sup\sup\frac{1}{N_S}\sum_{i=1}^{N_S}\epsilon_i h(\mathbf{X}_i, y_i, p_i)\right] + C\sqrt{\frac{ln(2/\delta)}{N_S}} \\
&+ \min\left(|\epsilon_S(h, \hat{h}_S(\mathbf{X})) - \epsilon_S(h, \hat{h}_T(\mathbf{X}))|, |\epsilon_T(h, \hat{h}_S(\mathbf{X})) - \epsilon_T(h, \hat{h}_T(\mathbf{X}))|\right) \\
&+ 2C_f C_g W_1\left(\mathbb{P}_S(G), \mathbb{P}_T(G)\right),
\end{aligned} \tag{6}$$

*where the (empirical) source and target risks are $\hat{\epsilon}_S(h, \hat{h}(\mathbf{S})) = \frac{1}{N_S}\sum_{n=1}^{N_S}|h(\mathbf{S}_n) - \hat{h}(\mathbf{S}_n)|$ and $\epsilon_T(h, \hat{h}(\mathbf{X})) = \mathbb{E}_{\mathbb{P}_T(G}\{|h(G) - \hat{h}(G)|\}$, respectively, where $\hat{h} : \mathcal{G} \to \mathcal{Y}$ is the labeling function*

*for graphs and $\omega = \min_{||g||_{Lip} \leq C_g, ||f||_{Lip} \leq C_f}\{\epsilon_S(h, \hat{h}(\mathbf{X})) + \epsilon_T(h, \hat{h}(\mathbf{X}))\}$, $\epsilon_i$ is the Rademacher variable and $p_i$ is the $i^{th}$ row of $\mathbf{P}$, which is the probability matrix with:*

$$\mathbf{P}_{kt} = \begin{cases} \exp\left(\frac{u_k(t) - V_{th}}{\sigma(u_k(t) - u_{reset})}\right), & if \quad u_\theta \leq u(t) \leq V_{th}, \\ 0, & if \quad u_{reset} \leq u_k(t) \leq u_\theta. \end{cases} \tag{7}$$

Theorem 2 proves the generalization bound of spiking graph domain adaptation. More details can be found in Appendix B.

### 4.3 PSEUDO-LABEL DISTILLATION FOR DISCRIMINATION LEARNING

To further address the variance in thresholds between the target and source domains, we incorporate the pseudo-label distillation module into the DeSGDA framework. With reliable pseudo-labels, we can effectively update the source degree thresholds in the target domain.

The goal of the pseudo-label distilling procedure is to keep those examples and their corresponding pseudo-labels from the deep feature space that aligns with the shallow feature space. Specifically, we denote $\mathbf{s}'^t_i$ as the shallow spiking graph representation on the $L'$-th layer, where $L' < L$, and $\hat{y}^t_i$ as the prediction of graph $G^t_i$ on the $L$-th layer. Then, to enhance alignment between the shallow and deep feature spaces and facilitate the generation of more accurate pseudo-labels, we cluster the shallow features $\mathbf{s}'^t$ into $C$ clusters and each cluster $\mathcal{E}_j$ includes graphs $\{G^t_j\}$. After that, we find the dominating labels $e_r$ in the cluster, i.e., $\max_r |\{\mathcal{E}_r : e_r = \hat{y}^t_j\}|$, and remove other instances with the same pseudo-label but in different clusters. Formally, the pseudo-labels are signed with:

$$\mathcal{P} = \left\{ (G^t_j, \hat{y}^t_j) : e_j = \max_r |\{\mathcal{E}_r : e_r = \hat{y}^t_j\}| \right\}. \tag{8}$$

Finally, we utilize the distilled pseudo-labels to guide the update of source degree thresholds on the target domain with Eq. 2, and to direct classification in the target domain:

$$\mathcal{L}_T = \mathbb{E}_{G^t_j \in \mathcal{P}} l\left(H(\mathbf{s}^t_j), \hat{y}^t_j\right), \tag{9}$$

where $H(\cdot)$ and $\mathbf{s}^t_j$ are the classifier and spiking graph representation, respectively, which are defined in Eq. 3. $l(\cdot)$ is the loss function, and we implement it with cross-entropy loss.

**Theorem 3** *Under the assumption of Theorem 1, we further assume that there exists a small amount of i.i.d. samples with pseudo labels $\{(G_n, Y_n)\}_{n=1}^{N'_T}$ from the target distribution $\mathbb{P}_T(G, Y)$ ($N'_T \ll N_S$) and bring in the conditional shift assumption that domains have different labeling function $\hat{h}_S \neq \hat{h}_T$ and $\max_{G_1, G_2} \frac{|\hat{h}_D(G_1) - \hat{h}_D(G_2)|}{\eta(G_1, G_2)} = C_h \leq C_f C_g (D \in \{S, T\})$ for some constant $C_h$ and distance measure $\eta$, and the loss function bounded by $C > 0$. Let $\mathcal{H} := \{h : \mathcal{G} \to \mathcal{Y}\}$ be the set of bounded real-valued functions with the pseudo-dimension $Pdim(\mathcal{H}) = d$, and provided the spike training data set $S_n = \{(\mathbf{X}^s_i, y^s_i)\}_{i \in [n]}$, with probability at least $1 - \delta$ the following inequality holds:*

$$\epsilon_T(h, \hat{h}_T(\mathbf{X})) \leq \frac{N'_T}{N_S + N'_T} \hat{\epsilon}_T(h, \hat{h}_T(S)) + \frac{N_S}{N_S + N'_T} \Bigg( \hat{\epsilon}_S(h, \hat{h}_S(S)) + 2C_f C_g W_1\left(\mathbb{P}_S(G), \mathbb{P}_T(G)\right)$$

$$+ 2\mathbb{E}\left[\sup \frac{1}{N_S} \sum_{i=1}^{N_S} \epsilon_i h(\mathbf{X}_i, y_i, p_i)\right] + C\sqrt{\frac{ln(2/\delta)}{N_S}}$$

$$+ \min\left(|\epsilon_S(h, \hat{h}_S(\mathbf{X}))) - \epsilon_S(h, \hat{h}_T(\mathbf{X})))|, |\epsilon_T(h, \hat{h}_S(\mathbf{X}))) - \epsilon_T(h, \hat{h}_T(\mathbf{X})))|\right) \Bigg)$$

$$\leq \hat{\epsilon}_S(h, \hat{h}_S(S)) + 2\mathbb{E}\left[\sup \frac{1}{N_S} \sum_{i=1}^{N_S} \epsilon_i h(\mathbf{X}_i, y_i, p_i)\right] + C\sqrt{\frac{ln(2/\delta)}{N_S}}$$

$$+ 2C_f C_g W_1\left(\mathbb{P}_S(G), \mathbb{P}_T(G)\right) + \omega', \tag{10}$$

Table 1: The graph classification results (in %) on PROTEINS under edge density domain shift (source→target). P0, P1, P2, and P3 denote the sub-datasets partitioned with edge density. **Bold** results indicate the best performance.

| Methods | P0→P1 | P1→P0 | P0→P2 | P2→P0 | P0→P3 | P3→P0 | P1→P2 | P2→P1 | P1→P3 | P3→P1 | P2→P3 | P3→P2 | Avg. |
|---|---|---|---|---|---|---|---|---|---|---|---|---|---|
| WL subtree | 68.7 | 82.3 | 50.7 | 82.3 | 58.1 | 83.8 | 64.0 | 74.1 | 43.7 | 70.5 | 71.3 | 60.1 | 67.5 |
| GCN | 73.4±0.2 | 83.5±0.3 | 57.6±0.2 | 84.2±1.8 | 24.0±0.1 | 16.6±0.4 | 57.6±0.2 | 73.7±0.4 | 24.0±0.1 | 26.6±0.2 | 39.9±0.9 | 42.5±0.1 | 50.3 |
| GIN | 62.5±4.7 | 74.9±3.7 | 53.0±4.6 | 59.6±4.2 | 73.7±0.8 | 64.7±3.4 | 60.6±2.7 | 69.8±0.6 | 31.1±2.8 | 63.1±3.4 | 72.3±2.7 | 64.6±1.4 | 62.5 |
| GMT | 73.4±0.3 | 83.5±0.2 | 57.6±0.1 | 83.5±0.3 | 24.0±0.1 | 83.5±0.1 | 57.4±0.2 | 73.4±0.2 | 24.1±0.1 | 73.4±0.3 | 24.0±0.1 | 57.6±0.2 | 59.6 |
| CIN | 74.5±0.2 | 84.1±0.5 | 57.8±0.2 | 82.7±0.9 | 75.6±0.6 | 79.2±2.2 | 61.5±2.7 | 74.0±1.0 | 75.5±0.8 | 72.5±2.1 | 76.0±0.3 | 60.9±1.2 | 72.9 |
| SpikeGCN | 71.8±0.8 | 79.5±1.3 | 63.8±1.0 | 78.9±1.4 | 68.6±1.1 | 76.5±1.8 | 62.3±2.2 | 72.1±1.5 | 68.1±2.1 | 67.2±1.9 | 69.2±2.1 | 64.2±1.8 | 70.2 |
| DRSGNN | 72.6±0.6 | 80.1±1.6 | 63.1±1.4 | 79.5±1.8 | 70.4±1.9 | 78.6±2.1 | 64.1±1.7 | 70.7±2.3 | 67.8±1.6 | 65.6±1.4 | 71.3±1.3 | 62.1±1.0 | 70.5 |
| CDAN | 72.2±1.8 | 82.4±1.6 | 59.8±2.1 | 76.8±2.4 | 69.3±4.1 | 71.8±3.7 | 64.4±2.5 | 74.3±0.4 | 46.3±2.0 | 69.8±1.8 | 74.4±1.7 | 62.6±2.3 | 68.7 |
| ToAlign | 73.4±0.1 | 83.5±0.2 | 57.6±0.1 | 83.5±0.2 | 24.0±0.3 | 83.5±0.4 | 57.6±0.1 | 73.4±0.1 | 24.0±0.2 | 73.4±0.2 | 24.0±0.1 | 57.6±0.3 | 59.6 |
| MetaAlign | 75.5±0.9 | 84.9±0.6 | 64.8±1.6 | **85.9±1.1** | 69.3±2.7 | 83.3±0.6 | 68.7±1.2 | 74.2±0.7 | 73.3±3.3 | 72.2±0.9 | 69.9±1.8 | 63.6±2.3 | 73.8 |
| DEAL | 76.5±0.4 | 83.1±0.4 | 67.5±1.3 | 77.6±1.8 | 76.0±0.2 | 80.1±2.7 | 66.1±1.3 | 75.4±1.5 | 42.3±4.1 | 68.1±3.7 | 73.1±2.2 | 67.8±1.2 | 71.1 |
| CoCo | 75.5±0.2 | 84.2±0.4 | 59.8±0.5 | 83.4±0.2 | 73.6±2.3 | 81.6±2.4 | 65.8±0.3 | 76.2±0.2 | 75.8±0.2 | 71.1±2.1 | 76.1±0.2 | 67.1±0.6 | 74.2 |
| SGDA | 63.8±0.6 | 65.2±1.3 | 66.7±1.0 | 59.1±1.5 | 60.1±0.8 | 64.4±1.2 | 65.2±0.7 | 63.9±0.9 | 64.5±0.6 | 61.1±1.3 | 58.9±1.4 | 64.9±1.2 | 63.2 |
| DGDA | 58.7±0.8 | 59.9±1.2 | 57.1±0.6 | 57.9±0.8 | 59.2±1.3 | 58.9±0.4 | 61.1±1.2 | 60.3±1.6 | 58.6±0.9 | 57.5±1.2 | 58.4±0.5 | 62.3±1.5 | 59.2 |
| A2GNN | 65.4±1.3 | 66.3±1.1 | 68.2±1.4 | 66.3±1.2 | 65.4±0.7 | 65.9±0.9 | 66.9±1.3 | 65.4±1.2 | 65.6±0.9 | 65.5±1.2 | 66.1±2.0 | 66.0±1.8 | 66.1 |
| PA-BOTH | 63.1±0.7 | 67.2±1.1 | 64.3±0.5 | 72.1±1.8 | 66.3±0.7 | 64.1±1.2 | 69.7±2.1 | 67.5±1.8 | 61.2±1.4 | 67.7±2.3 | 61.2±1.6 | 65.5±0.6 | 65.9 |
| DeSGDA | **76.7±0.8** | **84.6±0.9** | **69.4±0.6** | 85.2±1.5 | **76.2±1.1** | **83.9±1.2** | **69.9±0.6** | **76.3±1.4** | **75.9±1.0** | **73.5±1.3** | **76.3±1.6** | **68.3±0.7** | **76.4** |

*where the (empirical) source and target risks are $\hat{\epsilon}_S(h, \hat{h}) = \frac{1}{N_S} \sum_{n=1}^{N_S} |h(G_n) - \hat{h}(G_n)|$ and $\epsilon_T(h, \hat{h}) = \mathbb{E}_{\mathbb{P}_T(G} \{|h(G) - \hat{h}(G)|\}$, respectively, where $\hat{h} : \mathcal{G} \to \mathcal{Y}$ is the labeling function for graphs and $\omega' = \min_{||g||_{Lip} \leq C_g, ||f||_{Lip} \leq C_f} \{\epsilon_S(h, \hat{h}) + \epsilon_T(h, \hat{h})\}$, $\epsilon_i$ is the Rademacher variable and $p_i$ is the $i^{th}$ row of $\mathbf{P}$, which is defined in Eq. 7.*

The proof is detailed in Appendix C. From Theorem 3, we observe that the bound of DeSGDA is lower than simply aligning the distributions by incorporating the highly reliable pseudo-labels, demonstrating the effectiveness of pseudo labels for spiking graph domain adaptation.

### 4.4 LEARNING FRAMEWORK

Finally, the overall training objective of DeSGDA integrates classification loss $\mathcal{L}_S$, adversarial training loss $\mathcal{L}_{AD}$, and pseudo-label distillation loss $\mathcal{L}_T$, which is formulated as:

$$\mathcal{L} = \mathcal{L}_S + \mathcal{L}_T - \lambda \mathcal{L}_{AD}, \tag{11}$$

where $\lambda$ is a hyper-parameter to balance the adversarial training loss and classification loss. The learning procedure is illustrated in Algorithm D, and the complexity is shown in Appendix E .

## 5 EXPERIMENT

### 5.1 EXPERIMENTAL SETTINGS

**Dataset.** To demonstrate the effectiveness of DeSGDA, we conduct extensive experiments on four widely-used graph classification datasets from TUDataset [1], including PROTEINS (Dobson & Doig, 2003), NCI1 (Wale et al., 2008), FRANKENSTEIN (Orsini et al., 2015), and MUTAGENICITY (Kazius et al., 2005). To better address the variation in domain distributions within each dataset, we divided them into source and target domains based on the edge density, node density, and graph flux (i.e., the ratio of the number of nodes to the number of edges). The specific statistics, distribution visualization, and details introduction of experimental datasets are presented in Appendix F.

**Baselines.** We compare DeSGDA with competitive baselines on the aforementioned datasets, including one graph kernel method: WL subtree (Shervashidze et al., 2011); four general graph neural networks: GCN (Kipf & Welling, 2017), GIN (Xu et al., 2018), CIN (Bodnar et al., 2021) and GMT (Baek et al., 2021); two spiking graph neural networks: SpikeGCN (Zhu et al., 2022) and DRSGNN (Zhao et al., 2024); three recent domain adaptation methods: CDAN (Long et al., 2018), ToAlign (Wei et al., 2021b), and MetaAlign (Wei et al., 2021a); and six graph domain adaptation methods: DEAL (Yin et al., 2022), CoCo (Yin et al., 2023), SGDA (Qiao et al., 2023), DGDA (Cai

---

[1] https://chrsmrrs.github.io/datasets/

Table 2: The graph classification results (in %) on NCI1 under graph flux domain shift (source→target). N0, N1, N2, and N3 denote the sub-datasets partitioned with graph flux. **Bold** results indicate the best performance. OOM means out of memory.

| Methods | N0→N1 | N1→N0 | N0→N2 | N2→N0 | N0→N3 | N3→N0 | N1→N2 | N2→N1 | N1→N3 | N3→N1 | N2→N3 | N3→N2 | Avg. |
|---|---|---|---|---|---|---|---|---|---|---|---|---|---|
| WL subtree | **75.9** | 70.4 | 64.3 | 63.9 | 60.6 | 64.7 | **73.2** | **78.9** | 66.8 | 69.2 | 74.2 | **72.9** | 69.6 |
| GCN | 49.2±1.7 | 55.8±1.5 | 46.8±0.5 | 54.6±2.2 | 43.4±0.6 | 46.7±0.2 | 50.0±1.8 | 57.2±2.2 | 44.2±0.4 | 51.6±0.8 | 62.7±2.1 | 56.8±1.3 | 51.6 |
| GIN | 68.8±2.5 | 70.6±1.0 | 64.2±1.1 | 67.2±2.4 | 62.2±1.8 | 62.5±1.5 | 68.7±2.4 | 72.5±0.6 | 63.3±1.6 | 65.2±0.6 | 62.4±0.3 | 70.9±0.5 | 66.6 |
| GMT | 66.7±0.3 | 58.2±0.5 | 63.9±0.3 | 58.4±0.3 | 63.8±0.4 | 56.7±0.5 | 63.9±0.7 | 66.3±1.0 | 63.8±1.1 | 66.6±0.4 | 63.8±0.2 | 62.6±0.7 | 62.9 |
| CIN | 58.7±2.4 | 54.9±0.2 | 52.0±0.3 | 54.8±0.1 | 56.6±0.2 | 54.9±0.1 | 52.9±1.4 | 52.8±0.5 | 56.5±0.6 | 52.8±2.1 | 58.5±0.8 | 56.6±1.4 | 55.1 |
| SpikeGCN | 58.9±0.9 | 65.2±1.2 | 60.8±1.3 | 62.0±1.5 | 62.3±0.8 | 58.7±1.6 | 64.1±0.6 | 66.7±1.3 | 60.5±1.7 | 63.8±1.4 | 62.2±2.1 | 61.1±1.5 | 62.1 |
| DRSGNN | 58.0±0.6 | 64.3±1.1 | 61.2±0.8 | 62.2±1.0 | 62.9±1.5 | 64.0±1.3 | 60.6±1.6 | 64.0±1.4 | 67.6±2.1 | 62.4±1.9 | 71.3±2.3 | 68.8±2.0 | 63.9 |
| CDAN | 64.0±1.1 | 68.1±0.3 | 60.1±0.5 | 64.0±1.3 | 60.9±0.2 | 57.8±1.0 | 64.3±1.6 | 61.2±0.2 | 66.3±0.7 | 59.0±0.5 | 68.9±0.3 | 63.7±0.6 | 63.2 |
| ToAlign | 52.8±0.5 | 54.8±0.2 | 48.2±1.1 | 54.8±1.5 | 44.0±0.8 | 54.8±2.0 | 48.2±1.7 | 52.8±0.6 | 44.0±0.2 | 52.8±0.3 | 44.0±1.0 | 48.2±1.2 | 50.0 |
| MetaAlign | 63.1±0.3 | 63.8±1.3 | 58.9±2.4 | 58.5±0.4 | 59.1±2.1 | 59.2±1.6 | 70.1±0.8 | 63.3±1.4 | 66.5±2.7 | 60.9±1.1 | 71.4±0.2 | 67.5±0.8 | 63.5 |
| DEAL | 70.7±0.9 | **72.3±0.2** | 69.9±0.8 | 68.9±0.7 | 64.1±0.6 | 65.6±0.9 | 71.9±0.4 | 69.9±1.7 | 70.6±0.4 | 66.5±0.3 | 71.6±0.7 | 69.9±0.5 | 69.3 |
| CoCo | 64.0±1.3 | 63.9±0.6 | 65.8±1.8 | 59.9±1.7 | 62.2±2.1 | 60.6±1.6 | 65.0±2.1 | 64.8±1.4 | 60.0±0.8 | 61.3±0.5 | 68.5±0.4 | 67.1±0.6 | 63.6 |
| SGDA | OOM | OOM | OOM | OOM | OOM | OOM | OOM | OOM | OOM | OOM | OOM | OOM | OOM |
| DGDA | OOM | OOM | OOM | OOM | OOM | OOM | OOM | OOM | OOM | OOM | OOM | OOM | OOM |
| A2GNN | 58.9±0.9 | 60.1±0.7 | 59.8±1.2 | 59.4±1.0 | 62.3±1.5 | 60.9±1.6 | 61.6±1.3 | 59.9±1.9 | 64.9±1.6 | 62.9±2.1 | 65.4±1.5 | 63.3±2.3 | 61.7 |
| PA-BOTH | 61.1±0.5 | 60.9±0.4 | 61.6±0.6 | 61.2±0.8 | 60.8±0.6 | 61.5±0.5 | 62.2±1.0 | 61.9±0.7 | 61.8±1.1 | 61.1±0.9 | 60.9±0.8 | 61.3±1.2 | 61.6 |
| DeSGDA | 68.5±1.2 | 71.4±1.3 | **70.1±0.7** | **69.0±1.1** | **68.9±1.0** | **66.3±1.4** | 69.6±1.3 | 70.2±1.7 | **71.1±1.6** | **69.3±1.8** | **74.4±1.6** | 70.0±1.9 | **70.1** |

et al., 2024), A2GNN (Liu et al., 2024a) and PA-BOTH (Liu et al., 2024b). More details about the compared baselines can be found in Appendix G.

**Implementation Details.** DeSGDA and all baseline models are implemented using PyTorch[2] and PyTorch Geometric[3]. For DeSGDA, we deploy the GIN (Xu et al., 2018) as the backbone of the degree-aware personalized spiking graph encoder, incorporating a mean-pooling layer for the readout function. We conduct experiments for DeSGDA and all baselines on NVIDIA A100 GPUs for a fair comparison, where the learning rate of Adam optimizer set to $10^{-4}$, hidden embedding dimension 256, weight decay $10^{-12}$, and GNN layers 4. Additionally, DeSGDA and all baseline models are trained using all labeled source samples and evaluated on unlabeled target samples (Wu et al., 2020). The performances of all models are measured and averaged on all samples for five runs.

## 5.2 PERFORMANCE COMPARISION

We present the results of the proposed DeSGDA with all baseline models under the setting of graph domain adaptation on different datasets in Table 1, 2, 19. From these tables, we observe that: (1) The performance of graph domain adaptation methods surpasses that of graph and spike-based graph methods. We attribute this improvement to the fact that domain distribution shifts degrade the performance of traditional graph methods. (2) The graph domain adaptation methods (DEAL and CoCo) outperform the spike-based graph methods (SpikeGCN and DRSGNN), underscoring the necessity of the

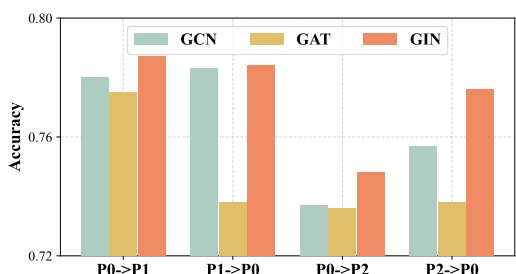

Figure 2: The performance with different GNN architectures on PROTEINS.

research in spiking graph domain adaptation. (3) The WL subtree method outperforms SGDA, DGDA, A2GNN, and PA-BOTH but falls short compared to DEAL and CoCO. We attribute this to the relatively limited research specifically addressing the graph domain adaptation problem (e.g., DEAL and CoCo). To bridge this gap, we adapted node classification methods for graph classification tasks (e.g., SGDA, DGDA, A2GNN, and PA-BOTH). While the WL subtree method demonstrates superior performance over these adapted node classification-based methods, it remains less effective than dedicated graph domain adaptation methods tailored for graph classification tasks. (4) Our DeSGDA outperforms all baselines for most cases, which demonstrates its superiority over other methods. The remarkable performance of DeSGDA lies in two main reasons: (i) The degree-aware personalized spiking representations can capture more expressive information for graph classification by dynamically adjusting the thresholds of nodes in SNNs. (ii) The adversarial distribution alignment effectively addresses domain discrepancies by adversarially training the encoder and domain

---

[2]https://pytorch.org/

[3]https://www.pyg.org/

Table 3: The results of ablation studies on PROTEINS (source → target). Bold results indicate the best performance. **Bold** results indicate the best performance.

| Methods | P0→P1 | P1→P0 | P0→P2 | P2→P0 | P0→P3 | P3→P0 | P1→P2 | P2→P1 | P1→P3 | P3→P1 | P2→P3 | P3→P2 | Avg. |
|---|---|---|---|---|---|---|---|---|---|---|---|---|---|
| DeSGDA w/o CA | 72.3 | 72.6 | 71.9 | 75.1 | 74.1 | 71.3 | 71.5 | 70.4 | 71.6 | 70.5 | 78.3 | 71.5 | 72.6 |
| DeSGDA w/o PL | 72.1 | 68.7 | 67.8 | 69.4 | 63.7 | 55.7 | 68.3 | 69.5 | 70.2 | 69.6 | 76.7 | 66.9 | 68.3 |
| DeSGDA w/o CF | 56.7 | 54.2 | 60.7 | 62.1 | 73.4 | 68.3 | 55.3 | 69.5 | 76.7 | 65.5 | 73.7 | 63.3 | 65.7 |
| DeSGDA w/o TL | 61.7 | 43.8 | 56.1 | 71.2 | 49.5 | 69.0 | 42.4 | 68.3 | 73.7 | 64.0 | 70.5 | 43.5 | 59.5 |
| DeSGDA w/ PT | 71.5 | 68.3 | 66.1 | 72.4 | 66.7 | 70.1 | 67.9 | 70.8 | 66.3 | 71.8 | 74.1 | 72.9 | 70.0 |
| DeSGDA w/ CL | 74.9 | 73.3 | 73.7 | 75.1 | 77.0 | 71.3 | 73.9 | 70.0 | 78.1 | **77.6** | 78.8 | 74.9 | 75.1 |
| DeSGDA | **78.7** | **78.4** | **74.8** | **77.6** | **79.5** | **76.7** | **74.9** | **71.2** | **79.5** | 72.8 | **81.0** | **75.1** | **76.4** |

discriminator to align feature spaces. Moreover, the pseudo-label distillation aids in updating the source degree thresholds in the target domain, thereby ensuring optimal performance. More results evaluated on other datasets can be found in Appendix H.1.

Additionally, we conduct experiments to explore the flexibility of the proposed DeSGDA. Specifically, we replace the backbone of the degree-aware personalized spiking graph encoder (GIN) with various GNNs methods (i.e., GCN and GAT), and the results are shown in Figure 2. From the results, we observe that GIN consistently outperforms other GNNs architectures in most cases, demonstrating its powerful representation capability. This phenomenon also justifies our choice of using GIN to enhance the performance of the proposed DeSGDA. More results are reported in Appendix H.1.

## 5.3 ENERGY EFFICIENCY ANALYSIS

To assess the energy efficiency of DeSGDA, we use the metric from (Zhu et al., 2022) and quantify the energy consumption for graph classification in the inference stage. Specifically, the graph domain adaption methods are evaluated on GPUs (NVIDIA A100), and the spiking-based methods are evaluated on neuromorphic chips (ROLLS (Indiveri et al., 2015)) following (Zhu et al., 2022). The results are shown in Figure 3, from the results, we find that compared with traditional graph domain adaptation methods, the spike-based methods (DeSGDA and DRSGNN) have significantly lower energy consumption,

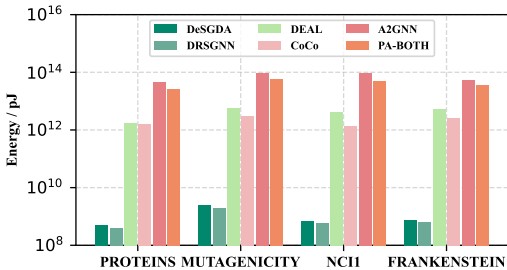

Figure 3: Energy consumption of DeSGDA and baselines on different datasets.

demonstrating the superior energy efficiency of SGNs. Moreover, although the energy consumption of DeSGDA is slightly higher than DRSGNN due to additional computations required for domain adaptation, the performance improvement justifies the deployment of DeSGDA in low-power devices. Additionally, we present a comparison of training time and memory usage between DeSGDA and other graph domain adaptation methods. The results are detailed in Table 10 and 11.

## 5.4 ABLATION STUDY

We conduct ablation studies to examine the contributions of each component in the proposed DeSGDA: (1) DeSGDA w/o CA: It removes the adversarial distribution alignment module; (2) DeSGDA w/o PL: It removes the pseudo-label distilling module; (3) DeSGDA w/o CF: It removes the classification loss $\mathcal{L}_S$; (4) DeSGDA w/o TL: It utilizes the global thresholds on all nodes; (5) DeSGDA w/ PT: It deploys the adaptive perturbations (Yin et al., 2022) on source data for alignment; (6) DeSGDA w/ CL: It replaces the adversarial learning with the cross-domain contrastive learning (Yin et al., 2023).

Experimental results are shown in Table 3. From the table, we find that: (1) DeSGDA outperforms DeSGDA w/o CA, DeSGDA w/o PL, and DeSGDA w/o CF, demonstrating that the adversarial distribution alignment module can effectively reduce domain discrepancies, ensuring well-aligned feature spaces between source and target domains. Additionally, the pseudo-label distillation module can address the variance in thresholds across domains, while the classification loss $\mathcal{L}_S$ enables DeSGDA to effectively learn from labeled source data and generalize to the target domain. (2) DeSGDA w/o TL shows lower performance compared to DeSGDA, showing that the degree-aware thresholds, which are iteratively updated during model training, can resolve the issue of the inflexible architecture in SGNs. By using these thresholds, DeSGDA can effectively learn meaningful representations for nodes

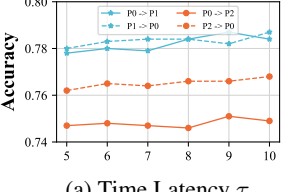 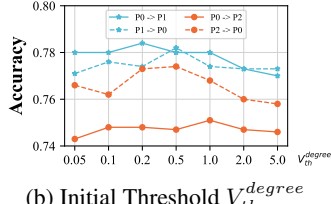 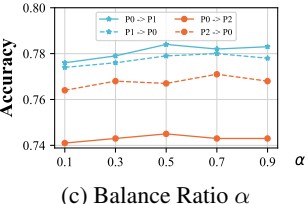

| (a) Time Latency $\tau$ | (b) Initial Threshold $V_{th}^{degree}$ | (c) Balance Ratio $\alpha$ |
|---|---|---|

Figure 4: Hyperparameter sensitivity analysis of time latency $\tau$, initial threshold $V_{th}^{degree}$ in SNNs, and balance ratio $\alpha$ on PROTEINS.

with various degrees. (3) DeSGDA outperforms DeSGDA w/ PT and DeSGDA w/ CL. We attribute that the adaptive perturbations method (DeSGDA w/ PT) can not effectively help DeSGDA overcome the domain discrepancy. Additionally, the cross-domain contrastive learning method (DeSGDA w/ CL) is less effective at aligning the source and target distribution compared to adversarial learning. Additionally, we provide the ablation studies to examine the effect of directly replacing the SGNs with commonly used Graph Neural Networks (GNNs) for generating representations for DeSGDA, and the results are shown in Table 12, 13. More details about ablation results on other datasets are reported in Appendix H.3.

## 5.5 SENSITIVITY ANALYSIS

We study the sensitivity analysis of DeSGDA with respect to the impact of its hyperparameters: time latency $\tau$, degree threshold value $V_{th}^{degree}$ in SNNs, and balance ratio $\alpha$, which plays a crucial role in the performance of DeSGDA. In particular, $\tau$ controls the number of SNNs propagation steps; $V_{th}^{degree}$ determines when a neuron fires; $\alpha$ governs the changing ratio of degree-aware thresholds.

Figure 4 illustrates how $\tau$, $V_{th}^{degree}$, and $\alpha$ affects the performance of DeSGDA on the PROTEINS dataset. More results on other datasets are shown in Appendix H.4. We vary $\tau$ within the range of $\{5, 6, 7, 8, 9, 10\}$, $V_{th}^{degree}$ in $\{0.05, 0.1, 0.2, 0.5, 1.0, 2.0, 5.0\}$, and $\alpha$ in $\{0.1, 0.3, 0.5, 0.7, 0.9\}$. From the results, we observe that: (1) The performance of DeSGDA in Figure 4a generally exhibits an increasing trend at the beginning and then stabilizes when $\tau$ is greater than 8. We attribute this to smaller values of $\tau$ potentially losing important information for representation, while larger values significantly increase model complexity. To balance performance and complexity, we set $\tau$ to 8 as default. (2) Figure 4b indicates an initial increase followed by a decreasing trend in performance as $V_{th}^{degree}$ increases. This trend occurs because a lower threshold may trigger more spikes for high-degree nodes, leading to a drastic change in the threshold, which can degrade performance. Conversely, a higher threshold for low-degree nodes could result in fewer spikes, affecting the model's ability to process information effectively. Thus, we set $V_{th}^{degree}$ to 0.5 as default. (3) From Figure 4c, we find that the performance of DeSGDA initially increases and then decreases as $\alpha$ increases. The potential reason is that the smaller $\alpha$ may delay the updating of the threshold, leading to performance degradation. Contrarily, a larger $\alpha$ tends to introduce more spikes that change dynamically at each step, resulting in instability in the model's performance. Therefore, we set $\alpha$ to 0.5 as default.

## 6 CONCLUSION

In this paper, we first propose the problem of spiking graph domain adaptation and introduce a novel framework DeSGDA for graph classification. This framework enhances the adaptability and performance of SGNs through three key aspects: node degree-aware personalized spiking representation, adversarial feature distribution alignment, and pseudo-label distillation. Our approach enables more expressive information capture through degree-dependent spiking thresholds, aligns feature distributions via adversarial training, and utilizes pseudo-labels to leverage unlabeled data effectively. The extensive experimental validation across benchmark datasets has demonstrated that DeSGDA not only surpasses existing methods in accuracy but also maintains efficient energy consumption, making it a promising solution for advancing the domain adaptation capabilities of spiking graph networks. In the future, we will apply SGNs in the scenarios of source-free domain adaptation and domain generalization.

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

## A  PROOF OF HYPOTHESIS 1

Assuming that the node feature $h_i$ follows a normal distribution $\mathcal{N}(\mu, \sigma^2)$, then for each node in the graph, we follow the message-passing mechanism and have the information aggregation as:

$$h_i = h_i + \sum_{j \in N(i)} w_{ij} h_j. \tag{12}$$

Therefore, we have the expectation:

$$\mathbb{E}(h_i) = \mathbb{E}(h_i) + \sum_{j \in N(i)} w_{ij} \mathbb{E}(h_j), \tag{13}$$

Since $\mathbb{E}(h_j) \sim \mathcal{N}(\mu, \sigma^2)$, we have:

$$\mathbb{E}(h_i) \sim \mathcal{N}\left( (1 + \sum_{j \in N(i)} w_{ij})\mu, (1 + \sum_{j \in N(i)} w_{ij})\sigma^2 \right). \tag{14}$$

From the results, we observe that node $i$ follows a normal distribution with a mean of $(1 + \sum_{j \in N(i)} w_{ij})\mu$, determined by the aggregated weights of its neighboring nodes. To provide a more intuitive understanding, we visualize the aggregated neighbor weights of GCN Kipf & Welling (2017) and GIN Xu et al. (2018) in Figure 5. The results show that as the degree increases, the aggregated weights also increase progressively. Consequently, high-degree nodes tend to follow a normal distribution with a higher mean and variance. In other words, nodes with higher degrees accumulate greater signals, making them more likely to trigger spiking. Based on this, we propose assigning higher thresholds to high-degree nodes and lower thresholds to low-degree nodes.

Another observation is that methods that normalize neighbor weights to 1 (e.g., GAT Veličković et al. (2017), GraphSAGE Hamilton et al. (2017)) still result in aggregated features following the same normal distribution. This normalization diminishes the ability to distinguish between nodes with varying degrees, ultimately degrading performance. This explains why, when using GAT as the backbone of DeSGDA, the performance is the weakest.

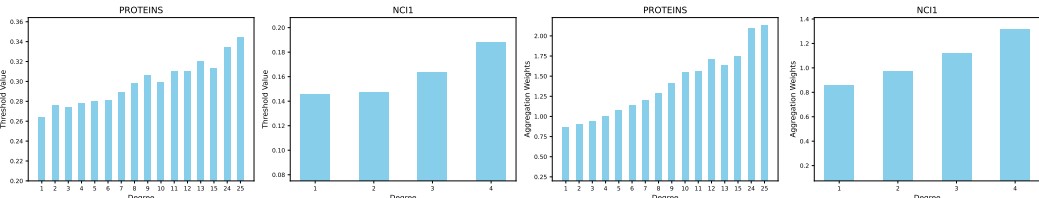

Figure 5: Visualization of degree-aware thresholds and aggregation weights on PROTEINS and NCI1.

## B  PROOF OF THEOREM 2

**Theorem 2** *Assuming that the learned discriminator is $C_g$-Lipschitz continuous as described in Theorem 1, the graph feature extractor $f$ (also referred to as GNN) is $C_f$-Lipschitz that $||f||_{Lip} = \max_{G_1, G_2} \frac{||f(G_1) - f(G_2)||_2}{\eta(G_1, G_2)} = C_f$ for some graph distance measure $\eta$ and the loss function bounded by $C > 0$. Let $\mathcal{H} := \{h : \mathcal{G} \to \mathcal{Y}\}$ be the set of bounded real-valued functions with the pseudo-dimension $Pdim(\mathcal{H}) = d$ that $h = g \circ f \in \mathcal{H}$, and provided the spike training data set $S_n = \{(\mathbf{X}_i, y_i) \in \mathcal{X} \times \mathcal{Y}\}_{i \in [n]}$ drawn from $\mathcal{D}^s$, with probability at least $1 - \delta$ the following inequality:*

$$
\begin{aligned}
\epsilon_T(h, \hat{h}_T(\mathbf{X})) \leq & \hat{\epsilon}_S(h, \hat{h}_S(S_n)) + 2\mathbb{E}\left[ \sup \frac{1}{N_S} \sum_{i=1}^{N_S} \epsilon_i h(\mathbf{X}_i, y_i, p_i) \right] + C\sqrt{\frac{ln(2/\delta)}{N_S}} \\
& + \min\left( |\epsilon_S(h, \hat{h}_S(\mathbf{X})) - \epsilon_S(h, \hat{h}_T(\mathbf{X}))|, |\epsilon_T(h, \hat{h}_S(\mathbf{X})) - \epsilon_T(h, \hat{h}_T(\mathbf{X}))| \right) \\
& + 2C_f C_g W_1\left( \mathbb{P}_S(G), \mathbb{P}_T(G) \right),
\end{aligned}
\tag{15}
$$

*where the (empirical) source and target risks are $\hat{\epsilon}_S(h, \hat{h}) = \frac{1}{N_S} \sum_{n=1}^{N_S} |h(G_n) - \hat{h}(G_n)|$ and $\epsilon_T(h, \hat{h}) = \mathbb{E}_{\mathbb{P}_T(G}\{|h(G) - \hat{h}(G)|\}$, respectively, where $\hat{h} : \mathcal{G} \to \mathcal{Y}$ is the labeling function for graphs and $\omega = \min_{||g||_{Lip} \leq C_g, ||f||_{Lip} \leq C_f} \{\epsilon_S(h, \hat{h}) + \epsilon_T(h, \hat{h})\}$, $\epsilon_i$ is the Rademacher variable and $p_i$ is the $i^{th}$ row of $\mathbf{P}$, which is the probability matrix with:*

$$\mathbf{P}_{kt} = \begin{cases} \exp\left(\frac{u_k(t) - V_{th}}{\sigma(u_k(t) - u_{reset})}\right), & if \quad u_\theta \leq u(t) \leq V_{th}, \\ 0, & if \quad u_{reset} \leq u_k(t) \leq u_\theta. \end{cases} \tag{16}$$

*Proof.* Before showing the designated lemma, we first introduce the following inequality to be used that:

$$\begin{aligned} |\epsilon_S(h, \hat{h}_S) - \epsilon_T(h, \hat{h}_T)| &= |\epsilon_S(h, \hat{h}_S) - \epsilon_S(h, \hat{h}_T) + \epsilon_S(h, \hat{h}_T) - \epsilon_T(h, \hat{h}_T)| \\ &\leq |\epsilon_S(h, \hat{h}_S) - \epsilon_S(h, \hat{h}_T)| + |\epsilon_S(h, \hat{h}_T) - \epsilon_T(h, \hat{h}_T)| \\ &\overset{(a)}{\leq} |\epsilon_S(h, \hat{h}_S) - \epsilon_S(h, \hat{h}_T)| + 2C_f C_g W_1\left(\mathbb{P}_S(G), \mathbb{P}_T(G)\right), \end{aligned} \tag{17}$$

where $(a)$ results from (Shen et al., 2018) Theorem 1 with the assumption $\max(||h||_{Lip}, \max_{G_1, G_2} \frac{|\hat{h}_D(G_1) - \hat{h}_D(G_2)|}{\eta(G_1, G_2)}) \leq C_f C_g$, $D \in \{S, T\}$. Similarly, we obtain:

$$|\epsilon_S(h, \hat{h}_S) - \epsilon_T(h, \hat{h}_T)| \leq |\epsilon_T(h, \hat{h}_S) - \epsilon_T(h, \hat{h}_T)| + 2C_f C_g W_1(\mathbb{P}_S(G), \mathbb{P}_T(G)). \tag{18}$$

We therefore combine them into:

$$\begin{aligned} |\epsilon_S(h, \hat{h}_S) - \epsilon_T(h, \hat{h}_T)| \leq &2C_f C_g W_1(\mathbb{P}_S(G), \mathbb{P}_T(G)) \\ &+ \min\left(|\epsilon_S(h, \hat{h}_S) - \epsilon_S(h, \hat{h}_T)|, |\epsilon_T(h, \hat{h}_S) - \epsilon_T(h, \hat{h}_T)|\right), \end{aligned} \tag{19}$$

i.e. the following holds to bound the target risk $\epsilon_T(h, \hat{h}_T)$:

$$\begin{aligned} \epsilon_T(h, \hat{h}_T) \leq &\epsilon_S(h, \hat{h}_S) + 2C_f C_g W_1(\mathbb{P}_S(G), \mathbb{P}_T(G)) \\ &+ \min\left(|\epsilon_S(h, \hat{h}_S) - \epsilon_S(h, \hat{h}_T)|, |\epsilon_T(h, \hat{h}_S) - \epsilon_T(h, \hat{h}_T)|\right). \end{aligned} \tag{20}$$

We next link the bound with the empirical risk and labeled sample size by showing, with probability at least $1 - \delta$ that:

$$\begin{aligned} \epsilon_T(h, \hat{h}_T) \leq &\epsilon_S(h, \hat{h}_S) + 2C_f C_g W_1(\mathbb{P}_S(G), \mathbb{P}_T(G)) \\ &+ \min\left(|\epsilon_S(h, \hat{h}_S) - \epsilon_S(h, \hat{h}_T)|, |\epsilon_T(h, \hat{h}_S) - \epsilon_T(h, \hat{h}_T)|\right) \end{aligned} \tag{21}$$

The $\hat{h}$ above is the abbreviation of $\hat{h}(x)$, which means the input is the continuous feature. Provided the spike training data set $S_n = \{(\mathbf{X}_i, y_i) \in \mathcal{X} \times \mathcal{Y}\}_{i \in [n]}$ drawn from $\mathcal{D}$, and motivated by (Yin et al., 2024), we have:

$$\lim_{\tau \to \infty} P\left(\hat{h}(S_n)_{\tau, i} > \hat{h}(\mathbf{X}_{\tau, i}) + \epsilon\right) \leq e^{-\epsilon^2 / 2(\sigma + \hat{w}_i \epsilon / 3)}, \tag{22}$$

where $\hat{w}_i = max\{w_{i1}, \cdots, w_{id}\}$ and $h(\mathbf{x}_{ij}) = \sum_{j=1}^d w_{ij} \mathbf{x}_{ij}$. From Equation 2, we observe that as $\tau \to \infty$, the difference between spike and real-valued features will be with the probability of $p = e^{-\epsilon^2 / 2(\sigma + \hat{w}_i \epsilon / 3)}$ to exceed the upper and lower bounds.

Furthermore, motivated by the techniques given by (Bartlett & Mendelson, 2002), we have:

$$\epsilon_S(h, \hat{h}_S(S_n)) \leq \hat{\epsilon}_S(h, \hat{h}_S(S_n)) + \underbrace{\sup[\epsilon_S(h, \hat{h}_S(S_n)) - \hat{\epsilon}_S(h, \hat{h}_S(S_n))]}_{R(S_n, \mathbf{P})}, \tag{23}$$

where $\mathbf{P}$ is the probability matrix with:

$$\mathbf{P}_{kt} = \begin{cases} \exp\left(\frac{u_k(t) - V_{th}}{\sigma(u_k(t) - u_{reset})}\right), & if \quad u_\theta \leq u(t) \leq V_{th}, \\ 0, & if \quad u_{reset} \leq u_k(t) \leq u_\theta, \end{cases} \tag{24}$$

where $k$ indicates the $k - th$ spiking neuron and the membrane threshold $u_{theta}$ is relative to the excitation probability threshold $p_\theta \in (0, 1]$. Let $p_k$ is the $k - th$ row vector of $\mathbf{P}$. Thus, we have the probability at least $1 - e^{-\epsilon^2/2(\sigma+\hat{w}_i\epsilon/3)}$ to hold:

$$\epsilon_S(h, \hat{h}_S(\mathbf{X}_n)) \leq \hat{\epsilon}_S(h, \hat{h}_S(S_n)) + \underbrace{\sup[\epsilon_S(h, \hat{h}_S(S_n)) - \hat{\epsilon}_S(h, \hat{h}_S(S_n))]}_{R(S_n, \mathbf{P})}, \tag{25}$$

Let $S'_n$ denote the sample set that the $i^{th}$ sample $(\mathbf{X}_i, y_i)$ is replaced by $(\mathbf{X}'_i, y'_i)$, and correspondingly $\mathbf{P}'$ is the possibility matrix that the $i^{th}$ row vector $p_i$ is replaced by $p'_i$, for $i \in [n]$. For the loss function bounded by $C > 0$, we have:

$$\begin{cases} |R(S_n, \mathbf{P}) - R(S'_n, \mathbf{P})| \leq C/n, \\ |R(S_n, \mathbf{P}) - R(S_n, \mathbf{P}')| \leq C/n. \end{cases} \tag{26}$$

From McDiarmid's inequality (McDiarmid et al., 1989), with probability at least $1 - \delta$, we have:

$$R(S_n, \mathbf{P}) \leq \mathbb{E}_{S_n \in \mathcal{D}, \mathbf{P}}[R(S_n, \mathbf{P})] + C\sqrt{\frac{ln(2/\delta)}{N_S}}. \tag{27}$$

It is observed that:

$$R(S_n, \mathbf{P}) = \sup \mathbb{E}_{\tilde{S}_n \in \mathcal{D}, \tilde{\mathbf{P}}}[\hat{\epsilon}(\hat{h}(S_n); \tilde{S}_n, \tilde{\mathbf{P}}) - \tilde{\mathbf{P}}[\hat{\epsilon}(\hat{h}(S_n); S_n, \mathbf{P})], \tag{28}$$

where $\tilde{S}_n$ is another collection drawn from $\mathcal{D}$ as well as $\tilde{\mathbf{P}}$. Thus, we have

$$\begin{aligned} \mathbb{E}_{S_n \in \mathcal{D}, \mathbf{P}}[R(S_n, \mathbf{P})] &\leq \mathbb{E}\left[\sup\left[\hat{\epsilon}(\hat{h}(S_n); \tilde{S}_n, \tilde{\mathbf{P}}) - \tilde{\mathbf{P}}[\hat{\epsilon}(\hat{h}(S_n); S_n, \mathbf{P})]\right]\right] \\ &= \mathbb{E}\left[\sup \frac{1}{n} \sum_{i=1}^{n}[\hat{h}(\tilde{\mathbf{X}}_i, \tilde{y}_i, \tilde{p}_i) - \hat{h}(\mathbf{X}_i, y_i, p_i)]\right] \\ &\leq 2\mathbb{E}\left[\sup \frac{1}{n} \sum_{i=1}^{n} \epsilon_i \hat{h}(\mathbf{X}_i, y_i, p_i)\right], \end{aligned} \tag{29}$$

where $\epsilon_i$ is the Rademacher variable. Combining Eq. 26 27 29, we have:

$$\epsilon_S(h, \hat{h}_S(\mathbf{X}_n)) \leq \hat{\epsilon}_S(h, \hat{h}_S(S_n)) + 2\mathbb{E}\left[\sup \frac{1}{N_S} \sum_{i=1}^{N_S} \epsilon_i h(\mathbf{X}_i, y_i, p_i)\right] + C\sqrt{\frac{ln(2/\delta)}{N_S}}. \tag{30}$$

Finally, we have:

$$\begin{aligned} \epsilon_T(h, \hat{h}_T(\mathbf{X})) &\leq \epsilon_S(h, \hat{h}_S(\mathbf{X})) + 2C_f C_g W_1\left(\mathbb{P}_S(G), \mathbb{P}_T(G)\right) \\ &\quad + \min\left(|\epsilon_S(h, \hat{h}_S(\mathbf{X})) - \epsilon_S(h, \hat{h}_T(\mathbf{X}))|, |\epsilon_T(h, \hat{h}_S(\mathbf{X})) - \epsilon_T(h, \hat{h}_T(\mathbf{X}))|\right) \\ &\leq \hat{\epsilon}_S(h, \hat{h}_S(S_n)) + 2\mathbb{E}\left[\sup \frac{1}{N_S} \sum_{i=1}^{N_S} \epsilon_i h(\mathbf{X}_i, y_i, p_i)\right] + C\sqrt{\frac{ln(2/\delta)}{N_S}} \\ &\quad + \min\left(|\epsilon_S(h, \hat{h}_S(\mathbf{X})) - \epsilon_S(h, \hat{h}_T(\mathbf{X}))|, |\epsilon_T(h, \hat{h}_S(\mathbf{X})) - \epsilon_T(h, \hat{h}_T(\mathbf{X}))|\right) \\ &\quad + 2C_f C_g W_1\left(\mathbb{P}_S(G), \mathbb{P}_T(G)\right). \end{aligned} \tag{31}$$

## C  PROOF OF THEOREM 3

**Theorem 3** *Under the assumption of Theorem 1, we further assume that there exists a small amount of i.i.d. samples with pseudo labels $\{(G_n, Y_n)\}_{n=1}^{N'_T}$ from the target distribution $\mathbb{P}_T(G, Y)$ ($N'_T \ll N_S$) and bring in the conditional shift assumption that domains have different labeling function $\hat{h}_S \neq \hat{h}_T$ and $\max_{G_1, G_2} \frac{|\hat{h}_D(G_1) - \hat{h}_D(G_2)|}{\eta(G_1, G_2)} = C_h \leq C_f C_g (D \in \{S, T\})$ for some constant $C_h$ and distance*

*measure $\eta$, and the loss function bounded by $C > 0$. Let $\mathcal{H} := \{h : \mathcal{G} \to \mathcal{Y}\}$ be the set of bounded real-valued functions with the pseudo-dimension $Pdim(\mathcal{H}) = d$, and provided the spike training data set $S_n = \{(\mathbf{X}_i, y_i) \in \mathcal{X} \times \mathcal{Y}\}_{i \in [n]}$ drawn from $\mathcal{D}^s$, with probability at least $1 - \delta$ the following inequality holds:*

$$\epsilon_T(h, \hat{h}_T(\mathbf{X})) \leq \frac{N'_T}{N_S + N'_T} \hat{\epsilon}_T(h, \hat{h}_T(S)) + \frac{N_S}{N_S + N'_T} \left( \hat{\epsilon}_S(h, \hat{h}_S(S)) + 2C_f C_g W_1 \left( \mathbb{P}_S(G), \mathbb{P}_T(G) \right) \right.$$

$$+ 2\mathbb{E}\left[ \sup \frac{1}{N_S} \sum_{i=1}^{N_S} \epsilon_i h(\mathbf{X}_i, y_i, p_i) \right] + C\sqrt{\frac{ln(2/\delta)}{N_S}}$$

$$\left. + \min\left( |\epsilon_S(h, \hat{h}_S(\mathbf{X}))) - \epsilon_S(h, \hat{h}_T(\mathbf{X})))|, |\epsilon_T(h, \hat{h}_S(\mathbf{X}))) - \epsilon_T(h, \hat{h}_T(\mathbf{X})))| \right) \right)$$

$$\leq \hat{\epsilon}_S(h, \hat{h}_S(S)) + 2\mathbb{E}\left[ \sup \frac{1}{N_S} \sum_{i=1}^{N_S} \epsilon_i h(\mathbf{X}_i, y_i, p_i) \right] + C\sqrt{\frac{ln(2/\delta)}{N_S}}$$

$$+ 2C_f C_g W_1 \left( \mathbb{P}_S(G), \mathbb{P}_T(G) \right) + \omega'.$$

$$(32)$$

*where the (empirical) source and target risks are $\hat{\epsilon}_S(h, \hat{h}) = \frac{1}{N_S} \sum_{n=1}^{N_S} |h(G_n) - \hat{h}(G_n)|$ and $\epsilon_T(h, \hat{h}) = \mathbb{E}_{\mathbb{P}_T(G}\{|h(G) - \hat{h}(G)|\}$, respectively, where $\hat{h} : \mathcal{G} \to \mathcal{Y}$ is the labeling function for graphs and $\omega' = \min_{||g||_{Lip} \leq C_g, ||f||_{Lip} \leq C_f} \{\epsilon_S(h, \hat{h}) + \epsilon_T(h, \hat{h})\}$, $\epsilon_i$ is the Rademacher variable and $p_i$ is the $i^{th}$ row of $\mathbf{P}$, which is the probability matrix with:*

$$\mathbf{P}_{kt} = \begin{cases} \exp\left( \frac{u_k(t) - V_{th}}{\sigma(u_k(t) - u_{reset})} \right), & if \quad u_\theta \leq u(t) \leq V_{th}, \\ 0, & if \quad u_{reset} \leq u_k(t) \leq u_\theta. \end{cases} \quad (33)$$

*Proof.* As proved in Theorem 2, we have:

$$\epsilon_T(h, \hat{h}_T(\mathbf{X})) \leq \hat{\epsilon}_S(h, \hat{h}_S(S_n)) + 2\mathbb{E}\left[ \sup \frac{1}{N_S} \sum_{i=1}^{N_S} \epsilon_i h(\mathbf{X}_i, y_i, p_i) \right] + C\sqrt{\frac{ln(2/\delta)}{N_S}}$$

$$+ \min\left( |\epsilon_S(h, \hat{h}_S(\mathbf{X})) - \epsilon_S(h, \hat{h}_T(\mathbf{X}))|, |\epsilon_T(h, \hat{h}_S(\mathbf{X})) - \epsilon_T(h, \hat{h}_T(\mathbf{X}))| \right)$$

$$+ 2C_f C_g W_1 \left( \mathbb{P}_S(G), \mathbb{P}_T(G) \right).$$

$$(34)$$

Similar with Eq. 30, there exists:

$$\epsilon_T(h, \hat{h}_T(\mathbf{X}_n)) \leq \hat{\epsilon}_T(h, \hat{h}_T(S_n)) + 2\mathbb{E}\left[ \sup \frac{1}{N'_T} \sum_{i=1}^{N'_T} \epsilon_i h(\mathbf{X}_i, y_i, p_i) \right] + C\sqrt{\frac{ln(2/\delta)}{N'_T}}. \quad (35)$$

Combining Eq. 34 and 35, we have:

$$\epsilon_T(h, \hat{h}_T(\mathbf{X})) \overset{(a)}{\leq} \frac{N'_T}{N_S + N'_T} \left( \hat{\epsilon}_T(h, \hat{h}_T(S)) + 2\mathbb{E}\left[ \sup \frac{1}{N'_T} \sum_{i=1}^{N'_T} \epsilon_i h(\mathbf{X}_i, y_i, p_i) \right] + C\sqrt{\frac{ln(2/\delta)}{N'_T}} \right)$$

$$+ \frac{N_S}{N_S + N'_T} \left( \hat{\epsilon}_S(h, \hat{h}_S(S)) + 2\mathbb{E}\left[ \sup \frac{1}{N_S} \sum_{i=1}^{N_S} \epsilon_i h(\mathbf{X}_i, y_i, p_i) \right] + C\sqrt{\frac{ln(2/\delta)}{N_S}} \right)$$

$$+ \frac{N_S}{N_S + N'_T} \left( 2C_f C_g W_1 \left( \mathbb{P}_S(G), \mathbb{P}_T(G) \right) \right.$$

$$+ \min \left( |\epsilon_S(h, \hat{h}_S(\mathbf{X})) - \epsilon_S(h, \hat{h}_T(\mathbf{X}))|, |\epsilon_T(h, \hat{h}_S(\mathbf{X})) - \epsilon_T(h, \hat{h}_T(\mathbf{X}))| \right) \Bigg)$$

$$\leq \frac{N'_T}{N_S + N'_T} \hat{\epsilon}_T(h, \hat{h}_T(S)) + \frac{N_S}{N_S + N'_T} \hat{\epsilon}_S(h, \hat{h}_S(S))$$

$$+ \frac{N_S}{N_S + N'_T} \Bigg( 2 C_f C_g W_1 \left( \mathbb{P}_S(G), \mathbb{P}_T(G) \right)$$

$$+ \min \left( |\epsilon_S(h, \hat{h}_S(\mathbf{X}) - \epsilon_S(h, \hat{h}_T((\mathbf{X}))|, |\epsilon_T(h, \hat{h}_S((\mathbf{X})) - \epsilon_T(h, \hat{h}_T((\mathbf{X}))| \right) \Bigg)$$

$$+ \frac{N'_T}{N_S + N'_T} \left( 2\mathbb{E} \left[ \sup \frac{1}{N'_T} \sum_{i=1}^{N'_T} \epsilon_i h(\mathbf{X}_i, y_i, p_i) \right] + C\sqrt{\frac{ln(2/\delta)}{N'_T}} \right)$$

$$+ \frac{N_S}{N_S + N'_T} \left( 2\mathbb{E} \left[ \sup \frac{1}{N_S} \sum_{i=1}^{N_S} \epsilon_i h(\mathbf{X}_i, y_i, p_i) \right] + C\sqrt{\frac{ln(2/\delta)}{N_S}} \right)$$

$$\overset{(b)}{\doteq} \frac{N'_T}{N_S + N'_T} \hat{\epsilon}_T(h, \hat{h}_T(S)) + \frac{N_S}{N_S + N'_T} \hat{\epsilon}_S(h, \hat{h}_S(S))$$

$$+ \frac{N_S}{N_S + N'_T} \left( 2\mathbb{E} \left[ \sup \frac{1}{N_S} \sum_{i=1}^{N_S} \epsilon_i h(\mathbf{X}_i, y_i, p_i) \right] + C\sqrt{\frac{ln(2/\delta)}{N_S}} \right)$$

$$+ \frac{N_S}{N_S + N'_T} \Bigg( 2 C_f C_g W_1 \left( \mathbb{P}_S(G), \mathbb{P}_T(G) \right)$$

$$+ \min \left( |\epsilon_S(h, \hat{h}_S(\mathbf{X})) - \epsilon_S(h, \hat{h}_T(\mathbf{X}))|, |\epsilon_T(h, \hat{h}_S(\mathbf{X})) - \epsilon_T(h, \hat{h}_T(\mathbf{X}))| \right) \Bigg)$$

$$= \frac{N'_T}{N_S + N'_T} \hat{\epsilon}_T(h, \hat{h}_T(S)) + \frac{N_S}{N_S + N'_T} \Bigg( \hat{\epsilon}_S(h, \hat{h}_S(S)) + 2 C_f C_g W_1 \left( \mathbb{P}_S(G), \mathbb{P}_T(G) \right)$$

$$+ 2\mathbb{E} \left[ \sup \frac{1}{N_S} \sum_{i=1}^{N_S} \epsilon_i h(\mathbf{X}_i, y_i, p_i) \right] + C\sqrt{\frac{ln(2/\delta)}{N_S}}$$

$$+ \min \left( |\epsilon_S(h, \hat{h}_S(\mathbf{X}))) - \epsilon_S(h, \hat{h}_T(\mathbf{X}))|, |\epsilon_T(h, \hat{h}_S(\mathbf{X}))) - \epsilon_T(h, \hat{h}_T(\mathbf{X}))| \right) \Bigg)$$

where (a) is the outcome of applying the union bound with coefficient $\frac{N'_T}{N_S + N'_T}$, $\frac{N_S}{N_S + N'_T}$ respectively; (b) additionally adopt the assumption $N'_T \ll N_S$, following the sleight-of-hand in (Li et al., 2021a) Theorem 3.2.

Due to the sampels are selected with high confidence, thus, we have the following assumption:

$$\hat{\epsilon}_T \leq \epsilon_T \leq \hat{\epsilon}_S(h, \hat{h}(\mathbf{X}))) + 2\mathbb{E} \left[ \sup \frac{1}{N_S} \sum_{i=1}^{N_S} \epsilon_i h(\mathbf{X}_i, y_i, p_i) \right]$$

$$+ C\sqrt{\frac{ln(2/\delta)}{N_S}} + 2 C_f C_g W_1(\mathbb{P}_S(G), \mathbb{P}_T(G)) + \omega', \tag{36}$$

where $\omega' = \min_{\|g\|_{Lip} \leq C_g, \|f\|_{Lip} \leq C_f} \{\epsilon_S(h, \hat{h}) + \epsilon_T(h, \hat{h})\}$, $\hat{\epsilon}_T$ is the empirical risk on the high confidence samples, $\epsilon_T$ is the empirical risk on the target domain. Besides, we have:

$$\min(|\epsilon_S(h, \hat{h}_S(\mathbf{X}))) - \epsilon_S(h, \hat{h}_T(\mathbf{X}))|, |\epsilon_T(h, \hat{h}_S(\mathbf{X}))) - \epsilon_T(h, \hat{h}_T)|(\mathbf{X}))) \leq$$

$$\min \left( \epsilon_S(h, \hat{h}_S(\mathbf{X}))) + \epsilon_T(h, \hat{h}_S(\mathbf{X}))) \right) \tag{37}$$

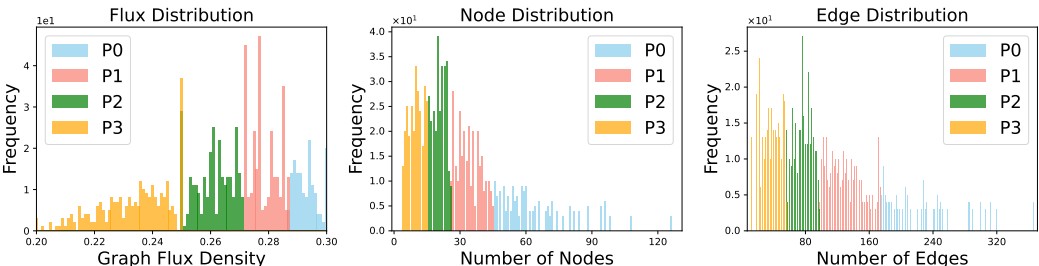

Figure 6: Visualization of different distributions on PROTEINS.

Then,

$$\epsilon_T(h, \hat{h}_T(\mathbf{X})) \leq \frac{N'_T}{N_S + N'_T} \hat{\epsilon}_T(h, \hat{h}_T(S)) + \frac{N_S}{N_S + N'_T} \Bigg( \hat{\epsilon}_S(h, \hat{h}_S(S)) + 2C_f C_g W_1 \left( \mathbb{P}_S(G), \mathbb{P}_T(G) \right)$$

$$+ 2\mathbb{E}\left[ \sup \frac{1}{N_S} \sum_{i=1}^{N_S} \epsilon_i h(\mathbf{X}_i, y_i, p_i) \right] + C\sqrt{\frac{ln(2/\delta)}{N_S}}$$

$$+ \min\left( |\epsilon_S(h, \hat{h}_S(\mathbf{X}))) - \epsilon_S(h, \hat{h}_T(\mathbf{X})))|, |\epsilon_T(h, \hat{h}_S(\mathbf{X}))) - \epsilon_T(h, \hat{h}_T(\mathbf{X})))| \right) \Bigg)$$

$$\leq \hat{\epsilon}_S(h, \hat{h}_S(S)) + 2\mathbb{E}\left[ \sup \frac{1}{N_S} \sum_{i=1}^{N_S} \epsilon_i h(\mathbf{X}_i, y_i, p_i) \right] + C\sqrt{\frac{ln(2/\delta)}{N_S}}$$

$$+ 2C_f C_g W_1 \left( \mathbb{P}_S(G), \mathbb{P}_T(G) \right) + \omega'. \tag{38}$$

# D  ALGORITHM

---
**Algorithm 1** Learning Algorithm of DeSGDA
---
**Input:** Source data $\mathcal{D}^s$; Target data $\mathcal{D}^t$.
**Output:** Node degree-aware personalized spiking graph encoder parameters $\theta$, domain discriminator $\gamma$.
1: Initialize model parameters.
2: **while** not convergence **do**
3:     Sample mini-batches $\mathcal{B}^s$ and $\mathcal{B}^t$ from source and target data, respectively;
4:     Forward propagation $\mathcal{B}^s$ and $\mathcal{B}^t$ through node degree-aware personalized spiking graph encoder;
5:     Pseudo-label Distilling;
6:     Calculate the loss function by Eq. 11;
7:     Update model parameters through back propagation;
8: **end while**
---

# E  COMPLEXITY ANALYSIS

Here we analyze the computational complexity of the proposed DeSGDA. The computational complexity primarily relies on node degree-aware personalized spiking representations. For a given graph $G$, $\|A\|_0$ denotes the number of nonzeros in the adjacency matrix. $d$ is the feature dimension. $L$ denote the layer number of GIN. $|V|$ is the number of nodes. $T$ denotes the number of time latency. The spiking graph encoder takes $\mathcal{O}\left( T \cdot L \cdot \left( \|A\|_0 \cdot d + |V| \cdot d^2 \right) \right)$ computational time for each graph. As a result, the complexity of our DeSGDA is proportional to both $|V|$ and $\|A\|_0$.

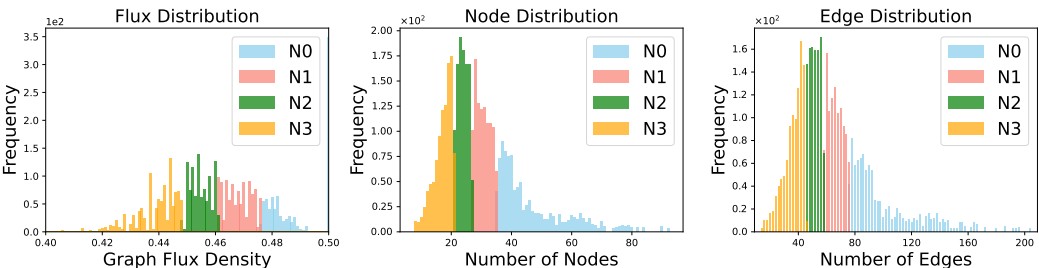

Figure 7: Visualization of different distributions on NCI1.

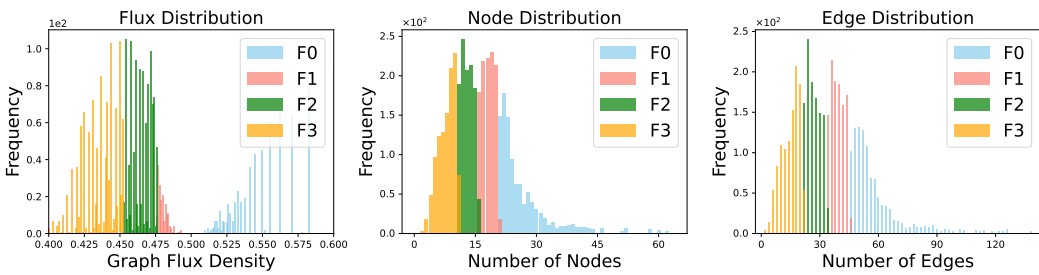

Figure 8: Visualization of different distributions on FRANKENSTEIN.

## F DATASET

Table 4: Statistics of the experimental datasets.

| Datasets | Graphs | Avg. Nodes | Avg. Edges | Classes |
|---|---|---|---|---|
| PROTEINS | 1,113 | 39.1 | 72.8 | 2 |
| NCI1 | 4,110 | 29.87 | 32.30 | 2 |
| MUTAGENICITY | 4,337 | 30.32 | 30.77 | 2 |
| FRANKENSTEIN | 4,337 | 16.9 | 17.88 | 2 |

### F.1 DATASET DESCRIPTION

We conduct extensive experiments on four public benchmark graph datasets from TUDataset. The dataset statistics can be found in Table 4, and their details are shown as follows:

- **PROTEINS.** The PROTEINS dataset (Dobson & Doig, 2003) consists of protein graphs. Each label indicates whether the protein graph is an enzyme. Nodes within these graphs represent amino acids connected by edges if neighbors along the amino acid sequence are spaced less than 6 Angstroms apart. We divide the PROTEINS dataset into four parts, i.e., P0, P1, P2, and P3, based on edge density, node density, and graph flux; the sub-datasets exhibit substantial domain disparities among them.

- **NCI1.** The NCI1 (Wale et al., 2008) dataset contains 4,100 chemical compounds with atoms as nodes and bonds as edges. Each label indicates the characteristics that hinder the growth of cancer cells. Like the PROTEINS dataset, we divide the NCI1 dataset into four parts, i.e., N0, N1, N2, and N3, based on edge density, node density, and graph flux.

- **FRANKENSTEIN.** The FRANKENSTEIN (Orsini et al., 2015) dataset consists of 4,337 molecular graphs, where nodes represent atoms and edges depict chemical bonds. Each graph is labeled to classify molecules based on their biological activity. Like the PROTEINS dataset, the entire FRANKENSTEIN dataset is divided into four segments (F0, F1, F2, and F3) based on edge density, node density, and graph flux.

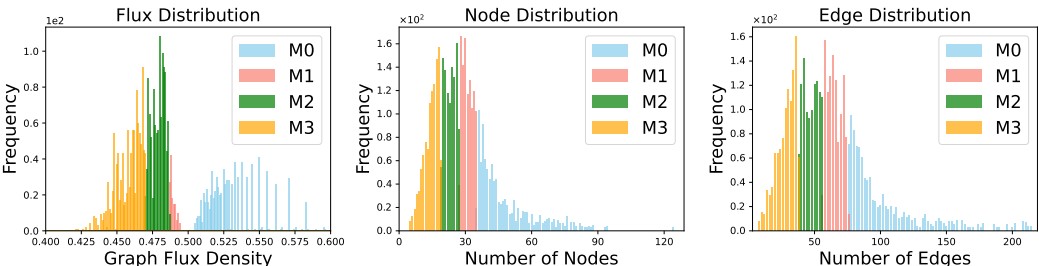

Figure 9: Visualization of different distributions on MUTAGNENICITY.

Table 5: The results of ablation studies on NCI1 (source → target). **Bold** results indicate the best performance per column.

| Methods | N0→N1 | N1→N0 | N0→N2 | N2→N0 | N0→N3 | N3→N0 | N1→N2 | N2→N1 | N1→N3 | N3→N1 | N2→N3 | N3→N2 | Avg. |
|---|---|---|---|---|---|---|---|---|---|---|---|---|---|
| DeSGDA w/o CA | 58.9 | 62.6 | 61.7 | 60.1 | 58.2 | 62.0 | 63.4 | 61.7 | 63.5 | 60.7 | 68.3 | 64.0 | 62.1 |
| DeSGDA w/o PL | 62.8 | 65.1 | 55.0 | 56.9 | 55.3 | 54.4 | 58.6 | 62.9 | 61.2 | 62.2 | 59.7 | 64.3 | 59.7 |
| DeSGDA w/o CF | 52.7 | 62.6 | 58.5 | 56.5 | 56.3 | 63.8 | 63.1 | 52.7 | 65.4 | 65.1 | 58.2 | 63.7 | 60.0 |
| DeSGDA w/o TL | 64.6 | 62.8 | 60.6 | 54.9 | 44.9 | 65.4 | 60.8 | 64.8 | 56.1 | 64.1 | 57.8 | 54.0 | 59.3 |
| DeSGDA w PT | 64.4 | 59.2 | 64.6 | 62.2 | 64.5 | 60.1 | 61.1 | 60.8 | 62.0 | 64.2 | 62.7 | 65.5 | 62.7 |
| DeSGDA w CL | 65.3 | 70.1 | 68.9 | 65.2 | 65.6 | 63.0 | 67.6 | 64.6 | 68.9 | 66.8 | 68.6 | **70.7** | 67.1 |
| DeSGDA | **68.5** | **71.4** | **70.1** | **69.0** | **68.9** | **66.3** | **69.6** | **70.2** | **71.1** | **69.3** | **74.4** | 70.0 | **70.1** |

- **MUTAGENICITY.** The MUTAGENICITY (Kazius et al., 2005) dataset contains 4,337 chemical compounds, each represented as a graph where nodes represent atoms and edges indicate bonds. Each graph can be used to identify mutagenic compounds, aiding studies in toxicology and chemical safety. Like the PROTEINS dataset, the entire MUTAGENICITY dataset is divided into four segments (M0, M1, M2, and M3) based on edge density, node density, and graph flux.

### F.2 DATA PROCESSING

In our implementation, we process these four TUDatasets by adding a self-loop connection for each node. Additionally, we utilize one-hot embeddings to represent node attributes for datasets (FRANKENSTEIN and MUTAGENICITY) where node features are unavailable.

## G BASELINES

In this part, we introduce the details of the compared baselines as follows:

**Graph kernel method.** We compare DeSGDA with one graph kernel method:

- **WL subtree:** Weisfeiler-Lehman (WL) subtree (Shervashidze et al., 2011) method is a graph kernel method, which calculates the graph similarity by a kernel function, where It encodes local neighborhood structures into subtree patterns, efficiently capturing the topology information contained in graphs.

**Graph neural networks.** We compare DeSGDA with four widely used general graph neural networks:

- **GCN**: GCN Kipf & Welling (2017) is a spectral-based neural network that iteratively updates node representations by aggregating information from neighboring nodes, effectively capturing both local graph structure and node features.
- **GIN**: GIN Xu et al. (2018) is a message-passing neural network designed to distinguish graph structures using an injective aggregation function, theoretically achieving the expressive power of the Weisfeiler-Lehman test.

Table 6: The results of ablation studies on FRANKENSTEIN (source → target). Bold results indicate the best performance. **Bold** results indicate the best performance.

| Methods | F0→F1 | F1→F0 | F0→F2 | F2→F0 | F0→F3 | F3→F0 | F1→F2 | F2→F1 | F1→F3 | F3→F1 | F2→F3 | F3→F2 | Avg. |
|---|---|---|---|---|---|---|---|---|---|---|---|---|---|
| DeSGDA w/o CA | 59.6 | 60.7 | 60.2 | 60.7 | 77.9 | 58.5 | 61.3 | 60.5 | 76.8 | 60.0 | 80.1 | 63.9 | 65.0 |
| DeSGDA w/o PL | 58.9 | 56.6 | 55.9 | 56.6 | 69.8 | 53.6 | 55.0 | 57.7 | 70.4 | 53.2 | 68.8 | 61.7 | 60.0 |
| DeSGDA w/o CF | 55.9 | 59.2 | 59.5 | 54.5 | 74.9 | 52.9 | 59.9 | 54.6 | 78.8 | 56.1 | 74.5 | 58.5 | 61.7 |
| DeSGDA w/o TL | 56.2 | 57.2 | 60.7 | 48.2 | 53.3 | 45.8 | 59.5 | 53.0 | 48.8 | 54.8 | 51.1 | 54.7 | 53.6 |
| DeSGDA w PT | 60.9 | 60.2 | 61.0 | 61.5 | 76.2 | 60.9 | 60.3 | 62.2 | 75.3 | 57.2 | 74.8 | 63.3 | 64.5 |
| DeSGDA w CL | 62.2 | 61.3 | 61.7 | 60.7 | 79.3 | 59.7 | 63.6 | 64.1 | 79.5 | **61.1** | 78.6 | 62.7 | 66.2 |
| DeSGDA | **63.5** | **64.1** | **62.6** | **63.0** | **81.1** | **62.4** | **65.5** | **65.4** | **81.9** | 60.8 | **82.0** | **65.9** | **68.1** |

Table 7: The results of ablation studies on MUTAGENICITY (source → target). Bold results indicate the best performance. **Bold** results indicate the best performance.

| Methods | M0→M1 | M1→M0 | M0→M2 | M2→M0 | M0→M3 | M3→M0 | M1→M2 | M2→M1 | M1→M3 | M3→M1 | M2→M3 | M3→M2 | Avg. |
|---|---|---|---|---|---|---|---|---|---|---|---|---|---|
| DeSGDA w/o CA | 58.3 | 63.3 | 62.1 | 60.1 | 81.4 | 62.7 | 63.8 | 59.1 | 80.3 | 57.1 | 78.9 | 63.0 | 65.8 |
| DeSGDA w/o PL | 58.8 | 57.6 | 60.2 | 61.4 | 73.3 | 54.5 | 63.7 | 61.0 | 61.2 | 61.9 | 71.0 | 60.2 | 62.1 |
| DeSGDA w/o CF | 54.7 | 58.3 | 59.6 | 58.7 | 72.1 | 60.7 | 57.9 | 54.9 | 78.0 | 57.4 | 63.2 | 60.4 | 61.3 |
| DeSGDA w/o TL | 56.5 | 60.7 | 59.1 | 60.7 | 43.4 | 56.3 | 51.0 | 56.7 | 53.3 | 56.9 | 63.1 | 52.5 | 55.9 |
| DeSGDA w PT | 62.1 | 64.3 | 63.9 | 61.8 | 77.3 | 62.4 | 66.6 | 63.9 | 73.5 | 66.1 | 82.1 | 64.9 | 67.4 |
| DeSGDA w CL | **66.1** | 62.6 | 63.3 | 64.7 | 81.5 | 60.7 | 69.8 | 67.0 | 82.2 | **67.8** | 82.4 | 63.6 | 69.6 |
| DeSGDA | 65.4 | **65.9** | **65.5** | **65.6** | **82.8** | **63.6** | **70.7** | **68.2** | **82.9** | 67.6 | **83.9** | **66.5** | **70.1** |

- **CIN**: CIN Bodnar et al. (2021) extends the Weisfeiler-Lehman framework by integrating cellular complexes into graph neural networks, allowing for the capture of higher-dimensional topological features.
- **GMT**: GMT Baek et al. (2021) utilizes self-attention mechanisms to dynamically adjust the importance of nodes based on their structural dependencies, thereby enhancing both adaptability and performance.

**Spiking graph neural networks.** We compare DeSGDA with two spiking graph neural networks:

- **SpikeGCN**: SpikeGCN (Zhu et al., 2022) introduces an end-to-end framework designed to integrate the fidelity characteristics of SNNs with graph node representations.
- **DRSGNN**: DRSGNN (Zhao et al., 2024) dynamically adapts to evolving graph structures and relationships through a novel architecture that updates node representations in real-time..

**Domain adaption methods.** We compare DeSGDA with two recent domain adaption methods:

- **CDAN**: CDAN (Long et al., 2018) employs a conditional adversarial learning strategy to reduce domain discrepancy by conditioning adversarial adaptation on discriminative information from multiple domains.
- **ToAlign**: ToAlign (Wei et al., 2021b) uses token-level alignment strategies within Transformer architectures to enhance cross-lingual transfer, optimizing the alignment of semantic representations across languages.
- **MetaAlign**: MetaAlign (Wei et al., 2021a) is a meta-learning framework for domain adaptation that dynamically aligns feature distributions across domains by learning domain-invariant representations.

**Graph domain adaptation methods.** We compare DeSGDA with six SOTA graph domain adaption methods:

- **DEAL**: DEAL (Yin et al., 2022) uses domain adversarial learning to align graph representations across different domains without labeled data, overcoming discrepancies between the source and target domains.
- **CoCo**: CoCo (Yin et al., 2023) leverages contrastive learning to align graph representations between source and target domains, enhancing domain adaptation by promoting intra-domain cohesion and inter-domain separation in an unsupervised manner.
- **SGDA**: SGDA (Qiao et al., 2023) utilizes labeled data from the source domain along with a limited amount of labeled data from the target domain to learn domain-invariant graph representations.

- **DGDA**: DGDA (Cai et al., 2024) employs generative models to capture the underlying distribution of graph data across domains, facilitating the transfer of graph structures and features by learning shared latent spaces.
- **A2GNN**: A2GNN (Liu et al., 2024a) introduces a novel propagation mechanism to enhance feature transferability across domains, improving the alignment of graph structures and node features in an unsupervised setting.
- **PA-BOTH**: PA-BOTH (Liu et al., 2024b) aligns node pairs between source and target graphs, optimizing feature correspondence at a granular level to improve the transferability of structural and feature information across domains.

For GCN and GIN, we use the Pytorch Geometric [4] to implement the model. For other baseline methods, we use the source codes provided by the corresponding paper. For all baseline methods, we vary the dropout rate in the range of {0.1,0.3,0.5,0.7} and then choose the best one. The hidden dimension in these methods is set to 256 for a fair comparison.

Table 8: Statistics of the SEED dataset.

| Datasets | Graphs | Avg. Nodes | Avg. Edges | Classes |
|---|---|---|---|---|
| SEED | 7,636 | 62.0 | 168.2 | 3 |

Table 9: The graph classification results (in %) on SEED dataset under edge density domain shift (source→target). E0, E1 and E2 denote the sub-datasets partitioned with edge density. **Bold** results indicate the best performance.

| Methods | E0→E1 | E1→E0 | E0→E2 | E2→E0 | E1→E2 | E2→E1 |
|---|---|---|---|---|---|---|
| GCN | $46.0 \pm 0.9$ | $47.8 \pm 1.0$ | $47.7 \pm 1.4$ | $49.7 \pm 0.7$ | $51.3 \pm 0.8$ | $49.8 \pm 1.0$ |
| GIN | $48.9 \pm 0.5$ | $50.3 \pm 0.6$ | $49.5 \pm 0.7$ | $50.7 \pm 1.0$ | $52.7 \pm 1.0$ | $52.1 \pm 0.9$ |
| DEAL | $53.5 \pm 0.4$ | $\mathbf{56.2 \pm 0.7}$ | $53.2 \pm 0.8$ | $53.7 \pm 1.1$ | $55.1 \pm 0.8$ | $56.0 \pm 0.7$ |
| CoCo | $53.9 \pm 0.5$ | $53.0 \pm 0.6$ | $54.1 \pm 0.7$ | $54.3 \pm 0.7$ | $55.3 \pm 1.0$ | $55.9 \pm 0.6$ |
| DeSGDA | $\mathbf{54.5 \pm 0.6}$ | $55.6 \pm 0.7$ | $\mathbf{54.6 \pm 0.5}$ | $\mathbf{54.5 \pm 0.7}$ | $\mathbf{55.8 \pm 1.1}$ | $\mathbf{56.6 \pm 0.8}$ |

## H MORE EXPERIMENTAL RESULTS

### H.1 MORE PERFORMANCE COMPARISON

In this part, we provide additional results for our proposed method DeSGDA compared with all baseline models across various datasets, as illustrated in Table 14 to Table 23. These results consistently show that DeSGDA outperforms the baseline models in most cases, validating the superiority of our proposed method. Additionally, the performance of DeSGDA with different GNN architectures on the NCI1, FRANKENSTEIN, and MUTAGENICITY datasets is shown in Figure 10. It is evident that GIN consistently outperforms other GNN architectures in most cases.

To verify the efffectiveness of DeSGDA in EEG data and multi-class scenarios, we conducted additional experiments using the SEED dataset (Duan et al., 2013; Zheng & Lu, 2015), which is a well-known EEG dataset for emotion classification. For EEG data processing, we utilized the torcheeg [5] library to convert standard EEG data into graph structures. During graph construction, we remove edges from each graph and partitioning the dataset into source and target domains based on edge density (Klepl et al., 2022). The statistics of SEED dataset is shown in Table 8. We compared the DeSGDA with two general graph neural networks (GCN Kipf & Welling (2017) and GIN Xu et al. (2018)) and two graph domain adaptation methods (DEAL Yin et al. (2022) and CoCo Yin et al. (2023)). The results in Table 9 show that DeSGDA still outperforms the other methods in most cases.

---

[4]https://www.pyg.org/
[5]https://github.com/torcheeg/torcheeg

Table 10: GPU memory consumption of different graph domain methods in training stage for each training epoch (in GB).

|  | DeSGDA | DEAL | CoCo | A2GNN | PA-BOTH |
|---|---|---|---|---|---|
| PROTEINS | 2.3 | 1.4 | 1.2 | 33.1 | 5.6 |
| NCI1 | 5.7 | 2.9 | 3.3 | 34.8 | 11.9 |
| MUTAGENICITY | 5.0 | 2.8 | 3.2 | 35.0 | 12.3 |
| FRANKENSTEIN | 2.9 | 2.4 | 1.9 | 33.6 | 7.3 |

Table 11: Time consumption of different graph domain methods in training stage for each training epoch (in seconds).

|  | DeSGDA | DEAL | CoCo | A2GNN | PA-BOTH |
|---|---|---|---|---|---|
| PROTEINS | 0.587 | 0.126 | 22.123 | 0.869 | 0.283 |
| NCI1 | 0.855 | 0.518 | 58.564 | 1.483 | 0.597 |
| MUTAGENICITY | 0.887 | 0.612 | 52.740 | 1.553 | 0.511 |
| FRANKENSTEIN | 0.663 | 0.375 | 26.837 | 1.016 | 0.275 |

## H.2 TRAINING TIME AND MEMORY COMPARISON

We provide detailed comparisons of GPU memory consumption and training time per epoch for DeSGDA and other graph domain adaptation methods under identical experimental settings in this part, as shown in Tables 10 and 11. It is worth noting that the training phase is typically conducted on more powerful hardware to achieve optimal performance within a reasonable time frame.

## H.3 MORE ABLATION STUDY

To validate the effectiveness of the different components in DeSGDA, we conduct more experiments with sive variants on NCI1, FRANKENSTEIN and MUTAGENICITY datasets, i.e., DeSGDA w/o CA, DeSGDA w/o PL, DeSGDA w/o CF, DeSGDA w/o TL, DeSGDA w PT and DeSGDA w CL. The results are shown in Table 5 , 6 and 7. From the results, we have similar observations as summarized in Section 5.4.

Additionally, we conduct ablation studies to examine the effect of directly replacing the SGNs with commonly used Graph Neural Networks (GNNs) for generating representations for DeSGDA: (1) DeSGDA w GCN: It replaces SGNs with GCN Kipf & Welling (2017); (2) DeSGDA w GIN: It replaces SGNs with GIN Xu et al. (2018); (3) DeSGDA w SAGE: It replaces SGNs with GraphSAGE Hamilton et al. (2017). The experimental results across the PROTEINS, NCI1, MUTAGENICITY, and FRANKENSTEIN datasets are shown in Table 12 and 13. However, the critical aspect of our work lies in the specific problem we set up, i.e., low-power and distribution shift environments. In this context, directly replacing SGNs with commonly used GNNs like GIN or GCN is not feasible, as these models are unsuitable for deployment on low-energy devices. As demonstrated in Section 5.3, GNN based methods have much higher energy consumption than the spike based methods.

## H.4 MORE SENSITIVITY ANALYSIS

In this part, we provide additional sensitivity analysis of the proposed DeSGDA with respect to the impact of its hyperparameters: the time latency $\tau$, degree threshold value $V_{th}^{degree}$ in SNNs, and balance ratio $\alpha$ on NCI1, FRANKENSTEIN, and MUTAGENICITY datasets. The results are illustrated in Figure 11, 12 and 13, where we observe trends similar to those discussed in Section 5.5.

Table 12: The results of DeSGDA with different widely used graph neural networks (GIN, GCN and SAGE) on PROTEINS and NCI1 dataset. Bold results indicate the best performance. **Bold** results indicate the best performance.

| Methods | P0→P1 | P1→P0 | P0→P2 | P2→P0 | N0→N1 | N1→N0 | N0→N2 | N2→N0 |
|---|---|---|---|---|---|---|---|---|
| DeSGDA w GCN | 76.6 | 70.5 | 71.8 | 74.1 | 66.3 | 68.3 | 68.5 | 67.1 |
| DeSGDA w SAGE | 75.8 | 73.3 | 72.4 | 75.2 | 67.2 | 69.5 | 66.6 | 68.4 |
| DeSGDA w GIN | 77.3 | 75.8 | 73.8 | 77.1 | **69.0** | 69.8 | 68.8 | 68.8 |
| DeSGDA | **78.7** | **78.4** | **74.8** | **77.6** | 68.5 | **71.4** | **70.1** | **69.0** |

Table 13: The results of DeSGDA with different widely used graph neural networks (GIN, GCN and SAGE) on MUTAGENICITY and FRANKENSTEIN dataset. Bold results indicate the best performance. **Bold** results indicate the best performance.

| Methods | M0→M1 | M1→M0 | M0→M2 | M2→M0 | F0→F1 | F1→F0 | F0→F2 | F2→F0 |
|---|---|---|---|---|---|---|---|---|
| DeSGDA w GCN | 60.9 | 60.7 | 63.6 | 60.6 | 61.3 | 62.2 | 60.7 | 61.1 |
| DeSGDA w SAGE | 61.1 | 62.3 | 64.2 | 61.1 | 61.9 | 62.6 | 61.3 | 61.6 |
| DeSGDA w GIN | 64.5 | 65.4 | 65.0 | 63.9 | 62.8 | 63.6 | 61.8 | 62.8 |
| DeSGDA | **65.4** | **65.9** | **65.5** | **65.6** | **63.5** | **64.1** | **62.6** | **63.0** |

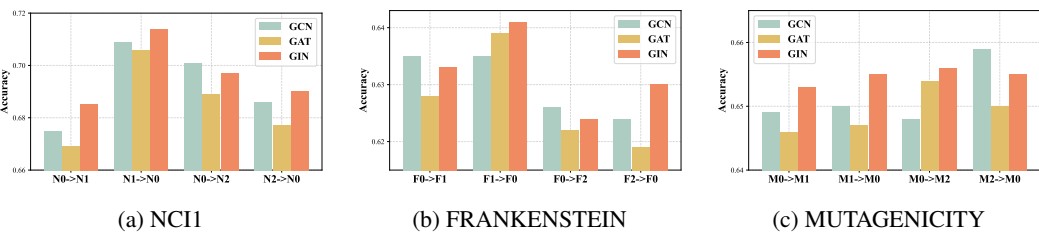

(a) NCI1  (b) FRANKENSTEIN  (c) MUTAGENICITY

Figure 10: The performance with different GNN architectures on different datasets.

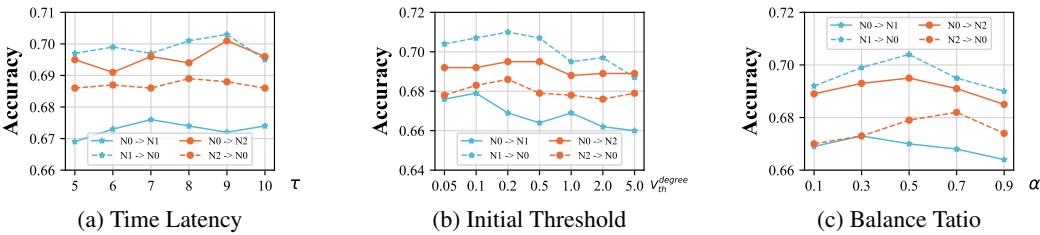

(a) Time Latency  (b) Initial Threshold  (c) Balance Tatio

Figure 11: Hyperparameter sensitivity analysis of time latency $\tau$, initial threshold $V_{th}^{degree}$ in SNNs, and balance ratio $\alpha$ on NCI1.

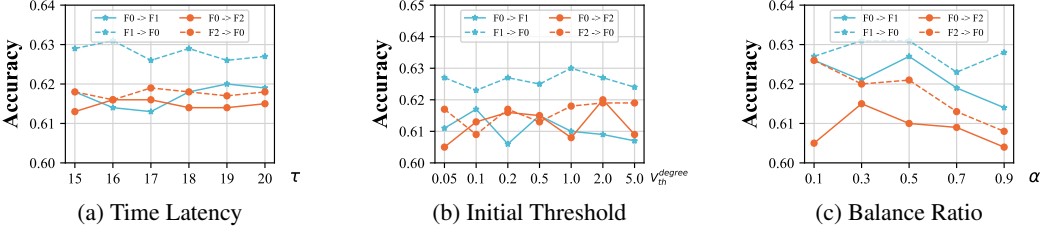

(a) Time Latency  (b) Initial Threshold  (c) Balance Ratio

Figure 12: Hyperparameter sensitivity analysis of time latency $\tau$, initial threshold $V_{th}^{degree}$ in SNNs, and balance ratio $\alpha$ on FRANKENSTEIN.

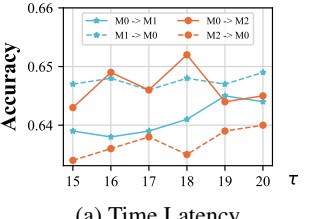 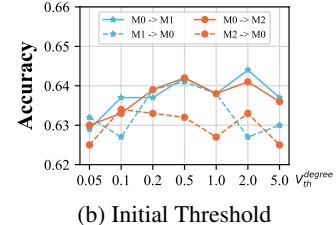 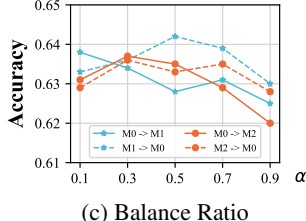

(a) Time Latency      (b) Initial Threshold      (c) Balance Ratio

Figure 13: Hyperparameter sensitivity analysis of time latency $\tau$, initial threshold $V_{th}^{degree}$ in SNNs, and balance ratio $\alpha$ on MUTAGENICITY.

Table 14: The graph classification results (in %) on PROTEINS under graph flux domain shift (source→target). P0, P1, P2, and P3 denote the sub-datasets partitioned with graph flux. **Bold** results indicate the best performance.

| Methods | P0→P1 | P1→P0 | P0→P2 | P2→P0 | P0→P3 | P3→P0 | P1→P2 | P2→P1 | P1→P3 | P3→P1 | P2→P3 | P3→P2 | Avg. |
|---|---|---|---|---|---|---|---|---|---|---|---|---|---|
| WL subtree | 73.4 | 72.7 | 70.5 | 73.0 | 72.8 | 59.0 | 66.5 | **71.6** | 60.6 | 58.3 | 76.3 | 64.0 | 68.2 |
| GCN | 57.2±2.7 | 62.8±1.7 | 67.6±0.5 | 58.5±1.3 | 67.7±0.4 | 61.0±0.3 | 65.0±0.8 | 51.1±1.3 | 65.6±2.2 | 55.4±0.4 | 68.5±3.1 | 67.7±0.5 | 62.3 |
| GIN | 69.3±2.3 | 65.8±0.8 | 69.3±1.7 | 69.8±1.6 | 71.4±2.1 | 52.4±1.8 | 64.0±2.4 | 65.7±3.2 | 53.4±3.7 | 58.1±0.8 | 72.6±0.3 | 64.6±2.3 | 64.7 |
| GMT | 67.8±1.3 | 69.6±0.7 | 74.5±0.5 | 67.6±2.5 | 69.9±2.1 | 55.8±0.7 | 74.8±1.4 | 60.1±2.4 | 71.4±3.3 | 51.5±0.5 | 69.0±0.5 | 63.3±1.3 | 66.3 |
| CIN | 62.6±0.5 | 59.4±0.5 | 64.0±0.9 | 58.5±1.8 | 71.9±1.7 | 60.6±2.1 | 63.7±0.5 | 61.2±2.1 | 73.2±0.5 | 57.7±3.0 | 68.1±0.4 | 58.5±2.7 | 63.3 |
| SpikeGCN | 66.1±0.4 | 67.8±0.7 | 68.5±0.9 | 67.2±0.3 | 70.1±1.1 | 68.6±1.3 | 66.1±0.7 | 65.8±1.2 | 74.5±0.9 | 65.4±0.6 | 73.8±1.1 | 70.1±1.6 | 68.6 |
| DRSGNN | 68.3±0.5 | 71.9±0.9 | 71.5±1.1 | 75.1±1.7 | 76.7±1.3 | 74.4±0.8 | 70.5±0.7 | 67.8±0.8 | 77.0±1.5 | 65.9±1.6 | 75.2±1.9 | 69.2±2.1 | 71.8 |
| CDAN | 75.6±0.5 | 70.5±0.6 | 71.6±0.5 | 69.8±0.5 | 76.6±0.8 | 71.4±0.3 | 71.4±0.3 | 72.1±0.3 | 75.5±0.7 | 74.3±0.8 | 78.2±1.1 | 74.0±0.8 | 73.4 |
| ToAlign | 51.1±0.6 | 55.8±0.1 | 63.3±0.2 | 55.8±0.4 | 68.1±0.7 | 55.8±0.3 | 63.3±0.5 | 51.1±0.2 | 68.1±1.0 | 51.1±0.4 | 68.1±0.6 | 63.3±0.2 | 59.6 |
| MetaAlign | 59.4±1.1 | 62.2±1.0 | 68.9±0.3 | 65.3±0.8 | 75.1±0.7 | 67.5±2.1 | 70.9±1.4 | 60.6±2.3 | 72.4±1.4 | 59.4±0.6 | 74.6±0.7 | 67.8±1.3 | 67.0 |
| DEAL | 76.6±0.4 | 62.8±0.8 | 72.8±1.3 | 67.3±2.2 | 77.2±2.3 | 67.6±1.9 | 71.2±1.6 | 56.0±2.5 | 73.9±2.1 | 66.0±0.3 | 76.4±1.1 | 65.5±2.1 | 69.4 |
| CoCo | 73.4±0.5 | 73.6±0.8 | 73.4±1.0 | 71.6±0.5 | 75.2±1.6 | 74.6±0.3 | 70.7±0.8 | 68.4±1.5 | 75.0±0.2 | 72.7±0.4 | 76.3±1.1 | 75.0±1.8 | 73.3 |
| SGDA | 63.8±0.8 | 65.2±0.5 | 66.7±0.3 | 59.1±1.2 | 62.3±0.7 | 60.6±0.4 | 65.2±0.8 | 61.8±1.0 | 64.5±1.3 | 60.9±0.8 | 69.0±0.5 | 64.9±1.1 | 62.9 |
| DGDA | 59.4±0.7 | 62.3±1.1 | 63.1±0.5 | 61.2±0.9 | 60.4±0.6 | 58.8±1.0 | 60.3±0.8 | 63.5±1.2 | 61.9±0.8 | 60.4±1.6 | 64.2±1.3 | 62.6±1.4 | 61.5 |
| A2GNN | 65.4±0.7 | 66.4±1.1 | 65.7±1.3 | 66.0±0.6 | 64.9±1.2 | 65.8±1.6 | 65.5±1.8 | 66.0±1.4 | 65.8±2.1 | 65.6±1.9 | 66.1±1.7 | 66.0±2.0 | 65.8 |
| PA-BOTH | 66.9±0.5 | 67.1±0.8 | 67.3±1.1 | 65.8±0.7 | 69.1±1.0 | 66.1±1.4 | 66.7±1.3 | 67.4±1.4 | 66.3±1.8 | 66.0±1.2 | 66.8±0.8 | 66.3±1.5 | 66.8 |
| DeSGDA | **78.7±1.3** | **78.4±1.1** | **74.8±0.6** | **77.6±0.9** | **79.5±1.2** | **76.7±0.8** | **74.9±0.7** | 71.2±1.7 | **79.5±1.4** | **72.8±0.8** | **81.0±1.5** | **75.1±1.0** | **76.7** |

Table 15: The graph classification results (in %) on FRANKENSTEIN under graph flux domain shift (source→target). F0, F1, F2, and F3 denote the sub-datasets partitioned with graph flux. **Bold** results indicate the best performance. OOM means out of memory.

| Methods | F0→F1 | F1→F0 | F0→F2 | F2→F0 | F0→F3 | F3→F0 | F1→F2 | F2→F1 | F1→F3 | F3→F1 | F2→F3 | F3→F2 | Avg. |
|---|---|---|---|---|---|---|---|---|---|---|---|---|---|
| WL subtree | 58.4 | 51.8 | 58.7 | 51.3 | 64.3 | 48.9 | 64.9 | 58.9 | 78.5 | 54.6 | 57.1 | 61.3 | 59.1 |
| GCN | 56.2±0.2 | 59.0±1.3 | 41.4±0.4 | 45.8±0.5 | 21.2±0.7 | 41.4±1.7 | 42.5±1.6 | 49.0±0.4 | 24.1±1.6 | 44.8±0.7 | 81.4±0.3 | 58.8±0.2 | 47.1 |
| GIN | 60.7±0.6 | 58.0±1.0 | 61.0±2.3 | 58.9±2.3 | 77.5±2.2 | 45.3±2.5 | 62.5±0.2 | 59.2±3.0 | 71.4±2.8 | 49.8±1.7 | 77.9±1.4 | 59.9±0.5 | 61.8 |
| GMT | 56.2±0.4 | 59.8±0.2 | 41.4±0.3 | 59.8±0.7 | 21.2±1.1 | 59.8±0.5 | 41.4±0.2 | 56.2±0.2 | 21.1±1.1 | 56.2±1.4 | 78.8±0.6 | 58.6±0.8 | 50.9 |
| CIN | 57.8±1.1 | 60.1±0.7 | 58.6±0.2 | 59.8±0.4 | 78.9±0.1 | 59.9±0.4 | 58.8±0.3 | 57.4±0.5 | 78.8±0.6 | 57.7±1.2 | 78.8±0.7 | 60.1±1.1 | 63.9 |
| SpikeGCN | 56.1±0.7 | 59.7±1.0 | 58.8±0.6 | 57.8±0.2 | 77.1±1.3 | 53.2±1.6 | 41.4±1.9 | 56.1±1.5 | 70.1±0.9 | 59.9±1.5 | 76.8±1.3 | 58.5±1.4 | 60.4 |
| DRSGNN | 60.2±0.9 | 59.9±0.8 | 57.3±1.2 | 59.0±1.0 | 74.2±1.9 | 54.6±1.7 | 58.5±1.5 | 58.9±1.8 | 77.7±2.3 | 56.9±2.0 | 78.9±2.4 | 58.8±1.6 | 62.9 |
| CDAN | 60.9±0.7 | 59.8±0.5 | 61.1±1.3 | 61.0±0.2 | 80.5±1.2 | 59.8±0.3 | 64.0±0.4 | 61.4±0.1 | 81.8±0.1 | 58.0±1.2 | 81.8±0.3 | 63.8±0.7 | 66.1 |
| ToAlign | 56.2±1.2 | 59.8±0.2 | 41.4±0.1 | 59.8±0.2 | 21.1±0.3 | 59.8±0.7 | 41.4±1.1 | 56.2±1.2 | 21.1±0.4 | 56.2±0.6 | 21.1±1.3 | 41.4±0.5 | 44.6 |
| MetaAlign | 57.3±2.4 | 59.1±1.1 | 60.9±1.5 | 60.2±0.4 | 80.3±2.1 | 60.4±0.6 | 64.0±1.1 | 64.0±1.1 | 64.9±0.6 | 81.4±1.2 | 58.5±2.3 | 80.8±0.5 | 63.4±1.8 | 65.9 |
| DEAL | **65.3±0.6** | 64.0±0.2 | 61.3±0.6 | 61.0±0.9 | 78.3±2.1 | 55.5±1.8 | 64.9±1.2 | 64.8±1.1 | 80.1±1.3 | 60.1±1.1 | 81.8±0.4 | 65.7±0.7 | 66.9 |
| CoCo | 63.5±2.4 | 61.5±1.0 | **64.4±1.0** | 61.2±0.7 | **81.7±0.4** | 55.0±1.6 | 64.5±0.6 | 64.6±1.1 | 80.4±1.5 | 60.6±1.5 | 81.5±0.6 | 62.2±1.7 | 66.8 |
| SGDA | 55.7±0.5 | 55.4±0.9 | 54.8±0.3 | 55.3±0.7 | 56.1±0.5 | 55.4±0.8 | 53.2±1.1 | 55.1±0.6 | 58.4±0.4 | 55.3±0.5 | 57.7±1.0 | 54.9±0.6 | 55.7 |
| DGDA | OOM | OOM | OOM | OOM | OOM | OOM | OOM | OOM | OOM | OOM | OOM | OOM | OOM |
| A2GNN | 56.0±0.3 | 56.3±0.6 | 55.6±0.4 | 57.3±0.7 | 58.6±0.6 | 55.9±0.9 | 55.5±0.5 | 55.3±0.2 | 61.2±1.3 | 56.6±0.9 | 65.5±0.8 | 56.0±1.0 | 57.5 |
| PA-BOTH | 62.2±0.5 | 60.7±0.7 | 61.5±0.6 | 61.2±1.0 | 61.9±1.3 | 61.1±0.8 | 62.3±0.4 | 61.7±0.8 | 62.0±0.9 | 61.1±1.2 | 61.2±1.5 | 60.9±0.6 | 61.5 |
| DeSGDA | 63.5±1.1 | **64.1±0.9** | 62.6±1.3 | **63.0±0.8** | 81.1±1.2 | **62.4±1.5** | **65.5±0.6** | **65.4±1.7** | **81.9±1.0** | **60.8±1.4** | **82.0±2.1** | **65.9±1.8** | **68.1** |

Table 16: The graph classification results (in %) on MUTAGENICITY under graph flux domain shift (source→target). M0, M1, M2, and M3 denote the sub-datasets partitioned with graph flux. **Bold** results indicate the best performance. OOM means out of memory.

| Methods | M0→M1 | M1→M0 | M0→M2 | M2→M0 | M0→M3 | M3→M0 | M1→M2 | M2→M1 | M1→M3 | M3→M1 | M2→M3 | M3→M2 | Avg. |
|---|---|---|---|---|---|---|---|---|---|---|---|---|---|
| WL subtree | **74.4** | 72.9 | 64.9 | **68.9** | 49.1 | 59.8 | 70.0 | 70.5 | 76.9 | 60.7 | 82.6 | 70.5 | 68.5 |
| GCN | 63.1±1.0 | 68.1±0.3 | 48.8±0.4 | 62.6±0.3 | 29.1±2.1 | 38.8±0.3 | 54.3±0.1 | 61.8±0.8 | 30.4±0.2 | 43.6±0.3 | 67.8±0.1 | 57.9±1.3 | 52.2 |
| GIN | 68.1±1.6 | 74.2±0.6 | 59.6±2.3 | 65.2±1.4 | 40.3±2.7 | 54.6±1.8 | 61.3±1.1 | 63.1±3.2 | 71.6±3.0 | 60.0±1.4 | 79.7±1.3 | 69.2±0.7 | 63.9 |
| GMT | 56.5±0.3 | 60.7±0.4 | 57.9±0.2 | 40.2±1.2 | 80.6±0.4 | 39.3±0.6 | 57.9±1.1 | 45.0±2.1 | 80.6±0.5 | 43.5±1.1 | 80.6±1.4 | 57.9±2.2 | 58.4 |
| CIN | 64.1±3.0 | 61.3±0.5 | 63.5±2.3 | 63.6±1.5 | 78.2±0.5 | 63.9±2.7 | 60.6±1.5 | 57.0±0.4 | 73.7±3.2 | 61.4±1.0 | 79.1±2.1 | 61.1±1.9 | 65.6 |
| SpikeGCN | 56.4±1.2 | 60.7±0.9 | 59.5±1.3 | 57.7±1.6 | 59.0±1.1 | 60.1±1.8 | 54.2±1.0 | 59.9±2.1 | 50.2±2.6 | 55.1±1.7 | 80.1±2.5 | 57.9±2.2 | 59.3 |
| DRSGNN | 56.7±0.7 | 61.0±1.1 | 57.2±1.0 | 57.7±1.6 | 52.1±1.4 | 55.2±1.5 | 59.4±1.8 | 56.3±1.7 | 75.9±2.3 | 60.7±1.9 | 80.6±0.8 | 58.0±1.3 | 60.9 |
| CDAN | 62.8±0.3 | 68.2±0.6 | 63.6±0.6 | 66.9±1.7 | 81.2±0.5 | 65.0±2.1 | 65.8±0.2 | 64.7±1.2 | 80.7±0.1 | 62.5±2.3 | 82.4±0.4 | 66.0±0.5 | 69.1 |
| ToAlign | 43.5±0.4 | 39.3±0.7 | 57.9±1.0 | 39.3±1.4 | 80.6±1.1 | 39.3±0.7 | 57.9±0.3 | 43.5±2.1 | 80.6±1.8 | 43.5±0.4 | 80.6±0.9 | 57.9±1.0 | 55.3 |
| MetaAlign | 63.1±2.5 | 68.8±2.6 | 63.3±0.6 | 65.2±2.2 | 81.9±0.1 | 64.5±1.4 | 65.0±0.6 | 68.3±0.6 | 81.0±0.3 | 65.2±0.2 | 82.5±0.4 | 68.3±0.6 | 69.7 |
| DEAL | 64.6±0.5 | 65.5±0.8 | 64.2±1.0 | 63.1±2.1 | 82.7±0.8 | 62.8±0.7 | 70.2±0.4 | 67.3±0.4 | 79.6±0.1 | 63.9±1.4 | 75.7±0.3 | 67.0±0.2 | 68.9 |
| CoCo | 65.7±1.8 | **74.1±0.7** | 65.1±0.2 | 67.6±0.9 | 80.5±1.3 | 56.5±1.7 | 68.4±1.3 | **70.7±0.4** | 78.9±1.2 | 67.3±0.3 | 83.7±0.1 | **71.5±0.9** | 70.8 |
| SGDA | OOM | OOM | OOM | OOM | OOM | OOM | OOM | OOM | OOM | OOM | OOM | OOM | OOM |
| DGDA | OOM | OOM | OOM | OOM | OOM | OOM | OOM | OOM | OOM | OOM | OOM | OOM | OOM |
| A2GNN | 55.4±0.3 | 55.7±0.7 | 55.6±0.5 | 54.7±0.8 | 63.3±1.0 | 56.6±0.9 | 55.3±0.6 | 55.7±0.5 | 65.5±0.8 | 56.6±1.2 | 69.9±1.4 | 55.0±0.5 | 58.3 |
| PA-BOTH | 61.2±0.9 | 62.0±0.4 | 60.7±0.8 | 61.7±0.5 | 60.9±1.2 | 61.1±0.7 | 61.5±0.9 | 60.2±1.2 | 61.3±1.5 | 61.8±0.8 | 62.2±0.9 | 62.0±1.0 | 61.4 |
| DeSGDA | 65.4±1.3 | 65.9±0.9 | **65.5±1.4** | 64.8±1.1 | **82.8±1.6** | **63.6±1.5** | **70.7±1.8** | 68.2±0.7 | **82.9±1.2** | **67.6±0.8** | **83.9±1.0** | 66.5±1.2 | **70.9** |

Table 17: The graph classification results (in %) on PROTEINS under node domain shift (source→target). P0, P1, P2, and P3 denote the sub-datasets partitioned with node. **Bold** results indicate the best performance. OOM means out of memory.

| Methods | P0→P1 | P1→P0 | P0→P2 | P2→P0 | P0→P3 | P3→P0 | P1→P2 | P2→P1 | P1→P3 | P3→P1 | P2→P3 | P3→P2 | Avg. |
|---|---|---|---|---|---|---|---|---|---|---|---|---|---|
| WL subtree | 69.1 | 59.7 | 61.2 | 75.9 | 41.6 | 83.5 | 61.5 | 72.7 | 24.7 | 72.7 | 63.1 | 62.9 | 62.4 |
| GCN | 73.7±0.3 | 82.7±0.4 | 57.6±0.2 | 84.0±1.3 | 24.4±0.4 | 17.3±0.2 | 57.6±0.1 | 70.9±0.7 | 24.4±0.5 | 26.3±0.1 | 37.5±0.2 | 42.5±0.8 | 49.9 |
| GIN | 71.8±2.7 | 70.2±4.7 | 58.5±4.3 | 56.9±4.9 | 74.2±1.7 | 78.2±3.3 | 63.3±2.7 | 67.1±3.8 | 35.9±4.2 | 61.0±2.4 | 71.9±2.1 | 65.1±1.0 | 64.5 |
| GMT | 73.7±0.2 | 82.7±0.1 | 57.6±0.3 | 83.1±0.5 | 75.6±1.4 | 17.3±0.6 | 73.7±0.6 | 75.6±0.4 | 26.3±1.2 | 75.6±0.7 | 42.4±0.5 | | 61.8 |
| CIN | 74.1±0.6 | 83.8±1.0 | 60.1±2.1 | 78.6±3.1 | 75.6±0.2 | 74.8±3.7 | 63.9±2.7 | 74.1±0.6 | 57.0±4.3 | 58.9±3.3 | 75.6±0.7 | 63.6±1.0 | 70.0 |
| SpikeGCN | 71.8±0.9 | 80.9±1.2 | 64.9±1.4 | 79.1±2.2 | 71.1±1.9 | 73.8±1.6 | 62.4±2.0 | 71.8±2.3 | 70.1±2.4 | 66.9±1.9 | 72.1±1.9 | 64.5±1.7 | 70.9 |
| DRSGNN | 73.6±1.1 | 81.3±1.5 | 64.6±1.2 | 80.6±1.4 | 70.2±1.7 | 76.1±2.3 | 64.1±1.5 | 71.9±1.9 | 70.4±2.0 | 64.1±3.1 | 74.7±1.4 | 64.3±1.1 | 71.3 |
| CDAN | 75.9±1.0 | 83.1±0.6 | 60.8±0.6 | 82.6±0.2 | 75.8±0.3 | 70.9±2.4 | 64.7±0.3 | **77.7±0.6** | 73.3±1.8 | 75.4±0.7 | 75.8±0.4 | 67.1±0.8 | 73.6 |
| ToAlign | 73.7±0.4 | 82.7±0.3 | 57.6±0.6 | 82.7±0.8 | 24.4±0.1 | 82.7±0.3 | 57.6±0.4 | 73.7±0.2 | 24.4±0.7 | 73.7±0.3 | 24.4±0.5 | 57.6±0.4 | 59.6 |
| MetaAlign | 74.3±0.8 | 83.3±2.2 | 60.6±1.7 | 71.2±2.1 | 76.3±0.3 | 77.3±2.4 | 64.6±1.2 | 72.0±1.0 | 76.0±0.5 | 73.3±1.8 | 74.4±1.7 | 56.9±1.4 | 71.7 |
| DEAL | 75.4±1.2 | 78.0±2.4 | 68.1±1.9 | 80.8±2.1 | 73.8±1.4 | 80.6±2.3 | 64.7±2.4 | 74.7±2.4 | 74.7±1.6 | 71.0±2.1 | 68.1±2.6 | 70.3±0.4 | 73.4 |
| CoCo | 74.8±0.6 | 84.1±1.1 | 65.5±0.4 | 83.6±1.1 | 72.4±2.9 | 83.1±0.4 | 69.7±0.5 | 75.8±0.7 | 71.4±2.3 | 73.4±1.3 | 72.5±2.7 | 66.4±1.7 | 74.4 |
| SGDA | 64.2±0.5 | 61.0±0.7 | 66.9±1.2 | 61.9±0.9 | 65.4±1.6 | 66.5±1.0 | 64.6±1.1 | 60.1±0.5 | 66.3±1.3 | 59.3±0.8 | 66.0±1.6 | 66.2±1.3 | 64.1 |
| DGDA | 58.1±0.4 | 58.6±0.6 | 58.9±1.0 | 61.0±0.9 | 59.6±0.7 | 60.2±1.5 | 56.7±0.6 | 56.8±0.8 | 58.1±0.4 | 57.0±1.2 | 62.2±1.6 | | 58.9 |
| A2GNN | 65.7±0.6 | 65.9±0.8 | 66.3±0.9 | 65.6±1.1 | 65.2±1.4 | 65.6±1.3 | 65.9±1.7 | 65.8±1.6 | 65.0±1.5 | 66.1±1.2 | 65.2±1.9 | 65.9±1.8 | 65.7 |
| PA-BOTH | 61.0±0.8 | 61.2±1.3 | 60.3±0.6 | 66.7±2.1 | 63.7±1.5 | 61.9±2.0 | 66.2±1.4 | 69.9±2.3 | 68.0±0.7 | 69.4±1.8 | 61.5±0.4 | 67.6±1.0 | 64.9 |
| DeSGDA | **77.6±0.9** | **84.3±1.1** | **70.5±0.6** | **84.8±1.4** | **76.6±0.7** | **83.9±0.9** | **71.9±0.6** | 76.9±1.1 | **76.1±0.8** | 73.7±1.0 | **76.0±1.2** | **70.4±0.7** | **76.8** |

Table 18: The graph classification results (in %) on NCI1 under node domain shift (source→target). P0, P1, P2, and P3 denote the sub-datasets partitioned with node. **Bold** results indicate the best performance.

| Methods | N0→N1 | N1→N0 | N0→N2 | N2→N0 | N0→N3 | N3→N0 | N1→N2 | N2→N1 | N1→N3 | N3→N1 | N2→N3 | N3→N2 | Avg. |
|---|---|---|---|---|---|---|---|---|---|---|---|---|---|
| WL subtree | **73.5** | 79.5 | 64.8 | 75.9 | 58.9 | 68.4 | 72.5 | 72.0 | 69.7 | 63.6 | **76.1** | **74.0** | 70.7 |
| GCN | 51.2±0.1 | 71.1±0.4 | 42.7±0.4 | 27.8±0.3 | 32.1±1.1 | 27.0±0.2 | 55.2±0.6 | 50.5±0.7 | 50.9±1.1 | 49.1±0.3 | 67.1±0.6 | 57.3±0.6 | 48.5 |
| GIN | 66.9±2.2 | 78.9±2.3 | 60.3±3.1 | 72.8±0.3 | 51.1±0.6 | 68.6±1.8 | 63.5±2.1 | 67.8±3.7 | 65.9±1.7 | 60.3±1.8 | 71.1±1.1 | 67.2±1.3 | 66.2 |
| GMT | 50.9±0.5 | 73.0±0.1 | 57.3±0.3 | 73.0±0.4 | 66.5±0.2 | 73.0±0.3 | 50.9±0.1 | 66.5±0.4 | 58.3±0.2 | 66.5±0.3 | 72.8±0.3 | | 65.1 |
| CIN | 60.1±0.7 | 73.1±1.1 | 57.5±0.2 | 73.0±0.4 | 66.5±1.1 | 73.1±0.7 | 58.5±2.1 | 52.9±1.4 | 66.5±1.3 | 56.1±0.1 | 66.5±0.4 | 57.4±0.7 | 63.4 |
| SpikeGCN | 63.3±0.4 | 72.6±0.7 | 60.6±0.2 | 73.1±0.6 | 65.4±0.6 | 66.6±1.2 | 64.3±1.4 | 64.7±1.0 | 65.6±1.2 | 59.8±0.8 | 70.1±1.7 | 60.9±1.4 | 65.6 |
| DRSGNN | 63.9±0.6 | 73.3±1.2 | 56.9±1.5 | 72.5±1.3 | 66.7±2.1 | 64.4±1.8 | 64.8±1.9 | 66.7±1.4 | 58.4±1.6 | 68.6±2.0 | 61.4±1.3 | | 65.3 |
| CDAN | 57.1±0.4 | 75.0±0.7 | 61.2±0.4 | 73.7±0.1 | 68.2±0.4 | 73.3±0.3 | 60.2±0.1 | 56.5±1.4 | 68.2±0.2 | 53.9±1.4 | 68.4±0.2 | 59.6±0.5 | 64.6 |
| ToAlign | 49.1±0.3 | 27.0±0.6 | 57.3±0.5 | 27.0±0.4 | 66.5±0.5 | 27.0±0.2 | 57.3±0.3 | 49.1±0.4 | 66.5±0.2 | 49.1±0.3 | 66.5±0.1 | 57.3±0.4 | 50.0 |
| MetaAlign | 65.6±1.8 | 77.7±0.2 | 63.5±1.4 | 75.7±0.7 | 66.4±0.3 | 74.0±0.3 | 66.3±1.1 | 64.6±1.2 | 66.7±0.2 | 59.5±2.6 | 66.7±0.3 | 66.7±2.7 | 67.8 |
| DEAL | 64.0±0.9 | 71.9±1.2 | 61.4±0.3 | 73.3±0.3 | 64.9±1.4 | 71.9±1.9 | 62.5±2.1 | 66.2±0.5 | 54.2±1.4 | 55.6±0.8 | 64.6±0.4 | 58.8±0.4 | 64.1 |
| CoCo | 69.7±0.1 | **80.4±0.4** | 64.7±1.2 | 76.5±0.4 | 65.0±1.7 | 73.9±0.3 | 68.9±1.3 | 70.7±0.9 | 68.2±1.2 | 61.4±1.7 | 73.0±0.1 | 65.2±0.9 | 69.8 |
| SGDA | OOM | OOM | OOM | OOM | OOM | OOM | OOM | OOM | OOM | OOM | OOM | OOM | OOM |
| DGDA | OOM | OOM | OOM | OOM | OOM | OOM | OOM | OOM | OOM | OOM | OOM | OOM | OOM |
| A2GNN | 59.0±0.6 | 58.3±1.1 | 58.5±0.8 | 58.6±1.3 | 58.7±1.0 | 59.0±0.7 | 58.5±1.1 | 58.7±1.5 | 59.1±0.6 | 58.3±1.2 | 58.6±0.7 | 59.0±0.5 | 58.7 |
| PA-BOTH | 57.7±0.4 | 58.0±0.6 | 57.9±0.5 | 56.9±0.8 | 57.4±0.6 | 58.3±0.5 | 57.1±1.2 | 58.8±0.9 | 58.1±0.7 | 58.0±0.9 | 57.9±0.5 | 58.3±0.8 | 57.9 |
| DeSGDA | 64.4±1.2 | 76.9±1.5 | **64.9±0.9** | **76.6±1.2** | **68.6±1.8** | **74.1±1.3** | 66.9±0.8 | 65.1±1.2 | **69.9±1.5** | **63.9±2.0** | 70.9±1.6 | 64.2±1.4 | 69.0 |

Table 19: The graph classification results (in %) on FRANKENSTEIN under node domain shift (source→target). F0, F1, F2, and F3 denote the sub-datasets partitioned with node. **Bold** results indicate the best performance. OOM means out of memory.

| Methods | F0→F1 | F1→F0 | F0→F2 | F2→F0 | F0→F3 | F3→F0 | F1→F2 | F2→F1 | F1→F3 | F3→F1 | F2→F3 | F3→F2 | Avg. |
|---|---|---|---|---|---|---|---|---|---|---|---|---|---|
| WL subtree | 65.7 | 71.8 | 57.9 | 71.1 | 47.4 | 43.4 | 65.5 | 75.1 | 45.3 | 34.9 | 52.7 | 49.8 | 56.7 |
| GCN | 70.6±2.1 | 60.3±1.5 | 60.5±3.4 | 62.3±1.1 | 58.4±0.5 | 43.2±0.2 | 63.8±1.0 | 70.3±0.3 | 50.6±1.0 | 32.8±0.3 | 50.1±0.4 | 42.2±0.2 | 55.4 |
| GIN | 66.7±2.1 | 73.7±2.4 | 57.3±3.1 | 69.4±2.3 | 58.6±0.4 | 43.1±0.3 | 66.4±2.7 | 74.8±1.8 | 42.2±1.6 | 33.5±1.0 | 57.4±0.8 | 43.9±2.3 | 57.2 |
| GMT | 67.3±0.3 | 56.8±0.4 | 58.0±0.2 | 56.8±0.2 | 60.6±0.3 | 56.8±0.5 | 57.8±0.1 | 67.3±0.1 | 39.5±0.3 | 67.3±0.2 | 39.5±0.5 | 57.8±0.4 | 57.1 |
| CIN | 67.6±0.4 | 63.7±2.1 | 58.9±1.0 | 56.8±0.4 | 63.6±0.4 | 59.5±2.7 | 58.7±1.2 | 67.0±0.5 | 61.7±1.6 | 67.8±0.7 | 62.2±2.1 | 56.0±1.3 | 61.9 |
| SpikeGCN | 67.2±0.5 | 57.2±1.2 | 57.9±0.8 | 57.1±0.6 | 58.9±1.6 | 60.0±1.2 | 67.2±0.9 | 53.9±2.1 | 64.4±0.8 | 57.8±1.0 | 59.9±1.0 | 59.9±1.0 | 60.2 |
| DRSGNN | 67.4±0.4 | 58.4±1.0 | 59.0±1.2 | 57.4±0.5 | 62.3±1.1 | 60.4±1.3 | 61.1±1.6 | 67.9±1.5 | 56.2±1.8 | 66.2±2.1 | 60.9±1.4 | 58.6±2.5 | 61.3 |
| CDAN | 72.9±0.4 | 72.7±0.4 | 65.4±0.3 | 72.9±0.1 | 61.2±0.3 | 70.3±0.2 | 65.7±0.4 | 72.7±0.1 | 61.0±0.1 | 72.1±1.2 | 60.7±0.2 | 65.3±0.6 | 67.7 |
| ToAlign | 32.7±2.0 | 43.2±0.1 | 42.2±1.3 | 43.2±0.9 | 60.5±0.7 | 43.2±1.2 | 42.2±0.4 | 32.7±1.2 | 60.5±0.9 | 32.7±0.3 | 60.5±0.7 | 42.2±0.4 | 44.7 |
| MetaAlign | 67.3±0.7 | 56.8±0.2 | 57.8±0.6 | 56.8±0.4 | 60.5±1.3 | 56.8±0.8 | 57.8±0.1 | 67.3±1.2 | 60.5±0.4 | 67.3±0.6 | 60.5±0.7 | 57.8±0.6 | 60.6 |
| DEAL | 75.0±0.9 | **76.3±2.4** | 65.9±1.8 | **77.5±2.7** | 60.3±4.5 | 69.7±3.2 | 67.2±1.5 | 75.3±1.7 | 57.4±4.1 | 71.1±2.2 | **65.7±2.7** | 66.4±1.6 | 69.0 |
| CoCo | 74.2±1.7 | 74.3±0.6 | 65.9±1.2 | 72.7±2.1 | 61.1±0.2 | 71.0±1.7 | 68.6±0.3 | 75.9±0.2 | 60.7±0.2 | **73.9±0.4** | 59.7±1.1 | **67.3±0.8** | 68.8 |
| SGDA | 55.9±0.6 | 57.1±0.5 | 56.1±0.4 | 54.6±0.8 | 55.8±1.1 | 57.7±0.6 | 54.3±0.7 | 53.6±1.3 | 59.1±0.8 | 56.7±0.6 | 55.4±1.2 | 53.8±0.5 | 55.9 |
| DGDA | OOM | OOM | OOM | OOM | OOM | OOM | OOM | OOM | OOM | OOM | OOM | OOM | OOM |
| A2GNN | 55.9±0.7 | 55.7±0.4 | 56.6±0.6 | 57.1±1.0 | 56.1±1.2 | 55.8±0.5 | 56.5±0.7 | 55.5±0.4 | 55.9±0.8 | 56.2±0.6 | 56.5±1.5 | 56.0±0.5 | 56.2 |
| PA-BOTH | 56.4±0.5 | 55.9±0.6 | 56.0±0.5 | 56.4±0.4 | 56.3±0.6 | 57.7±0.7 | 56.6±0.2 | 58.8±0.9 | 56.9±0.7 | 57.2±0.3 | 56.5±0.5 | 58.3±0.8 | 56.9 |
| DeSGDA | **75.2±0.8** | 74.3±1.2 | **68.0±1.5** | 73.6±2.0 | **62.3±1.7** | **71.2±2.5** | **68.8±1.3** | **76.0±1.2** | **61.9±2.2** | 71.4±1.6 | 62.8±1.2 | 65.7±0.9 | **69.3** |

Table 20: The graph classification results (in %) on MUTAGENICITY under node domain shift (source→target). P0, P1, P2, and P3 denote the sub-datasets partitioned with node. **Bold** results indicate the best performance. OOM means out of memory.

| Methods | M0→M1 | M1→M0 | M0→M2 | M2→M0 | M0→M3 | M3→M0 | M1→M2 | M2→M1 | M1→M3 | M3→M1 | M2→M3 | M3→M2 | Avg. |
|---|---|---|---|---|---|---|---|---|---|---|---|---|---|
| WL subtree | 78.0 | 68.7 | 70.1 | 70.5 | 59.0 | 61.2 | 71.7 | 78.0 | 49.9 | 56.3 | 69.4 | 71.9 | 67.1 |
| GCN | 74.5±0.2 | 60.8±2.1 | 69.7±0.4 | 68.5±1.7 | 54.1±0.9 | 55.2±0.9 | 68.6±1.6 | 75.5±0.5 | 51.5±1.3 | 46.4±1.7 | 58.6±0.4 | 60.2±0.2 | 61.9 |
| GIN | 77.9±3.1 | 70.7±2.4 | 70.9±0.8 | 69.2±1.2 | 64.1±1.0 | 61.9±2.4 | 78.5±0.2 | 79.8±3.3 | **65.5±2.7** | 71.5±0.9 | 69.5±1.8 | 73.5±2.6 | 71.1 |
| GMT | 67.3±0.2 | 52.5±0.1 | 59.9±0.3 | 47.5±0.2 | 53.5±0.2 | 52.5±0.4 | 59.9±0.1 | 67.3±0.2 | 46.7±0.5 | 51.0±0.3 | 53.3±0.1 | 59.9±0.4 | 57.1 |
| CIN | 70.8±1.1 | 66.9±3.4 | 61.7±0.6 | 62.6±2.4 | 56.3±3.1 | 62.9±1.3 | 65.1±1.0 | 68.8±1.7 | 56.6±1.4 | 66.9±1.0 | 58.1±1.3 | 62.5±0.9 | 63.3 |
| SpikeGCN | 63.2±0.6 | 55.9±1.7 | 59.8±1.2 | 56.5±1.6 | 54.1±2.2 | 60.3±1.2 | 60.1±0.8 | 68.8±1.4 | 54.4±2.0 | 64.2±1.6 | 56.6±0.8 | 58.5±1.5 | 59.4 |
| DRSGNN | 56.9±0.9 | 53.4±1.2 | 58.9±1.0 | 57.7±1.4 | 53.2±1.6 | 61.2±1.5 | 60.6±2.1 | 67.9±1.2 | 57.1±1.4 | 67.3±2.0 | 59.9±1.9 | 57.1±0.8 | 59.3 |
| CDAN | 75.5±0.1 | 71.3±0.4 | 70.7±0.3 | 70.3±0.1 | 58.7±0.6 | 58.4±0.6 | 70.2±0.5 | 76.1±0.5 | 58.5±0.6 | 69.4±1.5 | 59.0±0.1 | 63.7±1.4 | 66.8 |
| ToAlign | 67.3±0.2 | 47.5±0.4 | 59.9±0.6 | 47.5±0.5 | 46.7±0.4 | 47.5±0.2 | 59.9±0.7 | 67.3±0.3 | 46.7±0.1 | 67.3±0.4 | 46.7±0.5 | 59.9±0.3 | 55.4 |
| MetaAlign | 76.5±0.4 | 71.8±1.1 | 71.8±0.8 | 71.4±0.9 | 59.3±0.8 | 63.0±1.0 | 74.2±1.6 | 78.0±0.2 | 61.7±1.2 | 69.9±1.6 | 62.2±0.4 | 68.3±1.5 | 69.0 |
| DEAL | 76.6±0.8 | 68.8±1.0 | 69.9±0.4 | 66.4±0.8 | 59.3±2.1 | 64.2±2.2 | 79.1±0.1 | **81.9±0.6** | 64.5±1.1 | 75.3±0.6 | **69.8±1.6** | **76.5±0.2** | 71.0 |
| CoCo | 75.5±0.4 | 71.7±0.7 | 68.7±1.1 | 69.2±2.0 | 60.8±1.1 | 65.7±0.3 | **79.2±1.2** | 76.8±0.6 | 63.8±0.5 | 73.8±0.4 | 64.6±0.8 | 70.1±1.1 | 70.0 |
| SGDA | OOM | OOM | OOM | OOM | OOM | OOM | OOM | OOM | OOM | OOM | OOM | OOM | OOM |
| DGDA | OOM | OOM | OOM | OOM | OOM | OOM | OOM | OOM | OOM | OOM | OOM | OOM | OOM |
| A2GNN | 55.4±0.6 | 56.3±0.2 | 55.6±0.8 | 55.3±1.1 | 55.9±0.4 | 56.1±0.7 | 55.7±0.6 | 57.1±0.3 | 56.6±1.2 | 55.2±0.7 | 56.8±1.0 | 55.9 | 55.9 |
| PA-BOTH | 55.9±1.0 | 56.0±0.5 | 56.1±0.7 | 56.6±1.2 | 55.9±0.6 | 56.0±0.7 | 57.3±0.8 | 56.8±1.3 | 55.9±1.2 | 56.3±1.0 | 56.4±0.9 | 57.1±1.3 | 56.4 |
| DeSGDA | 76.7±1.0 | **72.5±0.6** | **72.0±2.3** | **71.5±1.5** | **61.0±1.2** | **67.6±0.8** | 75.5±1.0 | 79.8±1.2 | 62.8±1.6 | **75.9±1.3** | 65.5±2.1 | 72.3±1.7 | **71.2** |

Table 21: The graph classification results (in %) on NCI1 under edge density domain shift (source→target). N0, N1, N2, and N3 denote the sub-datasets partitioned with edge density. **Bold** results indicate the best performance. OOM means out of memory.

| Methods | N0→N1 | N1→N0 | N0→N2 | N2→N0 | N0→N3 | N3→N0 | N1→N2 | N2→N1 | N1→N3 | N3→N1 | N2→N3 | N3→N2 | Avg. |
|---|---|---|---|---|---|---|---|---|---|---|---|---|---|
| WL subtree | **72.6** | 80.3 | 62.7 | 75.5 | 52.0 | 63.6 | **69.1** | 69.8 | 70.7 | 59.4 | **80.0** | **70.6** | 68.9 |
| GCN | 49.5±0.4 | 71.1±0.4 | 46.8±0.5 | 33.7±2.8 | 32.7±0.4 | 27.4±0.1 | 56.2±1.5 | 55.3±0.4 | 58.2±1.7 | 51.0±0.2 | 60.7±3.7 | 53.2±0.2 | 49.6 |
| GIN | 67.3±2.7 | 67.9±4.8 | 61.5±4.2 | 65.4±3.7 | 58.9±4.1 | 61.0±3.4 | 62.5±3.2 | 66.2±2.1 | 69.7±0.9 | 56.8±0.7 | 72.4±2.8 | 64.0±1.6 | 64.5 |
| GMT | 50.3±1.2 | 42.5±3.4 | 51.1±3.7 | 42.5±4.5 | 56.1±4.7 | 42.5±4.1 | 53.2±4.9 | 51.0±0.2 | 68.2±0.4 | 51.0±0.3 | 68.2±0.5 | 53.2±0.4 | 52.5 |
| CIN | 51.1±0.2 | 72.6±0.1 | 54.0±0.9 | 72.6±0.2 | 68.2±0.3 | 71.5±1.3 | 55.0±2.1 | 53.5±1.8 | 68.2±0.3 | 52.0±0.1 | 68.3±0.1 | 53.6±0.6 | 61.7 |
| SpikeGCN | 62.8±0.8 | 73.1±1.2 | 61.4±0.8 | 70.9±0.6 | 57.7±1.6 | 66.2±1.1 | 61.2±0.6 | 64.5±1.0 | 62.3±1.4 | 63.9±0.9 | 68.9±1.2 | 60.1±1.0 | 63.6 |
| DRSGNN | 64.3±0.6 | 76.3±0.9 | 56.7±1.1 | 73.2±0.8 | 58.6±1.4 | 63.9±1.8 | 63.0±2.1 | 65.1±1.6 | 64.1±1.9 | 59.2±2.2 | 70.8±2.5 | 56.6±1.4 | 64.3 |
| CDAN | 59.6±0.3 | 73.8±0.5 | 56.7±1.4 | 73.7±0.3 | 71.2±0.4 | 73.2±0.3 | 55.5±0.2 | 57.3±1.1 | 69.9±0.2 | 54.6±2.0 | 69.8±1.4 | 56.6±0.3 | 64.3 |
| ToAlign | 51.0±0.2 | 27.4±0.1 | 53.2±0.4 | 27.4±0.2 | 68.2±0.3 | 27.4±0.3 | 53.2±0.1 | 51.0±0.2 | 68.2±0.2 | 51.0±0.4 | 68.2±0.3 | 53.2±0.2 | 50.0 |
| MetaAlign | 65.0±0.7 | 77.6±1.6 | 62.0±0.6 | 77.1±0.9 | 68.2±0.8 | 74.5±2.0 | 64.2±0.9 | 65.4±0.3 | 68.0±0.3 | 56.1±2.3 | 68.2±0.1 | 66.2±1.1 | 67.7 |
| DEAL | 65.6±0.6 | 73.0±0.9 | 58.0±0.3 | 71.6±1.6 | 60.1±2.8 | 73.1±0.5 | 62.8±1.0 | 65.0±2.4 | 65.8±0.8 | 53.9±2.6 | 57.6±2.8 | 56.7±3.1 | 63.6 |
| CoCo | 70.4±0.7 | **80.4±0.9** | 62.4±0.8 | 75.8±1.2 | 65.7±2.0 | 73.7±0.3 | 67.0±0.8 | **70.4±0.7** | 69.7±0.4 | 62.7±0.9 | 74.4±0.5 | 63.7±0.9 | **69.7** |
| SGDA | OOM | OOM | OOM | OOM | OOM | OOM | OOM | OOM | OOM | OOM | OOM | OOM | OOM |
| DGDA | OOM | OOM | OOM | OOM | OOM | OOM | OOM | OOM | OOM | OOM | OOM | OOM | OOM |
| A2GNN | 59.2±0.8 | 58.7±0.5 | 59.0±1.1 | 58.7±0.6 | 58.9±0.6 | 59.2±1.2 | 58.7±0.6 | 58.6±1.2 | 59.0±1.0 | 59.5±0.6 | 58.7±0.5 | 58.5±1.1 | 58.9 |
| PA-BOTH | 57.6±0.5 | 58.4±0.4 | 58.9±0.6 | 57.4±0.6 | 57.1±1.0 | 58.4±0.5 | 58.0±1.0 | 58.1±0.5 | 58.4±0.6 | 57.7±1.1 | 57.5±0.6 | 58.0±0.4 | 58.0 |
| DeSGDA | 66.0±0.5 | 76.2±0.4 | **62.8±0.6** | **77.6±0.6** | **68.5±1.0** | **74.6±0.5** | 65.1±1.0 | 65.4±0.5 | **70.9±0.6** | **64.8±1.1** | 72.6±0.6 | 66.3±0.4 | 69.3 |

Table 22: The graph classification results (in %) on FRANKENSTEIN under edge density domain shift (source→target). F0, F1, F2, and F3 denote the sub-datasets partitioned with edge density. **Bold** results indicate the best performance. OOM means out of memory.

| Methods | F0→F1 | F1→F0 | F0→F2 | F2→F0 | F0→F3 | F3→F0 | F1→F2 | F2→F1 | F1→F3 | F3→F1 | F2→F3 | F3→F2 | Avg. |
|---|---|---|---|---|---|---|---|---|---|---|---|---|---|
| WL subtree | 71.6 | 72.1 | 62.1 | 71.2 | 57.8 | 67.7 | 64.0 | 75.3 | 41.1 | 59.2 | 55.9 | 55.4 | 62.8 |
| GCN | 66.5±0.4 | 60.0±0.8 | 55.4±0.3 | 60.0±0.1 | 39.6±0.3 | 40.0±0.4 | 55.4±0.2 | 66.5±0.1 | 39.6±0.6 | 33.5±0.3 | 39.6±0.1 | 44.7±0.2 | 50.1 |
| GIN | 71.4±4.7 | 73.4±3.4 | 60.8±2.7 | 66.0±3.4 | 50.5±3.7 | 51.6±1.8 | 64.8±1.0 | 71.3±3.5 | 48.3±4.2 | 57.4±3.8 | 55.1±3.4 | 52.6±4.3 | 60.3 |
| GMT | 67.4±1.0 | 61.7±2.1 | 55.8±0.7 | 57.0±2.4 | 60.2±0.5 | 58.2±2.0 | 57.8±2.1 | 65.7±1.3 | 60.2±0.3 | 57.3±2.3 | 60.7±0.6 | 57.1±1.2 | 59.9 |
| CIN | 70.4±2.8 | 66.5±4.3 | 58.5±2.6 | 64.2±2.7 | 60.6±3.0 | 64.2±3.2 | 58.7±2.4 | 69.1±2.7 | 57.5±3.4 | 67.7±2.1 | 59.5±2.3 | 56.1±1.2 | 62.7 |
| SpikeGCN | 66.5±0.9 | 60.1±1.5 | 55.9±0.8 | 60.2±0.6 | 54.8±2.1 | 59.9±1.6 | 55.8±0.8 | 62.9±1.6 | 58.4±1.2 | 61.1±1.3 | 58.8±1.9 | 62.1±1.5 | 59.8 |
| DRSGNN | 67.5±1.2 | 61.2±0.7 | 55.6±1.4 | 61.1±0.9 | 52.4±2.3 | 61.0±1.2 | 56.9±0.7 | 66.7±1.5 | 60.3±0.5 | 62.0±2.0 | 59.8±1.4 | 59.2±1.1 | 60.3 |
| CDAN | 72.9±0.2 | 74.0±0.3 | 62.7±0.3 | 73.8±0.5 | 61.2±1.0 | 70.0±1.2 | 62.8±0.1 | 73.0±0.3 | 60.6±0.2 | 71.6±1.5 | 60.5±0.2 | 61.1±1.4 | 67.0 |
| ToAlign | 68.0±3.8 | 73.4±2.7 | 64.5±1.1 | 63.7±2.4 | 60.6±1.2 | 61.9±1.3 | 64.8±1.3 | 74.0±1.3 | 60.0±0.6 | 65.7±3.1 | 61.0±1.4 | 56.2±2.3 | 64.5 |
| MetaAlign | 73.6±0.2 | 72.7±1.9 | 63.9±1.0 | 67.9±4.3 | 60.4±0.7 | 65.4±1.8 | 65.2±0.8 | 73.2±2.3 | 60.0±0.6 | 66.7±2.4 | 61.2±1.1 | 56.8±2.1 | 65.6 |
| DEAL | 75.4±0.3 | 74.6±1.1 | 66.1±0.6 | **74.6±0.8** | 53.8±1.0 | 66.8±1.3 | 66.4±0.3 | 73.9±0.6 | 61.6±1.4 | 69.8±0.2 | 60.7±1.0 | 58.3±0.9 | 67.1 |
| CoCo | 74.6±0.9 | **77.2±0.6** | 64.1±3.4 | 73.8±1.1 | 60.5±0.2 | **71.5±0.7** | 65.9±0.5 | **76.0±0.5** | 61.4±0.4 | 72.6±0.6 | 59.6±1.0 | **64.7±1.0** | 68.5 |
| SGDA | 56.6±0.6 | 56.9±0.8 | 55.3±1.2 | 54.6±0.5 | 57.9±1.3 | 58.3±0.4 | 56.1±0.9 | 55.9±0.6 | 54.6±1.3 | 56.7±0.5 | 53.3±0.7 | 56.8±1.1 | 56.1 |
| DGDA | OOM | OOM | OOM | OOM | OOM | OOM | OOM | OOM | OOM | OOM | OOM | OOM | OOM |
| A2GNN | 55.4±0.8 | 56.1±0.6 | 56.7±1.0 | 55.3±0.5 | 54.9±0.7 | 57.2±0.9 | 55.7±0.5 | 56.5±1.3 | 54.5±0.6 | 56.8±0.5 | 56.2±1.0 | 58.8±0.8 | 56.1 |
| PA-BOTH | 56.1±0.5 | 56.0±0.4 | 56.3±0.7 | 56.4±0.4 | 56.0±0.6 | 57.1±0.7 | 56.2±1.1 | 58.3±0.9 | 56.5±0.6 | 57.2±0.9 | 56.9±0.4 | 57.7±0.8 | 56.8 |
| DeSGDA | **75.9±1.0** | 74.7±1.2 | **66.8±1.5** | 71.3±1.9 | **63.2±1.5** | 69.4±1.7 | **66.5±1.3** | 74.3±2.4 | **62.6±1.5** | **73.0±1.8** | **61.6±2.2** | 63.6±1.5 | **68.6** |

Table 23: The graph classification results (in %) on MUTAGENICITY under edge density domain shift (source→target). M0, M1, M2, and M3 denote the sub-datasets partitioned with edge density. **Bold** results indicate the best performance. OOM means out of memory.

| Methods | M0→M1 | M1→M0 | M0→M2 | M2→M0 | M0→M3 | M3→M0 | M1→M2 | M2→M1 | M1→M3 | M3→M1 | M2→M3 | M3→M2 | Avg. |
|---|---|---|---|---|---|---|---|---|---|---|---|---|---|
| WL subtree | 74.9 | 74.8 | 67.3 | 69.9 | 57.8 | 57.9 | 73.7 | **80.2** | 60.0 | 57.9 | 70.2 | 73.1 | 68.1 |
| GCN | 73.0±1.7 | 68.7±1.1 | 66.8±3.5 | 69.2±0.9 | 53.9±3.4 | 53.4±2.7 | 69.3±0.8 | 74.0±1.1 | 55.1±1.3 | 42.6±1.9 | 55.5±3.5 | 57.9±2.9 | 61.6 |
| GIN | 74.1±1.8 | 73.4±3.4 | 65.4±1.5 | 70.4±2.9 | 58.9±2.7 | 61.2±1.1 | 73.2±3.8 | 77.7±3.0 | 63.1±3.7 | 63.9±2.4 | 67.4±2.3 | 73.2±1.9 | 68.5 |
| GMT | 69.0±4.0 | 67.4±3.8 | 60.3±4.2 | 66.5±3.8 | 54.9±1.6 | 54.8±3.6 | 65.6±4.2 | 70.4±3.2 | 64.0±2.3 | 56.8±4.3 | 64.7±1.5 | 61.1±3.5 | 63.0 |
| CIN | 68.5±2.1 | 65.1±2.6 | 65.4±1.3 | 63.6±2.8 | 57.3±3.4 | 59.0±3.1 | 59.3±1.5 | 68.3±1.3 | 58.1±2.4 | 71.1±3.1 | 60.7±1.7 | 61.7±2.4 | 63.2 |
| SpikeGCN | 66.7±1.5 | 65.5±2.0 | 57.9±0.4 | 60.2±1.6 | 53.2±1.4 | 60.1±1.5 | 57.7±1.2 | 67.3±1.5 | 57.7±2.1 | 60.1±1.9 | 59.9±2.4 | 63.3±1.8 | 60.1 |
| DRSGNN | 66.9±1.2 | 62.1±0.7 | 57.1±1.2 | 63.3±2.1 | 56.6±0.9 | 62.1±1.3 | 56.9±1.0 | 67.2±1.8 | 58.1±0.6 | 61.3±2.5 | 58.8±1.0 | 64.7±1.7 | 61.3 |
| CDAN | 74.2±0.3 | 73.7±0.5 | 68.8±0.2 | 71.8±0.4 | 59.9±2.0 | 58.6±1.9 | 70.7±1.4 | 74.3±0.3 | 59.2±1.2 | 69.0±1.6 | 60.0±1.2 | 62.7±1.3 | 66.9 |
| ToAlign | 75.5±1.9 | 67.1±3.8 | 68.1±1.5 | 63.3±2.7 | 55.6±1.2 | 67.3±4.3 | 69.4±3.3 | 77.0±1.2 | 57.6±1.6 | 74.9±2.4 | 59.0±3.3 | 64.6±3.4 | 66.6 |
| MetaAlign | 74.5±0.9 | 73.8±0.6 | 69.4±1.2 | 72.6±1.3 | 59.8±1.8 | 70.7±2.7 | 72.0±0.5 | 75.6±0.6 | 62.4±2.1 | 72.3±1.9 | 62.2±1.1 | 72.0±1.2 | 69.7 |
| DEAL | 76.3±0.2 | 72.4±0.7 | 66.8±1.0 | 72.5±0.7 | 57.6±0.6 | 69.6±1.9 | **77.4±0.6** | 80.0±0.7 | 64.9±0.7 | 72.8±1.4 | **70.3±0.3** | 76.2±1.3 | 71.4 |
| CoCo | **77.5±0.4** | **75.7±1.3** | 68.3±3.7 | 74.9±0.5 | 65.1±2.1 | **74.0±0.4** | 76.9±0.6 | 77.4±3.4 | 66.4±1.5 | 71.2±2.7 | 62.8±4.2 | **77.1±0.6** | **72.2** |
| SGDA | OOM | OOM | OOM | OOM | OOM | OOM | OOM | OOM | OOM | OOM | OOM | OOM | OOM |
| DGDA | OOM | OOM | OOM | OOM | OOM | OOM | OOM | OOM | OOM | OOM | OOM | OOM | OOM |
| A2GNN | 55.3±0.3 | 54.9±0.6 | 55.8±0.4 | 55.1±0.6 | 54.2±1.0 | 57.1±1.2 | 56.1±0.5 | 55.2±0.7 | 57.9±1.5 | 56.3±0.6 | 54.4±0.5 | 58.1±1.5 | 55.8 |
| PA-BOTH | 56.3±0.5 | 57.7±0.9 | 56.9±0.6 | 56.2±1.0 | 55.7±0.8 | 56.5±0.9 | 57.8±1.2 | 56.9±2.1 | 56.5±1.5 | 56.2±1.8 | 56.8±1.4 | 57.4±0.7 | 56.8 |
| DeSGDA | 75.8±1.4 | 74.5±1.7 | **69.5±1.3** | **75.0±2.0** | **61.0±1.5** | 69.2±1.3 | 69.5±1.6 | 76.1±1.5 | **65.0±1.4** | **75.5±2.2** | 63.4±1.8 | 68.3±1.3 | 70.3 |

