# OpenReview forum: "Degree-aware Spiking Graph Domain Adaptation for Classification"
_ICLR.cc/2025/Conference — Submitted to ICLR 2025_

### Official Review · Reviewer_kdE8 · 2024-10-21

**Soundness:** 4
**Presentation:** 4
**Contribution:** 4
**Rating:** 6
**Confidence:** 5

**Summary:**

Spiking Graph Networks (SGNs) help reduce energy use in graph classification but fail with out-of-distribution data. This paper introduces a new framework, DeSGDA, for spiking graph domain adaptation. DeSGDA improves classification by using node degree-aware spiking signals, aligning feature distributions adversarially, and leveraging pseudo-labels from unlabeled data. Experiments show it outperforms other methods.

**Strengths:**

Problem Statement: The paper clearly articulates the domain adaptation problem in SGNs, providing a well-defined background, making it easy to understand.

Innovation: The design of effective pseudo-labels tailored to different distributions is a clever approach.

Methodology: The methods employed are appropriate and rigorous, with a well-reasoned experimental design and a transparent data collection and analysis process.

Writing Quality: The writing is fluent, the structure is logical, and it is easy to read and comprehend.

**Weaknesses:**

The description in figure 1 is too simplistic.

**Questions:**

The paper is well written and I don't have any concerns.

---

> ### Author Response · Authors · 2024-11-23
>
> We are truly grateful for the time you have taken to review our paper, your insightful comments and support. Your positive feedback is incredibly encouraging for us! In the following response, we would like to address your major concern and provide additional clarification.
>
> >Q1. The description in figure 1 is too simplistic.
>
> A1. Thanks for your feedback. We will improve it by adding more details for each compo- nent: the degree-aware personalized spiking representation, adversarial feature alignment using membrane potential, and pseudo-label distillation. We will make these revisions in the final version.
>
> In light of these responses, we hope we have addressed your concerns, and hope you will consider raising your score. If there are any additional notable points of concern that we have not yet addressed, please do not hesitate to share them, and we will promptly attend to those points.

---

> ### Author Response · Authors · 2024-11-26
>
> Dear Reviewer kdE8,
>
> Thank you very much for your thorough review of our manuscript and for your valuable suggestions. We are truly encouraged by your positive evaluation, which has greatly motivated our team. Based on your feedback, we have carefully revised and improved the manuscript to ensure it meets even higher standards of quality and clarity.
>
> If you feel that the improvements we made further enhance the scientific value and readability of the paper, we kindly hope you might consider reflecting this in the scoring. Your support and constructive feedback are invaluable to us, and we deeply appreciate the time and effort you have devoted to reviewing our work.
>
> Thank you once again for your time and thoughtful comments!
>
> Best regards,
>
> All the authors

---

> ### Comment · Reviewer_kdE8 · 2024-11-27
>
> Hi, Authors,
>
> Personally, I appreciate the contributions of this paper, but I also acknowledge the concerns raised by other reviewers, particularly regarding the experimental aspects. Given the current state of the paper, I believe that a 'weak accept' is the most appropriate recommendation.
>
> Thanks,
>
> Reviewer kdE8

---

> > ### Author Response · Authors · 2024-11-27
> >
> > Thank you for your thoughtful feedback and encouraging remarks about our work. We understand the concerns raised by reviewer 9h3R regarding the experimental aspects, and we are actively addressing their questions with detailed and targeted responses. We believe our results sufficiently resolve these concerns and demonstrate the strength of our approach, while also addressing other experimental questions raised by other reviewers. Your positive evaluation, though not fully reflected in the score, is highly motivating for us. We are committed to resolving all concerns and pursuing deeper, meaningful work in this area. Thanks again for your time.
> >
> > Hope you have a nice day!
> >
> > All authors

---

### Official Review · Reviewer_9h3R · 2024-10-31

**Soundness:** 1
**Presentation:** 2
**Contribution:** 2
**Rating:** 5
**Confidence:** 4

**Summary:**

This paper introduces a novel framework named DeSGDA (Degree-aware Spiking Graph Domain Adaptation for Classification) for addressing cross-domain graph classification. DeSGDA tackles the domain adaptation challenge by employing three components: degree-aware personalized spiking representation, adversarial feature distribution alignment, and pseudo-label distillation. Extensive experiments on benchmark datasets demonstrate the superiority of DeSGDA compared to other state-of-the-art methods, highlighting its potential for deployment on low-power devices due to its energy-efficient design.

**Strengths:**

1. This paper is a pioneer work to address domain adaptation in spiking graph neural networks.
2. Degree-aware spiking neural networks (SNNs) is an interesting and effective idea, where the spiking threshold is customized based on node degrees. This degree-dependent approach enhances the model's ability to capture informative node representations, significantly improving its performance in varied domain settings.
3. The authors conduct extensive experiments across various datasets and different domain shift scenarios.
4. The paper provides the theoretical analysis to present DeSGDA's generalization capabilities for cross-domain graph classification.

**Weaknesses:**

1. There shows many inconsistencies in the reported experimental results, which makes me much unconfident about the real performance of proposed method:
    - In Figure 2, the reported results for the "PROTEINS" dataset do not match those presented in Table 1, Table 9, or Table 12. Specifically, in Figure 2, P1-->P0 (GIN) performance is about 0.75~0.76. However, in Table 1, the performance is 84.6; in Table 9, the performance is 78.4; in Table 12, the performance is 84.3. Any of the data in the three tables is inconsistent with the results in Figure 2. There are also some other situations that do not match. This difference raises significant questions about the consistency of experimental results. Further clarification is must to confirm accuracy and reproducibility.
    - In Figure 3, there is no label corresponding to "DeSGDA".

 2. In this paper, the authors first make the assumption that  "Nodes with higher degrees have more neighbors, and the aggregation operation in Eq. 1 allows for more significant feature accumulation, making it easier for these nodes to trigger spikes compared to those with fewer neighbors. " This assumption is valid if a direct summation-based neighbor information aggregation is used. However, it remains unclear if this holds for other aggregation methods, such as GAT which is weighted average based on attention score, and GCN which is weighted average based on structural information. Further discussion on this assumption is needed.

3. I really appreciate the degree-aware personalized spiking representation design. However, this paper aims to tackle with the OOD problem. It shows limited technical contribution for graph domain adaptation. The adversarial learning and pseudo-labeling are both well-investigated method in existing DA works.

4. The experiments primarily focus on binary classification tasks using standard graph classification datasets. More complex experiment setting, e.g., multi-class scenarios rather than all binary classification, is welcome to fully assess the model’s capability.

5. The paper does not adequately explain why DeSGDA outperforms general graph domain adaptation methods, like DEAL, CoCo, and A2GNN. An analysis of how DeSGDA specifically enhance performance beyond standard graph DA approaches would better highlight its advantages.

6. Since the degree-aware component design is crucial  to the performance improvements of DeSGDA, it is recommended to provide specific theoretical analysis for this component to strengthen the core contribution. Of course, I understand that this is often not easy, just as a suggestion that the author can consider for future work.

**Questions:**

Please address my concerns in the weakness part.

Due to concerns about the inconsistency of the experimental results, I will initially give a neutral score. I will decide whether to raise (or lower) my score based on the author's responses during the discussion phase.

---

> ### Author Response · Authors · 2024-11-23
>
> We are truly grateful for the time you have taken to review our paper and your insightful review. Here we address your comments in the following.
>
> >Q1. There shows many inconsistencies in the reported experimental results, which makes me much unconfident about the real performance of proposed method.
>
> A1. Thanks for your question.
>
> 1. The reported results in the main tables of all models are measured and averaged on all samples for five runs, where we provide a more stable performance estimate. In contrast, the results shown in Figure 2 reflect single- run outcomes from our ablation studies, which led to the differences observed. To eliminate the deviations, we will update Figure 2 to report the averaged results presented in the main tables, ensuring better clarity and consistency throughout the manuscript.
>
>
> 2. In Figure 3, "SGNN" actually represents the DeSGDA method. This was an oversight during the figure labeling process, and we are sorry for the confusion.  We will revise it in the final vision.
>
>
> >Q2. This assumption is valid if a direct summation-based neighbor information aggregation is used. However, it remains unclear if this holds for other aggregation methods.
>
> A2. Thanks for your question. We first conduct additional experiments using GIN, GCN, and GAT as the backbone of DeSGDA across four datasets. The results are shown in Table 1-4. Although GCN and GAT use different methods to aggregate information from neighboring nodes, the key operation of them remains message passing. The hypothesis of this paper is that higher-degree nodes can aggregate more information and are therefore more likely to trigger spikes. This hypothesis consistently holds regardless of the aggregation strategies.
>
>
>
> Table 1: The results of DeSGDA with different backbones on PROTEINS dataset.
>
> | Methods        | P0→P1 | P1→P0 | P0→P2 | P2→P0 |
> |----------------|-------|-------|-------|-------|
> | DeSGDA w GCN  | 77.9  | **78.4**  | 73.9  | 76.3  |
> | DeSGDA w GAT  | 77.2  | 76.9  | 72.3  | 74.5  |
> | DeSGDA w GIN  | **78.7**  | 77.4  | **74.8**  | **77.6**  |
>
> Table 2: The results of DeSGDA with different backbones on NCI1 dataset.
>
> | Methods        | N0→N1 | N1→N0 | N0→N2 | N2→N0 |
> |----------------|-------|-------|-------|-------|
> | DeSGDA w GCN  | 67.5  | 70.9  | **70.1**  | 68.6  |
> | DeSGDA w GAT | 66.9  | 70.6  | 68.9  | 67.7  |
> | DeSGDA w GIN  | **68.5**  | **71.4**  | 69.7  | **69.0**  |
>
>
> Table 3: The results of DeSGDA with different backbones on MUTAGENICITY dataset.
>
> | Methods        | M0→M1 | M1→M0 | M0→M2 | M2→M0 |
> |----------------|-------|-------|-------|-------|
> | DeSGDA w GCN  | 64.9  | 65.0  | 64.8  | **64.8**  |
> | DeSGDA w GAT  | 64.6  | 64.7  | **65.3**  | 64.2  |
> | DeSGDA w GIN  | **65.4**  | **65.9**  | 64.2  | 64.5  |
>
>
> Table 4: The results of DeSGDA with different backbones ( GIN, GCN and GAT ) on FRANKENSTEIN dataset.
>
> | Methods        | F0→F1 | F1→F0 | F0→F2 | F2→F0 |
> |----------------|-------|-------|-------|-------|
> | DeSGDA w GCN  | **63.5**  | 63.4  | **62.6**  | 62.4  |
> | DeSGDA w GAT  | 62.8  | 63.9  | 62.2  | 61.9  |
> | DeSGDA w GIN  | 63.3  | **64.1**  | 62.4  | **63.0**  |
>
>
> To further explore the background mechanism of “setting higher thresholds for high-degree nodes and lower thresholds for low-degree node”, we have the following analysis.
>
> Assuming that the node feature follows the normal distribution with $\mathcal{N}(\mu, \sigma^2)$, then for each node in graphs, we follow the message-passing mechanism and have the information aggregation with:
>
>  $h_i=h_i+\sum_{j\in N(i)} w_{ij} h_j$.
>
> Therefore, we have the expect:
>
> $\mathbb{E}(h_i)= \mathbb{E}(h_i) +\sum_{j\in N(i)}w_{ij} \mathbb{E}(h_j)$,
>
> due to $\mathbb{E}(h_j )\sim \mathcal{N}(\mu, \sigma^2)$, we have:
>
> $\mathbb{E}(h_i)\sim \mathcal{N}((1+\sum_{j\in N(i)}w_ij)\mu, (1+\sum_{j\in N(i)}w_{ij})\sigma^2)$.
>
> From the results, we observe that node $ i$  follows a normal distribution with a mean of  $(1 + \sum_{j \in N(i)} w_{ij})\mu$ , determined by the aggregated weights of its neighboring nodes. To provide a more intuitive understanding, we visualize the aggregated neighbor weights of GCN and GIN in Figure 5. The results show that as the degree increases, the aggregated weights also increase progressively. Consequently, high-degree nodes tend to follow a normal distribution with higher mean and variance. In other words, nodes with higher degrees accumulate greater signals, making them more likely to trigger spiking. Based on this, we propose assigning higher thresholds to high-degree nodes and lower thresholds to low-degree nodes.
>
> Another observation is that methods that normalize neighbor weights to 1 (e.g., GAT, GraphSAGE) still result in aggregated features following the same normal distribution. This normalization diminishes the ability to distinguish between nodes with varying degrees, ultimately degrading performance. This explains why, when using GAT as the backbone of DeSGDA, the performance is the weakest.

---

> > ### Author Response · Authors · 2024-11-23
> >
> > >Q3. I really appreciate the degree-aware personalized spiking representation design. However, this paper aims to tackle with the OOD problem. It shows limited technical contribution for graph domain adaptation. The adversarial learning and pseudo-labeling are both well-investigated method in existing DA works.
> >
> > A3. Thank you for your question.
> >
> > **Firstly**, we appreciate your recognition of the degree-aware personalized spiking representation design, which is indeed a significant contribution of our method.
> >
> > **Secondly**, we want to emphasize that the primary contributions of our work lie in introducing a novel problem setup and proposing a framework with theoretical guarantees.
> >
> > **Thirdly**, given the specific scenarios of low-energy consumption and distribution shift, it is crucial to design a framework with simple submodules to meet the low-energy requirements. To this end, we employ straightforward methods, such as adversarial learning and pseudo-labeling, to implement the framework. However, it is important to note that we do not merely use simple methods for implementation but also analyze the underlying mechanisms of our proposed approach.
> >
> > For instance, we thoroughly analyze the rationale behind assigning higher thresholds to high-degree nodes and lower thresholds to low-degree nodes. Additionally, we provide a detailed analysis of the error bound of DeSGDA, demonstrating the framework’s efficiency.
> >
> > In summary, we utilize simple yet effective techniques to address a novel problem while providing insightful analysis of the background mechanisms and model capabilities of our proposed method. Therefore, employing straightforward methods is not a limitation of the paper. On the contrary, solving an important problem with simple methods combined with in-depth analysis is one of our key contributions.
> >
> >
> > >Q4. The experiments primarily focus on binary classification tasks using standard graph classification datasets. More complex experiment setting, e.g., multi-class scenarios rather than all binary classification, is welcome to fully assess the model’s capability.
> >
> >
> > A4. Thanks for your question. To address your concern, we have conducted experiments on SEED dataset [1,2], which is a well-known EEG dataset for multi-class emotion classification. For EEG data processing, we utilized the torcheeg library to
> > convert standard EEG data into graph structures. During graph construction, we introduced complexity by randomly removing edges [4] from each graph and partitioning the dataset into source and target domains based on edge density. The statistics of SEED dataset is shown in Table 1. We compared the DeSGDA model with two general graph neural networks (GCN and GIN) and two graph domain adaptation methods (DEAL and CoCo) under this edge density domain shift. The results in Table 2 show that DeSGDA still can outperform the other methods on most cases.
> >
> > Table 1: Statistics of the experimental datasets.
> >
> > | Datasets | Graphs | Avg. Nodes | Avg. Edges | Classes |
> > |----------|--------|------------|------------|---------|
> > | SEED     | 7,636  | 62.0       | 168.2      | 3       |
> >
> > Table 2: The graph classification results (in %) on SEED under edge density domain shift (source→target). E0, E1 and E2 denote the sub-datasets partitioned with edge density. Bold results indicate the best performance.
> >
> > | Methods | E0→E1 | E1→E0 | E0→E2 | E2→E0 | E1→E2 | E2→E1 |
> > |---------|-------|-------|-------|-------|-------|-------|
> > | GCN     | 46.9  | 48.6  | 47.2  | 50.6  | 51.3  | 50.5  |
> > | GIN     | 49.7  | 50.6  | 49.1  | 51.3  | 52.6  | 51.9  |
> > | DEAL    | 53.9  | **56.4**  | 53.6  | 53.9  | 55.4  | 56.5  |
> > | CoCo    | 54.1  | 53.2  | **54.1**  | 54.3  | 55.2  | 56.1  |
> > | **DeSGDA** | **54.6** | 55.9 | 53.3 | **54.4** | **55.6** | **56.7** |

---

> > > ### Comment · Reviewer_9h3R · 2024-11-24
> > >
> > > According to your EGG experimental results, I would first like to request the variance results, as the current model outcomes are very close to other baselines. Some small improvements could very well be influenced by different random seeds. The combined evaluation of variance from multiple trials would make the results more reliable.

---

> > > > ### Author Response · Authors · 2024-11-26
> > > >
> > > > Thank you for your question. We conducted the experiments five times and reported the average and variance results in Table 9. From these results, it is evident that DeSGDA outperforms other baselines in most cases, demonstrating the superiority of our method.
> > > >
> > > > It is important to note that DeSGDA is deployed in a low-energy consumption scenario. While the spiking-based representation may result in some information loss compared to other methods, DeSGDA leverages a degree-aware personalized spiking representation. This approach enables the generation of more balanced and informative representations across nodes with varying degrees, effectively addressing the inflexible architectural limitations often associated with traditional graph domain adaptation methods, thereby achieving superior performance compared to other methods.
> > > >
> > > > Table 9. The results on SEED under edge density domain shift. E0, E1 and E2 denote the sub-datasets partitioned with edge density.
> > > >
> > > > | Methods    | E0→E1              | E1→E0              | E0→E2              | E2→E0              | E1→E2              | E2→E1              |
> > > > | ---------- | ------------------ | ------------------ | ------------------ | ------------------ | ------------------ | ------------------ |
> > > > | GCN        | 46.0 $\pm$ 0.9     | 47.8 $\pm$ 1.0     | 47.7 $\pm$ 1.4     | 49.7 $\pm$ 0.7     | 51.3 $\pm$ 0.8     | 49.8 $\pm$ 1.0     |
> > > > | GIN        | 48.9 $\pm$ 0.5     | 50.3 $\pm$ 0.6     | 49.5 $\pm$ 0.7     | 50.7 $\pm$ 1.0     | 52.7 $\pm$ 1.0     | 52.1 $\pm$ 0.9     |
> > > > | DEAL       | 53.5 $\pm$ 0.4     | **56.2 $\pm$ 0.7** | 53.2 $\pm$ 0.8     | 53.7 $\pm$ 1.1     | 55.1 $\pm$ 0.8     | 56.0 $\pm$ 0.7     |
> > > > | CoCo       | 53.9 $\pm$ 0.5     | 53.0 $\pm$ 0.6     | 54.1 $\pm$ 0.7     | 54.3 $\pm$ 0.7     | 55.3 $\pm$ 1.0     | 55.9 $\pm$ 0.6     |
> > > > | **DeSGDA** | **54.5 $\pm$ 0.6** | 55.6 $\pm$ 0.7     | **54.6 $\pm$ 0.5** | **54.5 $\pm$ 0.7** | **55.8 $\pm$ 1.1** | **56.6 $\pm$ 0.8** |

---

> > ### Comment · Reviewer_9h3R · 2024-11-24
> >
> > Thanks for your response to my questions. Regarding the inconsistency in the experimental results (Q1), if one is based on five trials and the other on a single trial, the variance of the results in the table is relatively small, so there shouldn't be such a large difference.
> >
> > In Figure 2, not only does P1-->P0 (GIN) not match, but P2-->P0 and P0-->P2 also do not align. This raises significant doubts about your experimental results. Either the variance was incorrectly calculated, and the method actually has a very large variance, or the experiment selectively picked the more favorable model outcomes.

---

> > > ### Author Response · Authors · 2024-11-26
> > >
> > > Thank you for your question. To address your concerns, we have presented the original experimental results in Table 7. The average and variance results are reported in Table 14. Based on these results, we confirm that the GIN results in Figure 2 fall within the range of the five runs. We fully understand your concerns regarding data consistency and sincerely apologize for any confusion. However, upon reviewing the original results, we find that the rerun results shown in Figure 2 are entirely consistent with those reported in Table 14. If you still have concerns about data, we are more than willing to provide all the original experimental data for your verification.
> > >
> > > Additionally, we recognize that comparing the results of a single GCN or GAT run with those of multiple GIN runs may be perceived as unfair. To address this, we have included the mean of the five results for GCN and GAT in Table 8. From these results, it is evident that GIN, as the backbone, outperforms both GCN and GAT.
> > >
> > > Table 7. The original results on PROTEINS under graph flux domain shift (GIN as backbone).
> > >
> > > | P0->P1 | P1->P0 | P0->P2 | P2->P0 |
> > > | ------ | ------ | ------ | ------ |
> > > | 78.91  | 79.46  | 73.62  | 78.33  |
> > > | 76.80  | 77.58  | 75.45  | 76.12  |
> > > | 79.69  | 79.21  | 74.76  | 78.74  |
> > > | 77.95  | 76.71  | 75.92  | 77.01  |
> > > | 80.01  | 78.89  | 74.30  | 77.92  |
> > >
> > >
> > > Table 8. The performance with different GNN architectures on PROTEINS.
> > >
> > > | Method | P0->P1               | P1->P0               | P0->P2               | P2->P0               |
> > > | ------ | -------------------- | -------------------- | -------------------- | -------------------- |
> > > | GCN    | 78.0 $\pm$ 0.7     | 78.3 $\pm$ 0.6 | 73.7 $\pm$ 0.5     | 75.7 $\pm$ 0.8     |
> > > | GAT    | 77.5 $\pm$ 0.7     | 73.8 $\pm$ 1.2      | 73.6 $\pm$ 0.7     | 73.8 $\pm$ 1.6     |
> > > | GIN    | **78.7 $\pm$ 1.3** | **78.4 $\pm$ 1.1** | **74.8 $\pm$ 0.6** | **77.6 $\pm$ 0.9** |

---

> ### Author Response · Authors · 2024-11-23
>
> >Q5. The paper does not adequately explain why DeSGDA outperforms general graph domain adaptation methods.
>
> A5. Thanks for your question. The key advantage of DeSGDA lies in its **degree-aware personalized spiking representations**, which dynamically adjust thresholds based on node degrees. This design allows the model to generate more balanced and informative representations across nodes with varying degrees, addressing the inflexible architecture issues often encountered in traditional graph domain adaptation methods. Specifically, high-degree nodes are assigned higher thresholds to prevent over-representation, while low-degree nodes are given lower thresholds to ensure adequate contribution, resulting in a more robust representation for classification tasks under domain shifts.
>
> Moreover, the integration of theoretical guarantees strengthens DeSGDA’s framework. The adversarial alignment mechanism ensures effective feature distribution alignment between the source and target domains, minimizing domain discrepancies. Additionally, the pseudo-label distillation process further refines the model’s performance in the target domain by iteratively updating and improving the reliability of target predictions. These theoretical foundations provide a clear explanation for the superior performance of DeSGDA in handling domain shifts.
>
> In comparison, general graph domain adaptation methods like DEAL and CoCo lack the personalized adjustment mechanisms and the explicit theoretical analysis that underpin DeSGDA. For instance, DEAL and CoCo focus on aligning global feature spaces but do not address node-level heterogeneity caused by varying degrees. In contrast, DeSGDA’s degree-aware spiking representations and pseudo-label distillation jointly enhance the model’s adaptability, making it more effective in capturing nuanced structural variations across domains.
>
> As demonstrated in our experiments, DeSGDA consistently outperforms DEAL, CoCo, and A2GNN across most datasets and domain shifts. This highlights the practical benefits of combining degree-aware design with robust theoretical underpinnings, which together enable DeSGDA to handle challenging graph domain adaptation scenarios more effectively.
>
>
> >Q6. It is recommended to provide specific theoretical analysis for this component to strengthen the core contribution.
>
> A6. Thanks for your comment. Intuitively, nodes with higher degrees receive more information from neighboring nodes, which makes them more likely to trigger spikes; nodes with lower degrees are less likely to spike.
>
> To further explore the background mechanism of “setting higher thresholds for high-degree nodes and lower thresholds for low-degree node”, we have the following analysis.
>
> Assuming that the node feature follows the normal distribution with $\mathcal{N}(\mu, \sigma^2)$, then for each node in graphs, we follow the message-passing mechanism and have the information aggregation with:
>
>  $h_i=h_i+\sum_{j\in N(i)} w_{ij} h_j$.
>
> Therefore, we have the expect:
>
> $\mathbb{E}(h_i)= \mathbb{E}(h_i) +\sum_{j\in N(i)}w_{ij} \mathbb{E}(h_j)$,
>
> due to $\mathbb{E}(h_j )\sim \mathcal{N}(\mu, \sigma^2)$, we have:
>
> $\mathbb{E}(h_i)\sim \mathcal{N}((1+\sum_{j\in N(i)}w_ij)\mu, (1+\sum_{j\in N(i)}w_{ij})\sigma^2)$.
>
> From the results, we observe that node $ i$  follows a normal distribution with a mean of  $(1 + \sum_{j \in N(i)} w_{ij})\mu$ , determined by the aggregated weights of its neighboring nodes. To provide a more intuitive understanding, we visualize the aggregated neighbor weights of GCN and GIN in Figure 5. The results show that as the degree increases, the aggregated weights also increase progressively. Consequently, high-degree nodes tend to follow a normal distribution with higher mean and variance. In other words, nodes with higher degrees accumulate greater signals, making them more likely to trigger spiking. Based on this, we propose assigning higher thresholds to high-degree nodes and lower thresholds to low-degree nodes.
>
> **Reference:**
>
> [1] Pairwise alignment improves graph domain adaptation. ICML, 2024.
>
> [2] Rethinking propagation for unsupervised graph domain adaptation. AAAI, 2024.
>
> [3] Graph domain adaptation: A generative view. TKDD, 2024.
>
> [4] EEG-based graph neural network classification of Alzheimer’s disease: An empirical evaluation of functional connectivity methods. TNSRE 22.
>
> In light of these responses, we hope we have addressed your concerns, and hope you will consider raising your score. If there are any additional notable points of concern that we have not yet addressed, please do not hesitate to share them, and we will promptly attend to those points.

---

> ### Author Response · Authors · 2024-12-01
>
> Dear reviewer,
>
> We sincerely appreciate your valuable feedback.
>
> As the deadline for the author-reviewer discussion phase is approaching, we would like to check if you have any other remaining concerns about our paper.
>
> We sincerely thank you for your dedication and effort in evaluating our submission. Please do not hesitate to let us know if you need any clarification or have additional suggestions.
>
> Best Regards,
>
> Authors.

---

### Official Review · Reviewer_JxPu · 2024-11-03

**Soundness:** 2
**Presentation:** 3
**Contribution:** 3
**Rating:** 6
**Confidence:** 3

**Summary:**

This paper proposes the problem of spiking graph domain adaptation and introduce a novel framework DeSGDA. This framework enhances the adaptability and performance of SGNs through three key aspects: node degree-aware personalized spiking representation, adversarial feature distribution alignment, and pseudo-label distillation. DeSGDA enables more expressive information capture through degree-dependent spiking thresholds, aligns feature distributions via adversarial training, and utilizes pseudo-labels to leverage unlabeled data effectively.

**Strengths:**

1. The structure of the paper is clear and easy to follow.
2. This paper explores the spiking graph domain adaptation problem, which has been neglected in graph domain adaptation.
3. The paper conducts comprehensive experiments to demonstrate the performance of proposed method.

**Weaknesses:**

1. The novelty seems limited. The core idea of DeSGDA is three parts, i.e., node degree-aware personalized spiking representation, adversarial feature distribution alignment, and pseudo-label distillation. However, the second part is domain alignment[1] and the third part is pseudo-labeling[2, 3]. There are both popular ideas in domain adaptation. The technical contribution is a little weak.
2. The motivation of this paper is not clear. It is recommended that the authors further explain the purpose of using spiking graph neural networks in graph domain adaptation so that readers can understand the contribution of this paper.
3. The authors provide an energy efficiency analysis, which is commendable. Can the authors further compare the training time and memory of DeSGDA with graph domain adaption methods?
[1] Unsupervised domain adaptive graph convolutional networks. WWW, 2020.
[2] Do we really need to access the source data? source hypothesis transfer for unsupervised domain adaptation. ICML, 2020.
[3] Coco: A coupled contrastive framework for unsupervised domain adaptive graph classification. ICML, 2023.

**Questions:**

1. From Tables 1 and 2 in the experimental section, the WL subtree method achieves better performance in many cases than the well-designed graph domain adaptation methods. Can the author explain why?
2. In Figure 3, the DeSGDA method is not found. Does SGNN in the figure represents DeSGDA?
3. What is the effect of directly replacing the spiking graph neural networks with commonly used graph neural networks, such as GIN and GCN, combined with adversarial feature distribution alignment and pseudo-label distillation?

---

> ### Author Response · Authors · 2024-11-23
>
> We are truly grateful for the time you have taken to review our paper and your insightful review. Here we address your comments in the following.
>
>
> >Q1. The novelty seems limited.
>
> A1. Thanks for your question.
>
> **Firstly**, we emphasize that the degree-aware personalized spiking representation design is a significant contribution of the proposed method. This design enables the model to generate more balanced and informative representations across nodes with varying degrees, effectively addressing the inflexible architectural issues commonly encountered in traditional graph domain adaptation methods.
>
> **Secondly**, we want to emphasize that the primary contributions of our work lie in introducing a novel problem setup and proposing a framework with theoretical guarantees.
>
> **Thirdly**, given the specific scenarios of low-energy consumption and distribution shift, it is crucial to design a framework with simple submodules to meet the low-energy requirements. To this end, we employ straightforward methods, such as adversarial learning and pseudo-labeling, to implement the framework. However, it is important to note that we do not merely use simple methods for implementation but also analyze the underlying mechanisms of our proposed approach.
>
> For instance, we thoroughly analyze the rationale behind assigning higher thresholds to high-degree nodes and lower thresholds to low-degree nodes. Additionally, we provide a detailed analysis of the error bound of DeSGDA, demonstrating the framework’s efficiency.
>
> In summary, we utilize simple yet effective techniques to address a novel problem while providing insightful analysis of the background mechanisms and model capabilities of our proposed method. Therefore, employing straightforward methods is not a limitation of the paper. On the contrary, solving an important problem with simple methods combined with in-depth analysis is one of our key contributions.
>
>
>
>
> >Q2. The motivation of this paper is not clear. It is recommended that the authors further explain the purpose of using spiking graph neural networks in graph domain adaptation so that readers can understand the contribution of this paper.
>
> A2. Thanks for your question. We explain the motivation of this paper from an application perspective, highlighting the real-world relevance of Spiking Graph Networks (SGNs) and domain adaptation in addressing critical challenges like low energy consumption and distribution shifts:
>
> In this paper, we focus on implementing energy-efficient Spiking Graph Networks (SGNs) under distribution shift circumstances. Both domain adaptation and SGNs are well-suited for real-world scenarios where data distributions change across environments and graph-structured, dynamic data must be processed efficiently under resource constraints. These challenges are prevalent in various applications that simultaneously demand solutions for distribution shifts and low energy consumption. For instance:
>
> **IoT and Sensor Network Applications.**
> In Internet of Things (IoT) and sensor network applications, domain adaptation and Spiking Graph Neural Networks (SGNNs) complement each other to tackle the challenges of dynamic, resource-constrained environments. Domain adaptation allows models to transfer knowledge across different configurations or locations (e.g., geographic regions or sensor setups) without extensive retraining, effectively handling distribution shifts. Meanwhile, SGNNs process graph-structured data, where nodes represent devices or sensors and edges capture communication links, using event-driven computation to minimize energy consumption. This synergy enables robust, scalable, and low-power solutions, making SGNNs ideal for long-term IoT and sensor network deployments.
>
> **Environmental Monitoring.**
> Environmental monitoring systems face dynamic and resource-constrained settings, where domain adaptation and SGNNs provide an effective combination. Domain adaptation allows models to transfer knowledge across regions with varying conditions, such as different weather patterns or pollutant distributions, reducing the need for costly retraining. SGNNs efficiently process graph-structured data, where nodes represent monitoring stations or sensors and edges capture spatial, temporal, or communication relationships, leveraging event-driven computation to significantly minimize energy use. Together, these technologies enable scalable, robust, and energy-efficient environmental monitoring systems suitable for diverse and remote deployment scenarios.
>
> By addressing these practical challenges, the proposed method integrates domain adaptation with SGNs to deliver energy-efficient and distribution-robust graph learning solutions. This application-driven perspective highlights the relevance and contribution of our work in real-world scenarios.

---

> > ### Author Response · Authors · 2024-11-23
> >
> > >Q3. The authors provide an energy efficiency analysis, which is commendable. Can the authors further compare the training time and memory of DeSGDA with graph domain adaption methods? Do we really need to access the source data?
> >
> > A3. Thanks for your questions.
> >
> > 1.	For the first question, we would like to emphasize that the scenarios we investigated focus on low-energy devices. Our research prioritizes the inference performance of SGNs, aiming for low power consumption and long battery life rather than the computational cost of training. To address your concerns, we provide detailed comparisons of GPU memory consumption and training time per epoch for DeSGDA and other graph domain adaptation methods under identical experimental settings, as shown in Tables 1 and 2. It is worth noting that the training phase is typically conducted on more powerful hardware to achieve optimal performance within a reasonable time frame. However, the energy consumption during training is not the primary focus of our study.
> >
> > 2.	For the second question, the accessing of source data is crucial for the domain alignment. In adversarial distribution alignment module, source data helps minimize feature discrepancies between source and target domains.
> >
> >
> > Table 1: GPU memory consumption of different graph domain methods in training stage for each training epoch (in GB).
> >
> > | Dataset        | DeSGDA | DEAL | CoCo | A2GNN | PA-BOTH |
> > |----------------|--------|------|------|-------|---------|
> > | PROTEINS       | 2.3    | 1.4  | 1.2  | 33.1  | 5.6     |
> > | NCI1           | 5.7    | 2.9  | 3.3  | 34.8  | 11.9    |
> > | MUTAGENICITY   | 5.0    | 2.8  | 3.2  | 35.0  | 12.3    |
> > | FRANKENSTEIN   | 2.9    | 2.4  | 1.9  | 33.6  | 7.3     |
> >
> > Table 2: Time consumption of different graph domain methods in training stage for each training epoch (in seconds).
> >
> > | Dataset        | DeSGDA  | DEAL  | CoCo    | A2GNN  | PA-BOTH |
> > |----------------|---------|-------|---------|--------|---------|
> > | PROTEINS       | 0.587   | 0.126 | 22.123  | 0.869  | 0.283   |
> > | NCI1           | 0.855   | 0.518 | 58.564  | 1.483  | 0.597   |
> > | MUTAGENICITY   | 0.887   | 0.612 | 52.740  | 1.553  | 0.511   |
> > | FRANKENSTEIN   | 0.663   | 0.375 | 26.837  | 1.016  | 0.275   |
> >
> >
> >
> > >Q4. From Tables 1 and 2 in the experimental section, the WL subtree method achieves better performance in many cases than the well-designed graph domain adaptation methods. Can the author explain why?
> >
> > A4. Thank you for your question. There is still limited research focusing on the graph domain adaptation problem (e.g., DEAL, CoCo). Therefore, we adapted node classification methods for graph classification tasks (e.g., SGDA, DGDA, A2GNN, PA-BOTH). While the WL subtree method outperforms node classification-based methods, it remains inferior to graph domain adaptation methods specifically designed for graph classification tasks.
> >
> > >Q5. In Figure 3, the DeSGDA method is not found. Does SGNN in the figure represents DeSGDA?
> >
> > A5. Sorry for the confusion. In Figure 3, "SGNN" actually represents the DeSGDA method. This was an oversight during the figure labeling process, and we are sorry for the confusion. We will correct it in the revised manuscript.

---

> ### Author Response · Authors · 2024-11-23
>
> >Q6. What is the effect of directly replacing the spiking graph neural networks with com- monly used graph neural networks, such as GIN and GCN, combined with adversarial feature distribution alignment and pseudo-label distillation?
>
> A6. Thanks for your question. First, we have conducted ablation studies to examine the effect of directly replacing the SGNs with commonly used Graph Neural Networks (GNNs) for generating representations for DeSGDA: (1) DeSGDA w GCN: It replaces SGNs with GCN; (2) DeSGDA w GIN: It replaces SGNs with GIN; (3) DeSGDA w SAGE: It replaces SGNs with GraphSAGE. The experimental results across the PROTEINS, NCI1, MUTAGENICITY, and FRANKENSTEIN datasets are shown in Table 3, 4, 5 and 6.
> However, the critical aspect of our work lies in the specific problem we set up, i.e., low-power and distribution shift environments. In this context, directly replacing SGNs with commonly used GNNs like GIN or GCN is not feasible, as these models are unsuitable for deployment on low-energy devices. As demonstrated in Section 5.3, GNN based methods have much higher energy consumption than the spike based methods.
>
>
> Table 3: The results of DeSGDA with different GNNs (GIN, GCN and SAGE) on PROTEINS dataset. Bold results indicate the best performance. Bold results indicate the best performance.
>
> | Methods            | P0→P1 | P1→P0 | P0→P2 | P2→P0 |
> |--------------------|-------|-------|-------|-------|
> | DeSGDA w GCN      | 76.6  | 70.5  | 71.8  | 74.1  |
> | DeSGDA w SAGE     | 75.8  | 73.3  | 72.4  | 75.2  |
> | DeSGDA w GIN      | 77.3  | 75.8  | 73.8  | 77.1  |
> | DeSGDA            | **78.7**  | **78.4**  | **74.8** | **77.6**  |
>
>
>
> Table 4: The results of DeSGDA with different GNNs (GIN, GCN and SAGE) on NCI1 dataset. Bold results indicate the best performance. Bold results indicate the best performance.
>
> | Methods            | N0→N1 | N1→N0 | N0→N2 | N2→N0 |
> |--------------------|-------|-------|-------|-------|
> | DeSGDA w GCN      | 66.3  | 68.3  | 68.5  | 67.1  |
> | DeSGDA w SAGE     | 67.2  | 69.5  | 66.6  | 68.4  |
> | DeSGDA w GIN      | **69.0**  | 69.8  | 68.8  | 68.8  |
> | DeSGDA            | 68.5  | **71.4**  | **70.1**  | **69.0**  |
>
> Table 5: The results of DeSGDA with different GNNs (GIN, GCN and SAGE) on MUTAGENICITY dataset. Bold results indicate the best performance. Bold results indicate the best performance.
>
> | Methods            | M0→M1 | M1→M0 | M0→M2 | M2→M0 |
> |--------------------|-------|-------|-------|-------|
> | DeSGDA w GCN      | 60.9  | 60.7  | 63.6  | 60.6  |
> | DeSGDA w SAGE     | 61.1  | 62.3  | 64.2  | 61.1  |
> | DeSGDA w GIN      | 64.5  | 65.4  | 65.0  | 63.9  |
> | DeSGDA            | **65.4**  | **65.9**  | **65.5**  | **65.6**  |
>
>
> Table 6: The results of DeSGDA with different GNNs (GIN, GCN, SAGE) on FRANKENSTEIN dataset. Bold results indicate the best performance. Bold results indicate the best performance.
>
> | Methods            | F0→F1 | F1→F0 | F0→F2 | F2→F0 |
> |--------------------|-------|-------|-------|-------|
> | DeSGDA w GCN      | 61.3  | 62.2  | 60.7  | 61.1  |
> | DeSGDA w SAGE     | 61.9  | 62.6  | 61.3  | 61.6  |
> | DeSGDA w GIN      | 62.8  | 63.6  | 61.8  | 62.8  |
> | DeSGDA            | **63.5**  | **64.1**  | **62.6**  | **63.0**  |
>
>
> In light of these responses, we hope we have addressed your concerns, and hope you will consider raising your score. If there are any additional notable points of concern that we have not yet addressed, please do not hesitate to share them, and we will promptly attend to those points.

---

> ### Comment · Reviewer_JxPu · 2024-11-25
>
> Thanks for the authors' responses which addressed most of my concerns. However, I hope our discussions can be included in the revised version. I will boost my score to 6.

---

> > ### Author Response · Authors · 2024-11-26
> >
> > Thank you for your valuable feedback and suggestions. We have incorporated the discussed content into the main text and appendix of the manuscript as per your request. The revisions are highlighted in blue to make it easier for you to review the changes.
> >
> > We kindly ask you to check the updated sections and let us know if further adjustments are needed. We truly appreciate your time and effort in reviewing our work and providing constructive input.

---

### Official Review · Reviewer_TfEH · 2024-11-04

**Soundness:** 3
**Presentation:** 3
**Contribution:** 2
**Rating:** 6
**Confidence:** 4

**Summary:**

This paper proposes a framework called DeSGDA to tackle the problem of domain adaptation in spiking graph neural networks. DeSGDA combines three key components: a degree-aware spiking representation that adapts spiking thresholds based on node degree, adversarial distribution alignment to minimize discrepancies between source and target domains, and pseudo-label distillation to improve model performance on unlabeled target data. The authors provide theoretical bounds on the generalization error of their framework and validate its effectiveness through extensive experiments on multiple benchmark datasets, demonstrating DeSGDA’s superiority over other baseline methods in terms of classification accuracy and energy efficiency.

**Strengths:**

1. The DeSGDA framework is a multi-faceted approach that includes personalized spiking representation, adversarial distribution alignment, and pseudo-label distillation.
2. The authors provide theoretical bounds on the generalization error for spiking graph domain adaptation.
3. The paper conducts extensive experiments on several benchmark datasets, comparing DeSGDA with a wide range of competitive baselines. This comprehensive evaluation demonstrates the model’s effectiveness and superiority over other methods.

**Weaknesses:**

1. This paper studies an A+B problem. The authors bring together two distinct challenges—spiking neural networks and domain adaptation—within the context of graph data. However, the novelty of this problem setup raises questions about the practical relevance and contribution of the paper, as such a scenario may be uncommon in real-world applications.
2. The authors mention in the Introduction that this problem may exist in Electroencephalography (EEG) data, however, they did not conduct experiments on such data. The datasets used in this paper are mainly protein, molecular, and chemical graphs, can the authors give explanations about what are the applicable scenarios on these data? especially given that, in these domains, accuracy often takes precedence over timeliness.
3. The adversarial learning on both source and target domains, and the pseudo-labeling strategy on target graphs are very familiar techniques in domain adaptation methods. The contribution is limited.
4. The methodology assumes that the node degree is a key factor for domain adaptation in spiking graph networks. More explanation on why “setting higher thresholds for high-degree nodes and lower thresholds for low-degree node” should be added.
5. In some real-world datasets, besides the degree, other structural or feature-based factors might impact the domain shift. Focusing heavily on degree-aware thresholds may overlook other graph properties.
6. The definition of V_th is not given.

**Questions:**

see above.

---

> ### Author Response · Authors · 2024-11-23
>
> We are truly grateful for the time you have taken to review our paper and your insightful review. Here we address your comments in the following.
>
> >Q1. This paper studies an A+B problem. The authors bring together two distinct challenges—spiking neural networks and domain adaptation—within the context of graph data. However, the novelty of this problem setup raises questions about the practical relevance and contribution of the paper, as such a scenario may be uncommon in real-world applications.
>
> A1. Thanks for your feedback. In this paper, we focus on implementing energy-efficient Spiking Graph Networks (SGNs) under distribution shift circumstances. Domain adaptation and SGNs are well-suited for scenarios where data distributions change across environments, and graph-structured, dynamic data must be processed efficiently under resource constraints. In the real-world, various applications require the consideration of distribution shifts and low energy consumption problems simultaneously. For instance:
>
> (1)	In IoT and sensor network applications, domain adaptation and spiking graph neural networks (SGNNs) complement each other to address challenges in dynamic, resource- constrained environments. Domain adaptation enables models to transfer knowledge across different settings (e.g., geographic regions or sensor configurations) without extensive retraining, handling shifts in data distribution. Meanwhile, SGNNs efficiently process graph- structured data, where nodes represent devices or sensors and edges capture communication links, using event-driven computation to minimize energy consumption. Together, they enable robust, scalable, and low-power performance, making them ideal for long-term IoT and sensor network deployments.
>
> (2)	In environmental monitoring, domain adaptation and spiking graph neural networks (SGNNs) are crucial for addressing challenges in dynamic and resource-constrained set- tings. Domain adaptation allows models to transfer knowledge across regions with varying conditions, such as different weather patterns or pollutant distributions, without requiring extensive retraining. SGNNs efficiently process graph-structured data, where nodes repre
> sent monitoring stations or sensors and edges capture spatial, temporal, or communication relationships, using event-driven computation to minimize energy consumption. Together, they enable scalable, robust, and low-power environmental monitoring systems suitable for diverse and remote deployment scenarios.
>
> >Q2. The authors mention in the Introduction that this problem may exist in Electroen- cephalography (EEG) data, however, they did not conduct experiments on such data. The datasets used in this paper are mainly protein, molecular, and chemical graphs, can the authors give explanations about what are the applicable scenarios on these data? especially given that, in these domains, accuracy often takes precedence over timeliness.
>
> A2. Thanks for your question. To address your concern, we conducted additional experiments using the SEED dataset [1, 2], which is a well-known EEG dataset for emotion classification. For EEG data processing, we utilized the **torcheeg** library to convert standard EEG data into graph structures. During graph construction, we remove edges from each graph as introduced in [3], and partitioning the dataset into source and target domains based on edge density. The statistics of SEED dataset is shown in Table 1. We compared the DeSGDA model with two general graph neural networks (GCN and GIN) and two graph domain adaptation methods (DEAL and CoCo). The results in Table 2 show that DeSGDA still outperforms the other methods in most cases.
>
> Table 1: Statistics of the experimental datasets.
>
> | Datasets | Graphs | Avg. Nodes | Avg. Edges | Classes |
> |----------|--------|------------|------------|---------|
> | SEED     | 7,636  | 62.0       | 168.2      | 3       |
>
> Table 2: The graph classification results (in %) on SEED under edge density domain shift (source→target). E0, E1 and E2 denote the sub-datasets partitioned with edge density.
>
> | Methods | E0→E1 | E1→E0 | E0→E2 | E2→E0 | E1→E2 | E2→E1 |
> |---------|-------|-------|-------|-------|-------|-------|
> | GCN     | 46.9  | 48.6  | 47.2  | 50.6  | 51.3  | 50.5  |
> | GIN     | 49.7  | 50.6  | 49.1  | 51.3  | 52.6  | 51.9  |
> | DEAL    | 53.9  | **56.4**  | 53.6  | 53.9  | 55.4  | 56.5  |
> | CoCo    | 54.1  | 53.2  | **54.1**  | 54.3  | 55.2  | 56.1  |
> | **DeSGDA** | **54.6** | 55.9 | 53.3 | **54.4** | **55.6** | **56.7** |
>
> Since collecting real-world data is often expensive (e.g., in IoT, sensor networks, and environmental monitoring), we follow the methodology outlined in [4, 5] and utilize commonly used publicly available datasets to test our model. Our objective is to evaluate the model’s effectiveness across diverse datasets, demonstrating its potential to achieve robust performance in real-world application scenarios with practical requirements.

---

> > ### Author Response · Authors · 2024-11-23
> >
> > >Q3. The adversarial learning on both source and target domains, and the pseudo- labeling strategy on target graphs are very familiar techniques in domain adaptation methods. The contribution is limited.
> >
> > A3. Thanks for your question.
> >
> > **Firstly**, we emphasize that the degree-aware personalized spiking representation design is a significant contribution of the proposed method. This design enables the model to generate more balanced and informative representations across nodes with varying degrees, effectively addressing the inflexible architectural issues commonly encountered in traditional graph domain adaptation methods.
> >
> > **Secondly**, we want to emphasize that the primary contributions of our work lie in introducing a novel problem setup and proposing a framework with theoretical guarantees.
> >
> > **Thirdly**, given the specific scenarios of low-energy consumption and distribution shift, it is crucial to design a framework with simple submodules to meet the low-energy requirements. To this end, we employ straightforward methods, such as adversarial learning and pseudo-labeling, to implement the framework. However, it is important to note that we do not merely use simple methods for implementation but also analyze the underlying mechanisms of our proposed approach.
> >
> > For instance, we thoroughly analyze the rationale behind assigning higher thresholds to high-degree nodes and lower thresholds to low-degree nodes. Additionally, we provide a detailed analysis of the error bound of DeSGDA, demonstrating the framework’s efficiency.
> >
> > In summary, we utilize simple yet effective techniques to address a novel problem while providing insightful analysis of the background mechanisms and model capabilities of our proposed method. Therefore, employing straightforward methods is not a limitation of the paper. On the contrary, solving an important problem with simple methods combined with in-depth analysis is one of our key contributions.
> >
> >
> > >Q4. The methodology assumes that the node degree is a key factor for domain adaptation in spiking graph networks. More explanation on why “setting higher thresholds for high-degree nodes and lower thresholds for low-degree nodes” should be added.
> >
> > A4. Thanks for your comment. Intuitively, nodes with higher degrees receive more information from neighboring nodes, which makes them more likely to trigger spikes; nodes with lower degrees are less likely to spike.
> >
> > To further explore the background mechanism of “setting higher thresholds for high-degree nodes and lower thresholds for low-degree nodes”, we have the following analysis.
> >
> > Assuming that the node feature follows the normal distribution with $\mathcal{N}(\mu, \sigma^2)$, then for each node in graphs, we follow the message-passing mechanism and have the information aggregation with:
> >
> >  $h_i=h_i+\sum_{j\in N(i)} w_{ij} h_j$.
> >
> > Therefore, we have the expect:
> >
> > $\mathbb{E}(h_i)= \mathbb{E}(h_i) +\sum_{j\in N(i)}w_{ij} \mathbb{E}(h_j)$,
> >
> > due to $\mathbb{E}(h_j )\sim \mathcal{N}(\mu, \sigma^2)$, we have:
> >
> > $\mathbb{E}(h_i)\sim \mathcal{N}((1+\sum_{j\in N(i)}w_{ij})\mu, (1+\sum_{j\in N(i)}w_{ij})\sigma^2)$.
> >
> > From the results, we observe that node $ i$  follows a normal distribution with a mean of  $(1 + \sum_{j \in N(i)} w_{ij})\mu$ , determined by the aggregated weights of its neighboring nodes. To provide a more intuitive understanding, we visualize the aggregated neighbor weights of GCN and GIN in Figure 5. The results show that as the degree increases, the aggregated weights also increase progressively. Consequently, high-degree nodes tend to follow a normal distribution with higher mean and variance. In other words, nodes with higher degrees accumulate greater signals, making them more likely to trigger spiking. Based on this, we propose assigning higher thresholds to high-degree nodes and lower thresholds to low-degree nodes.
> >
> > Another observation is that methods that normalize neighbor weights to 1 (e.g., GAT, GraphSAGE) still result in aggregated features following the same normal distribution. This normalization diminishes the ability to distinguish between nodes with varying degrees, ultimately degrading performance. This explains why, when using GAT as the backbone of DeSGDA, the performance is the weakest.

---

> ### Author Response · Authors · 2024-11-23
>
> >Q5. In some real-world datasets, besides the degree, other structural or feature-based factors might impact the domain shift. Focusing heavily on degree-aware thresholds may overlook other graph properties.
>
> A5. Thanks for your question. In our proposed DeSGDA framework, the primary purpose of setting degree-aware thresholds is to generate personalized spiking representations that capture structural information and enhance classification performance. However, as the reviewer correctly pointed out, other structural or feature-based factors could also significantly influence domain shifts. To address your concern, our method incorporates the following mechanisms:
>
> **Adversarial Distribution Alignment**: We employ adversarial training to align the feature distributions of the source and target domains. This approach goes beyond degree-dependent spiking representations by aligning the overall distributions across domains, effectively mitigating the impact of domain shifts. This alignment ensures that features from both domains are represented in a unified feature space, addressing potential biases introduced by relying solely on degree-aware thresholds.
>
> **Integrated Framework Design:** Our method does not treat degree-aware thresholding and domain alignment as isolated components. Instead, they are integrated into a unified framework where the aligned distributions serve as the foundation for learning improved spiking graph representations. This holistic design ensures that degree-aware thresholding and feature distribution alignment complement each other, working synergistically to improve performance under domain shifts.
>
> **Pseudo-Label Distillation:** To further address domain discrepancies, we introduce a pseudo-label distillation module. This module leverages reliable pseudo-labels from the target domain to guide the adjustment of degree thresholds and improve spiking representations. This mechanism effectively compensates for unseen or uncommon degrees in the target domain, enhancing the robustness of our approach.
>
> In summary, degree-aware thresholding is not an isolated feature extraction method but an integral part of a cohesive framework that includes adversarial distribution alignment and pseudo-label distillation. Together, these components enable the model to learn more effective spiking graph representations while addressing domain shifts comprehensively. By aligning distributions and improving representations simultaneously, our method achieves superior efficiency in handling domain adaptation tasks.
>
>
> >Q6. The definition of Vth is not given.
>
> A6. Thanks for your question. In the preliminaries section of our paper, $V_th$ is defined as the threshold voltage in SGNs. It represents the value that the membrane potential must exceed for a spike to be generated. In the section of Degree-aware personalized spiking representation, $V_th$ is updated in Equation (2) of our paper:
> $s_\tau^{d_i^s}=\mathbb{H}(u_\tau-V_{th}^{d_i^s}), S^{d_i^s}=avg(s_\tau^{d_i^s}), V_{th}^{d_i^s}=(1-\alpha) V_{th}^{d_i^s}+\alpha S^{d_i^s},$
>
> where $V_{th}^{d_i^s}$ is the threshold of degree $d_i^s\in D^s$, initially set to $V_{th}$, and $\alpha$ is a hyper-parameter. The $avg(\cdot)$ operation takes the average of spiking representation with degree $d_i^s$. Consequently, by updating the value of $V_{th}$, high-degree nodes tend to achieve high $S^{d_i^s}$, which leads to an iterative increase in the threshold corresponding to degree $d_i^s$ and conversely for lower-degree nodes.
>
>
>
>
>
>
>
> **Reference:**
>
> [1] Differential Entropy Feature for EEG-based Emotion Classification. NER.
>
> [2]Investigating Critical Frequency Bands and Channels for EEG-based Emotion Recognition with Deep Neural Networks. TAMD.
>
> [3] EEG-based graph neural network classification of Alzheimer’s disease: An empirical evaluation of functional connectivity methods. TNSRE 22.
>
> [4] Spiking Graph Convolutional Networks. IJCAI 2022.
>
> [5] Dynamic Reactive Spiking Graph Neural Network. AAAI 2024.
>
>
> In light of these responses, we hope we have addressed your concerns, and hope you will consider raising your score. If there are any additional notable points of concern that we have not yet addressed, please do not hesitate to share them, and we will promptly attend to those points.

---

> > ### Comment · Reviewer_TfEH · 2024-11-25
> >
> > Thank you to the author for the detailed response, which has resolved most of my concerns. I will increase my score. However, I still feel that for a task on Spiking Graph Networks, experiments should be conducted on datasets where process efficiency is crucial, like signal datasets. I suggest that the authors include the EEG experiments in the paper.

---

> > > ### Author Response · Authors · 2024-11-25
> > >
> > > Thank you for your recognition of our response and for your valuable suggestions! Regarding your suggestion to conduct experiments on signal datasets such as EEG, we fully agree with its importance. We have already started working on these experiments to demonstrate the applicability and efficiency of our method in this critical domain. Once the experiments are completed, we will include the results in the paper to enhance its comprehensiveness and impact.
> > >
> > > Thank you again for your thoughtful review and professional insights!

---

### Meta-Review · Area_Chair_v68y · 2024-12-18

**Metareview:**

### Summary
The paper proposes DeSGDA, a framework for addressing domain adaptation in spiking graph neural networks (SGNNs). DeSGDA incorporates three key components: (1) degree-aware personalized spiking representation, which adapts spiking thresholds based on node degrees, (2) adversarial feature distribution alignment, to reduce source-target domain discrepancies, and (3) pseudo-label distillation, to improve performance on unlabeled target data. The authors provide theoretical bounds for the generalization error and validate DeSGDA through experiments on benchmark datasets, demonstrating improved classification accuracy and energy efficiency over baseline methods.

### Strengths
- The paper is one of the first to explore domain adaptation in SGNNs, combining spiking neural networks with graph domain adaptation techniques.
- DeSGDA is designed with low-power applications in mind, leveraging spiking representation for energy-efficient computations.
- The paper evaluates DeSGDA across multiple datasets and domain shift scenarios, with theoretical generalization bounds supporting its methodology.

### Weaknesses
- Limited Novelty in Core Techniques:The adversarial distribution alignment and pseudo-labeling strategies are widely used in domain adaptation literature, contributing little innovation.
The degree-aware spiking representation is specific but lacks theoretical justification or exploration of alternatives (e.g., beyond node degree).

- While the problem setup (spiking graph domain adaptation) is novel, its real-world applicability is unclear, especially in datasets like protein or molecular graphs, where accuracy often takes precedence over timeliness or energy efficiency. The paper mentions potential applications in EEG data but does not include experiments on such data.

- Results reported in different tables and figures (e.g., Table 1, Figure 2) are inconsistent, raising concerns about the reliability of the findings. The experiments focus primarily on binary classification tasks using standard datasets, without exploring multi-class scenarios or more complex settings. The paper does not adequately compare DeSGDA against state-of-the-art graph domain adaptation methods (e.g., DEAL, CoCo, A2GNN), limiting its evaluation scope.


While the paper addresses a novel intersection of spiking graph neural networks and domain adaptation, its contributions are limited by reliance on well-established techniques, questionable practical relevance, and experimental inconsistencies. Expanding the applicability, providing stronger theoretical insights, and ensuring experimental rigor are necessary to strengthen the paper’s impact.

**Additional Comments On Reviewer Discussion:**

The major concerns raised by the reviewers are summarized as above weaknesses. Although the authors tried to address the concerns, there is a crucial reliability issue in the reported results. Moreover, Reviewer kdE8's review lacked sufficient detail to substantiate the positive rating, so it is intentionally neglected in the final decision.

---

### Decision · Program_Chairs · 2025-01-22

Reject